# Distinct cytoskeletal proteins define zones of enhanced cell wall synthesis in *Helicobacter pylori*

Jennifer A Taylor[1,2], Benjamin P Bratton[3,4], Sophie R Sichel[2,5], Kris M Blair[2,6], Holly M Jacobs[2,6], Kristen E DeMeester[7], Erkin Kuru[8], Joe Gray[9], Jacob Biboy[10], Michael S VanNieuwenhze[11], Waldemar Vollmer[10], Catherine L Grimes[7,12], Joshua W Shaevitz[3,13], Nina R Salama[1,2,5,6]*

[1]Department of Microbiology, University of Washington, Seattle, United States; [2]Human Biology Division, Fred Hutchinson Cancer Research Center, Seattle, United States; [3]Lewis-Sigler Institute for Integrative Genomics, Princeton University, Princeton, United States; [4]Department of Molecular Biology, Princeton University, Princeton, United States; [5]Molecular Medicine and Mechanisms of Disease Graduate Program, University of Washington, Seattle, United States; [6]Molecular and Cellular Biology Graduate Program, University of Washington, Seattle, United States; [7]Department of Chemistry and Biochemistry, University of Delaware, Newark, United States; [8]Department of Genetics, Harvard Medical School, Boston, United States; [9]Biosciences Institute, Newcastle University, Newcastle upon Tyne, United Kingdom; [10]Centre for Bacterial Cell Biology, Biosciences Institute, Newcastle University, Newcastle upon Tyne, United Kingdom; [11]Department of Chemistry, Indiana University, Bloomington, United States; [12]Department of Biological Sciences, University of Delaware, Newark, United States; [13]Department of Physics, Princeton University, Princeton, United States

*For correspondence:
nsalama@fhcrc.org

Competing interests: The authors declare that no competing interests exist.

**Abstract** Helical cell shape is necessary for efficient stomach colonization by *Helicobacter pylori*, but the molecular mechanisms for generating helical shape remain unclear. The helical centerline pitch and radius of wild-type *H. pylori* cells dictate surface curvatures of considerably higher positive and negative Gaussian curvatures than those present in straight- or curved-rod *H. pylori*. Quantitative 3D microscopy analysis of short pulses with either *N*-acetylmuramic acid or D-alanine metabolic probes showed that cell wall growth is enhanced at both sidewall curvature extremes. Immunofluorescence revealed MreB is most abundant at negative Gaussian curvature, while the bactofilin CcmA is most abundant at positive Gaussian curvature. Strains expressing CcmA variants with altered polymerization properties lose helical shape and associated positive Gaussian curvatures. We thus propose a model where CcmA and MreB promote PG synthesis at positive and negative Gaussian curvatures, respectively, and that this patterning is one mechanism necessary for maintaining helical shape.

## Introduction

*Helicobacter pylori* is a helical Gram-negative bacterium that colonizes the human stomach and can cause stomach ulcers and gastric cancers (*Correa, 1988*). Helical cell shape is necessary for efficient stomach colonization (*Bonis et al., 2010*; *Sycuro et al., 2012*; *Sycuro et al., 2010*), underscoring its importance. *H. pylori* is a main model organism for studying helical cell shape, in part because it is a genetically tractable organism with a compact genome that minimizes redundancy (*Tomb et al.,*

**eLife digest** Round spheres, straight rods, and twisting corkscrews, bacteria come in many different shapes. The shape of bacteria is dictated by their cell wall, the strong outer barrier of the cell. As bacteria grow and multiply, they must add to their cell wall while keeping the same basic shape. The cells walls are made from long chain-like molecules via processes that are guided by protein scaffolds within the cell. Many common antibiotics, including penicillin, stop bacterial infections by interrupting the growth of cell walls.

*Helicobacter pylori* is a common bacterium that lives in the gut and, after many years, can cause stomach ulcers and stomach cancer. *H. pylori* are shaped in a twisting helix, much like a corkscrew. This shape helps *H. pylori* to take hold and colonize the stomach.

It remains unclear how *H. pylori* creates and maintains its helical shape. The helix is much more curved than other bacteria, and *H. pylori* does not have the same helpful proteins that other curved bacteria do. If *H. pylori* grows asymmetrically, adding more material to the cell wall on its long outer side to create a twisting helix, what controls the process?

To find out, Taylor et al. grew *H. pylori* cells and watched how the cell walls took shape. First, a fluorescent dye was attached to the building blocks of the cell wall or to underlying proteins that were thought to help direct its growth. The cells were then imaged in 3D, and images from hundreds of cells were reconstructed to analyze the growth patterns of the bacteria's cell wall.

A protein called CcmA was found most often on the long side of the twisting *H. pylori*. When the CcmA protein was isolated in a dish, it spontaneously formed sheets and helical bundles, confirming its role as a structural scaffold for the cell wall. When CcmA was absent from the cell of *H. pylori,* Taylor et al. observed that the pattern of cell growth changed substantially.

This work identifies a key component directing the growth of the cell wall of *H. pylori* and therefore, a new target for antibiotics. Its helical shape is essential for *H. pylori* to infect the gut, so blocking the action of the CcmA protein may interrupt cell wall growth and prevent stomach infections.

*1997*). Key non-redundant, non-essential contributors to cell shape have been identified, but the question of how they enable *H. pylori* to be helical remains largely unsolved.

As is the case for most bacteria (*Höltje, 1998*), the structure of the *H. pylori* peptidoglycan (PG) cell wall (sacculus) is ultimately responsible for the shape of the cell; purified cell walls maintain helical shape (*Sycuro et al., 2010*). PG is a polymer of alternating *N*-acetylglucosamine (GlcNAc) and *N*-acetylmuramic acid (MurNAc) with an attached peptide stem that can be crosslinked to a peptide stem of an adjacent PG strand. Crosslinked PG strands form the cell wall, a large mesh-like macromolecule that surrounds the cell and counteracts the cell's turgor pressure (*Höltje, 1998*; *Typas et al., 2012*). The PG monomer is synthesized in the cytoplasm and subsequently flipped across the inner membrane and incorporated into the existing PG by the glycosyltransferase activities of penicillin binding proteins (PBPs) and shape, elongation, division, and sporulation (SEDS) proteins, and the transpeptidation activities of PBPs (*Meeske et al., 2016*; *Sauvage et al., 2008*).

Helical cell shape maintenance in *H. pylori* requires a suite of both PG-modifying enzymes (Csd1, Csd3/HdpA, Csd4, and Csd6) to remodel the cell wall and non-enzymatic proteins (Csd2, Csd5, CcmA, and Csd7) that may act as scaffolds or play other structural roles (*Bonis et al., 2010*; *Sycuro et al., 2013*; *Sycuro et al., 2012*; *Sycuro et al., 2010*; *Yang et al., 2019*). One of the non-enzymatic proteins is the putative bactofilin CcmA. Bactofilins are bacteria-specific cytoskeletal proteins with diverse functions, including playing a role in stalk elongation in *Caulobacter crescentus* (*Kühn et al., 2010*) and helical pitch modulation in *Leptospira biflexa* (*Jackson et al., 2018*). CcmA loss in *H. pylori* results in rod-shaped cells with minimal sidewall curvature (*Sycuro et al., 2010*). As with other organisms, *H. pylori* CcmA has been shown to self-oligomerize (*Holtrup et al., 2019*). Recently CcmA was shown to co-purify with Csd5 and the PG biosynthetic enzyme MurF (*Blair et al., 2018*), suggesting CcmA may influence cell wall growth.

Patterning PG synthesis has been shown to be an important mechanism for cell shape maintenance in several model organisms. In the rod shaped *Escherichia coli*, MreB helps direct synthesis preferentially to sites at or below zero Gaussian curvature. One working model is that this growth

pattern promotes rod shape by accelerating growth at dents and restricting growth at bulges along the sidewall, thereby enforcing diameter control (*Bratton et al., 2018*; *Ursell et al., 2014*). In the Gram-positive *Bacillus subtilis*, MreB filaments have been shown to move in paths oriented approximately perpendicular to the long axis of rod shaped cells. The relative organization of path orientations decreases with an increase in rod diameter, suggesting that filament orientation is sensitive to changes in cell surface curvatures (*Hussain et al., 2018*).

Here, we demonstrate that the surface of helical *H. pylori* cells is characterized by large regions of both positive and negative Gaussian curvature. To investigate how *H. pylori* achieves diameter control while simultaneously maintaining sidewall curvature, we employed two metabolic probes to investigate PG synthesis patterning in *H. pylori*. Using superresolution microscopy and 3D quantitative image analysis, we show that synthesis is enhanced at negative Gaussian curvature as well as at a limited range of positive Gaussian curvatures. We furthermore investigate the localization of cytoskeletal proteins MreB and CcmA. We demonstrate that, as in straight-rod shaped *E. coli* cells, MreB is enriched at negative curvature. CcmA is enriched at the window of positive Gaussian curvatures where enhanced synthesis is observed. We propose that both MreB and CcmA help maintain PG synthesis activity locally and that PG synthesis patterning is one mechanism that plays a fundamental role in helical cell shape maintenance.

## Results

### Helical cells maintain areas of positive and negative Gaussian curvature on the sidewall

Unlike straight-rod shaped bacteria, helical *H. pylori* cells maintain distinct and diverse cell surface curvatures along the sidewall (*Figure 1* and *Figure 2*). To characterize the cell surface curvature features of *H. pylori* in detail, we stained permeabilized cells with fluorescent wheat germ agglutinin (WGA), which binds GlcNAc and thus labels the cell wall. Since the dimensions of *H. pylori* cells (1.5–3.5 μm in length and 0.45 μm in diameter *Figure 3*) are near the limit of light microscopy resolution, we employed 3D structured illumination microscopy (SIM) to more clearly resolve cells in three dimensions (*Figure 1A*). We adapted previous image processing software (*Bartlett et al., 2017*; *Morgenstein et al., 2015*) to accommodate characteristic SIM artifacts and enhanced resolution in order to generate a 3D triangular meshwork surface with roughly 30 nm precision from the SIM z-stack images (*Figure 1A and B*, matched SIM image volumes and surface reconstructions). Display of the Gaussian curvature, which is the product of the two principal curvatures, at each point on the meshwork shows the distinct curvatures on opposite sides of helical cells (*Figure 1B*). Using Gaussian curvature allows us to focus on local curvature geometry. We operationally define the minor helical axis as the shortest helical path along the sidewall within the zone of moderate negative curvature (minor helical axis area, −15 to −5 μm$^{-2}$, blue), and define the major axis as the path opposite the minor helical axis, which resides within the zone of moderate positive curvature (major helical axis area, 5 to 15 μm$^{-2}$, red) (*Figure 1C*). The cell poles are characterized by high positive curvature (>15 μm$^{-2}$, gray).

Our image reconstruction method performs faithful reconstructions of straight- and curved-rod cells (*Figure 2*, inset). To compare the surface curvatures maintained by helical (wild-type), curved-rod (Δcsd2), and straight-rod (Δcsd6) cells, we pooled reconstructions of hundreds of non-septating cells for each genotype and plotted a histogram of the proportion of surface curvature points with a given Gaussian curvature value (*Figure 2*). All three cell shapes share a tail of high positive curvatures from the cell poles (*Figure 2A*, right of the dotted line). In order to study the sidewall alone, we developed an algorithm to computationally define and exclude poles (*Figure 1B*, black lines). With the poles removed, the extended tail disappears for each cell shape. In contrast to the other shapes, helical cells have a large proportion of sidewall area with curvatures less than −5 μm$^{-2}$ and an even larger proportion with curvatures greater than 5 μm$^{-2}$ (*Figure 2B*). Rather than having a unimodal distribution, helical cells have a multimodal distribution that includes an apparent peak at negative curvature and another at positive curvature.

The sidewall curvature distribution informed us about the overall types of surface curvature wild-type cells need to achieve, but was not sufficient to let us directly compare the surface properties of the major and minor axes, specifically the relative lengths of the major and minor axes and the

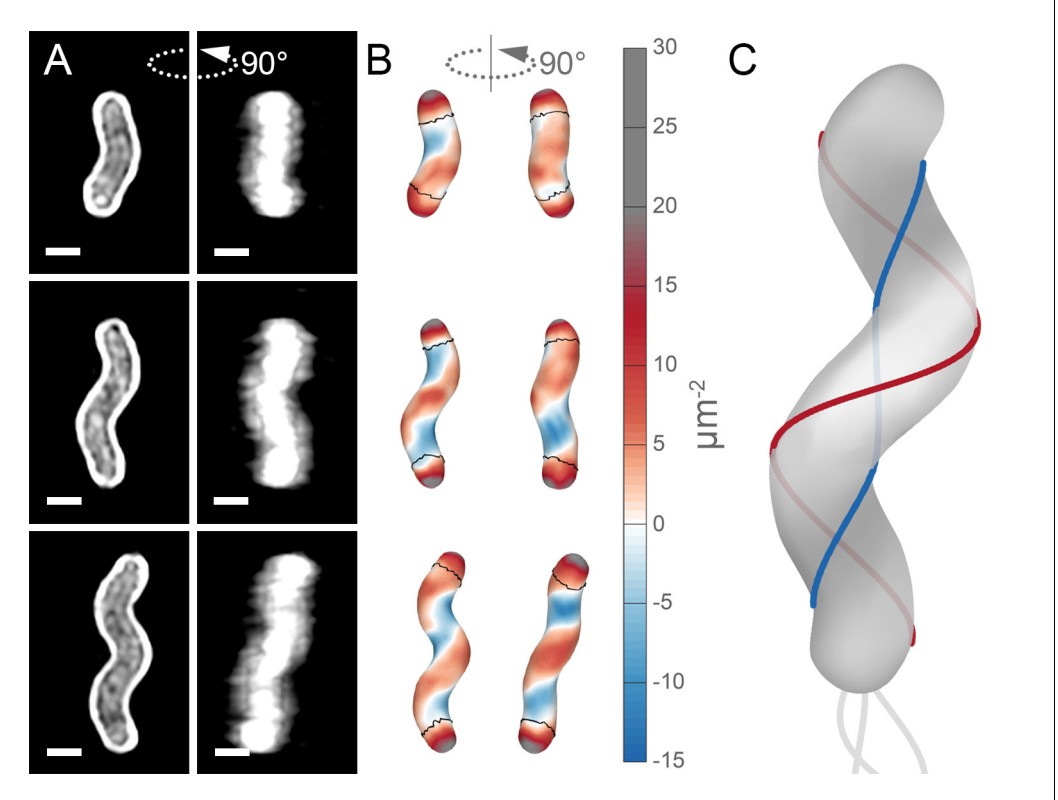

**Figure 1.** Helical cell surfaces feature areas of distinct curvatures. (**A**) 3D SIM images of individual *H. pylori* cells stained with fluorescent wheat germ agglutinin (WGA). Top-down view (left) and 90-degree rotation about the long axis (right). Scale bar = 0.5 µm; images from one experiment. (**B**) Corresponding views of computational surface reconstructions of cells in (**A**). with Gaussian curvature plotted (scale at right - blue: moderate negative; white: zero; red: moderate positive; gray: high positive). Computationally-defined polar regions are delineated by the thin black line. Polar regions correspond to regions whose centerline points are within 0.75 of a cell diameter to the terminal pole positions. (**C**) Schematic of minor (blue line) and major (red line) helical axes.

average Gaussian curvature along both axes. Furthermore, prior shape parameter characterizations of *H. pylori* have been performed using 2D images (*Martínez et al., 2016*; *Sycuro et al., 2013*; *Sycuro et al., 2012*; *Sycuro et al., 2010*; *Yang et al., 2019*); measurement of pitch and helical radius from 2D images is subject to systematic errors for short cells (approximately <1.5 helical turns) depending on their orientation on the coverslip. Therefore, we also wished to determine *H. pylori* population shape parameters from our 3D dataset. To characterize the major and minor axes, we needed to find these axes on each reconstructed cell surface. While cells in our experiments appear helical, in reality they have surface imperfections and centerlines with kinks, bends, or variation in pitch along the centerline (*Sycuro et al., 2010*). We therefore limited ourselves to considering the relative length of the major and minor helical axes of a population of simulated, idealized cells, each of which mimics a cell from the wild-type population described in *Figure 2* (for full details see Appendix 1). In brief, to both derive the cell shape parameters necessary to generate the simulated cells and to further characterize the 3D shape parameters of the wild-type population, we measured the cell lengths from one pole to the other along the curved centerlines (*Figure 3A and C*, gray); the diameters of the cells (*Figure 3A and D*, purple); the helical pitches of the centerlines (*Figure 3A and E*, pink); and the helical diameters of the centerlines (*Figure 3A and F*, green).

Wild-type cells are 2.5 ± 0.5 µm long and 0.45 ± 0.02 µm in diameter, have a helical pitch of 1.7 ± 1 µm, and have a helical diameter of 0.3 ± 0.1 µm (mean ± standard deviation, *Figure 3C–F*). These parameters are derived from a subset of the wild-type population that can be modeled as a uniform helix (*Figure 3—figure supplement 1* and *Figure 3—video 1*). The distribution of cell lengths, diameters, and surface curvatures of the subset closely match that of the whole population

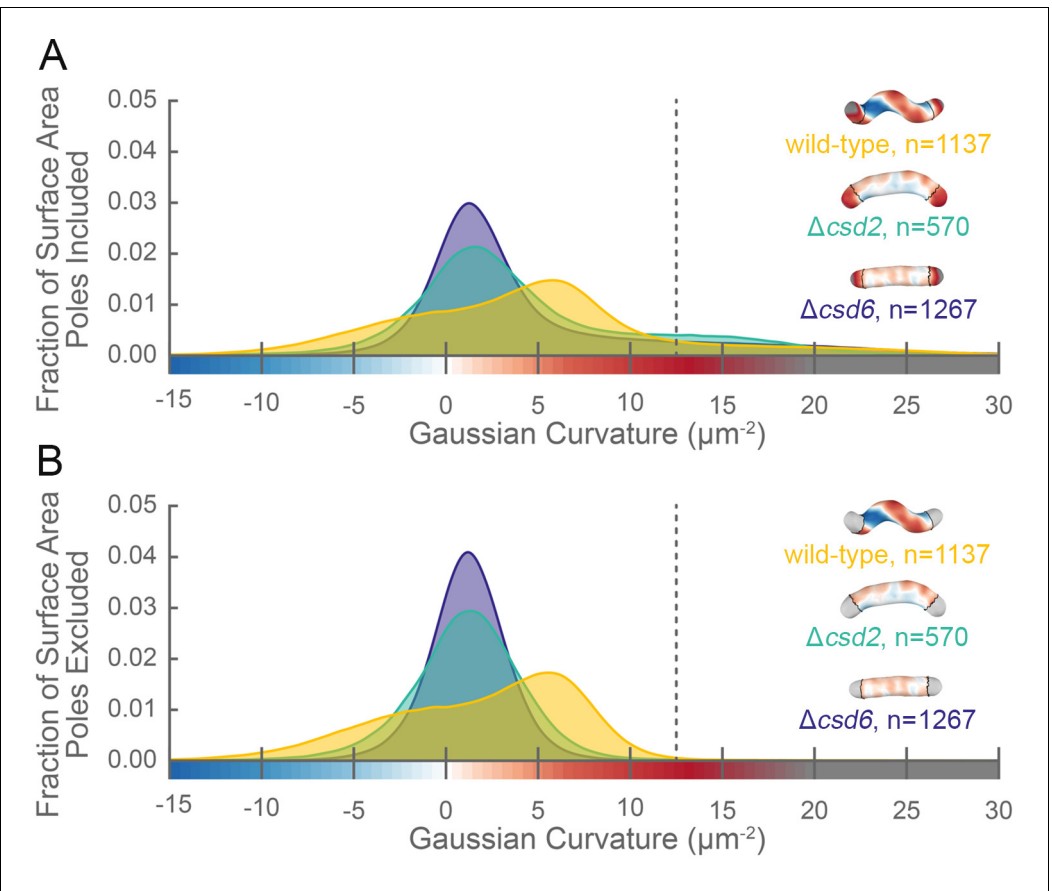

**Figure 2.** The distribution of surface Gaussian curvature for helical cells is distinct from that of curved- and straight-rod cells. Smooth histograms of the distribution of surface Gaussian curvatures for a population of cells (wild-type helical, yellow; curved-rod Δcsd2, teal; straight-rod Δcsd6, indigo) with poles included (**A**) or sidewall only (**B**, poles excluded). The region to the right of the dotted vertical lines corresponds to curvatures contributed almost exclusively by the poles. Histograms are derived using a bin size of 0.2 $\mu m^{-2}$. Example computational surface reconstructions (top right of each histogram) of a wild-type helical, curved-rod Δcsd2, and straight-rod Δcsd6 cell with Gaussian curvatures displayed as in *Figure 1*. The data represented are from one replicate.

(*Figure 3—figure supplement 1C–E*). Using the simulated counterparts to these cells, we determined that the average major to minor length ratio is 1.69 ± 0.16, meaning that the major axis is on average 70% longer than the minor axis (*Figure 3G*). We also determined from the simulated cells that the average Gaussian curvature at the major axis is 5 ± 1 $\mu m^{-2}$, and the average Gaussian curvature at the minor axis is −11 ± 4 $\mu m^{-2}$ (*Figure 3H*).

We next used our simulation framework to explore how the four helical-rod shape parameters affect the length ratio of the major to minor helical axes. Changes in cell length and cell diameter had almost no effect, whereas increasing the helical diameter or decreasing the helical pitch increased the relative length of the major axis (*Figure 3—figure supplement 2*, right column), consistent with the idea that a helix is formed by differential expansion of the major and minor axes. We then investigated how each of these parameters influences the distribution of surface curvatures along the sidewall. We began with a cell simulated from the population average of all four parameters (cell length, cell diameter, helical pitch, and helical diameter), and changed each property individually within the range of variation represented in the wild-type population (±1.5 standard deviations) while holding the other three constant (*Figure 3—figure supplements 2 and 3*). Each of the dashed colored lines in *Figure 3C–F* correspond to the parameters used to simulate these altered cell shapes. Changing cell length had a negligible impact on the distribution of surface curvatures along the sidewall (*Figure 3—figure supplement 2A*). Decreasing the cell diameter had a relatively small effect given the narrow distribution of cell diameters observed in the wild-type

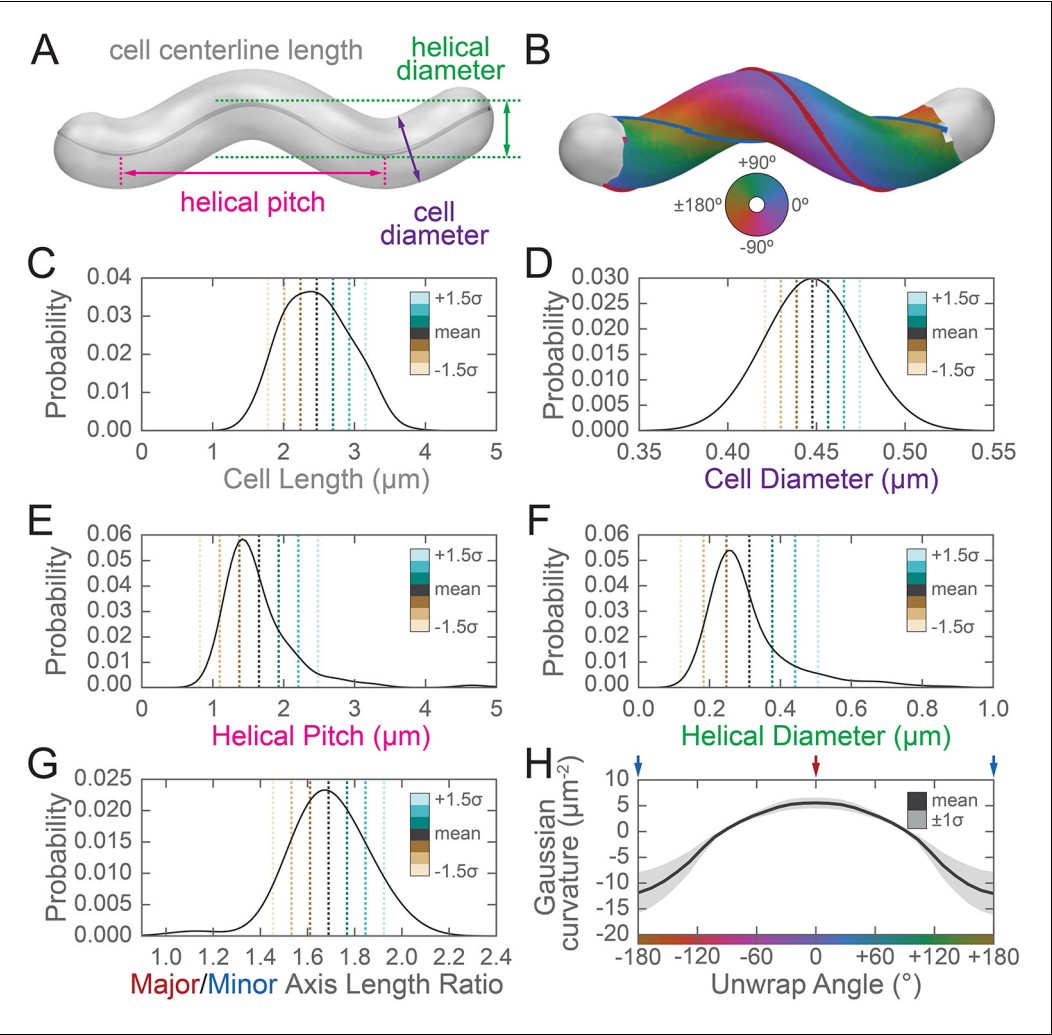

**Figure 3.** Three-dimensional shape properties of a wild-type helical population. Analysis of the wild-type population in *Figure 2* from the 231 wild-type cells for which the cell centerline was well-fit by a helix. (**A**) Schematic of helical-rod shape parameters (cell centerline length, gray; cell diameter, purple; helical pitch, pink; and helical diameter, green). (**B**) Example cell with helical coordinate system and the major (red line, 0°) and minor (blue line, 180°) helical axes shown on the cell sidewall. Population distributions of (**C**) cell centerline lengths, (**D**) average cell diameters, (**E**) helical pitch, (**F**) helical diameter, (**G**) major to minor axis length ratio, and (**H**) the average Gaussian curvature for a given helical coordinate system unwrap angle. Colored dotted lines in (**C–G**) indicate the mean ±1.5 standard deviations in 0.5 standard deviation steps. Shaded line in (**H**) indicates ±1 standard deviation about the mean. Distributions of parameters (**C–D**) are from real cells, parameters (**E–F**) are from helical centerline fits, and properties (**G–H**) are measured from the matched synthetic cell sidewalls.

The online version of this article includes the following video and figure supplement(s) for figure 3:

**Figure supplement 1.** Evaluation of the subset of the wild-type population used to generate synthetic cells.
**Figure supplement 2.** Change in the distribution of cell surface Gaussian curvatures based on modulating helical rod parameters.
**Figure supplement 3.** Simulated helical cells demonstrating how variation in helical parameters alters surface Gaussian curvature.
**Figure 3—video 1.** Rotation of example cell centerlines (gray dots) and calculated helical fits (red lines), arranged from good (left) to poor (right) fit from *Figure 3—figure supplement 1A*.
https://elifesciences.org/articles/52482#fig3video1

population (*Figure 3—figure supplement 2B*). Changing the two parameters describing the properties of the helix had a larger impact on the distribution of Gaussian curvatures. Decreasing the pitch resulted in a helix with tighter coils and a greater distance between the peak of negative and positive Gaussian surface curvatures (*Figure 3—figure supplement 2C*). Increasing the helical diameter resulted in cells that looked less like straight-rod cells and had a greater distance between the peak of negative and positive Gaussian surface curvatures (*Figure 3—figure supplement 2D*). In holding with the Gauss-Bonnet theorem, cells had a greater proportion of sidewall area with positive Gaussian curvature than with negative, and the magnitude of the positive Gaussian curvature was less than that of the negative Gaussian curvature.

Having established the substantial difference in the length of the major and minor axes, we wondered if differential synthesis at these cellular landmarks might help explain helical shape maintenance. Although it is not currently possible to computationally define the helical axes on surface reconstructions of actual cells due to their imperfections, our data indicate that we can use Gaussian curvatures of 5 $\mu m^{-2}$ and $-11$ $\mu m^{-2}$ as a proxy for the major and minor axes, respectively, in population level data.

## *H. pylori* can incorporate modified D-alanine and modified MurNAc into peptidoglycan

Since a helical cell must maintain large regions of positive and of negative curvatures, we hypothesized that *H. pylori* may have a different growth pattern than that of *E. coli*, where the majority of the sidewall regions have Gaussian curvature near zero. To determine where new PG is preferentially inserted, we used two metabolic probes of PG incorporation. First, we attempted labeling wild-type cells with MurNAc-alkyne (MurNAc-alk), but *H. pylori* is unable to readily use exogenous MurNAc. We then engineered a strain, HJH1, containing recycling enzymes AmgK and MurU from *Pseudomonas putida* (*Gisin et al., 2013*) at the *rdxA* locus, a neutral locus routinely used for expression of genes in *H. pylori* (*Goodwin et al., 1998*; *Smeets et al., 2000*). These enzymes convert MurNAc into UDP-MurNAc, which can then be used to form PG subunit precursors (*Figure 4—figure supplement 1*). To verify that HJH1 can indeed use exogenous MurNAc, we assayed rescue from fosfomycin treatment. Fosfomycin blocks the first committed step in PG precursor synthesis by preventing the conversion of UDP-GlcNAc into UDP-MurNAc (*Figure 4—figure supplement 1*). We determined the minimum inhibitory concentration (MIC) of fosfomycin of our strain to be 25 $\mu$g/ml (*Figure 4—figure supplement 2*). Supplementation with 4 mg/ml MurNAc partially rescued growth of HJH1 in the presence of 50 $\mu$g/ml fosfomycin, but not the parental strain (LSH108) (*Figure 4A*).

To verify that clickable MurNAc-alk is indeed incorporated into the cell wall, we purified sacculi from HJH1 labeled with MurNAc-alk for six doublings for MS/MS analysis. We positively identified MurNAc-alk-pentapeptide and MurNAc-alk-tetra-pentapeptide, the most abundant monomeric and dimeric species in the *H. pylori* cell wall, (*Figure 4B,C* and *Figure 4—figure supplement 3*), as well as less-abundant species (*Table 1*), confirming incorporation. Cells were labeled without the addition of fosfomycin, indicating the HJH1 strain can use MurNAc-alk even when unmodified MurNAc is available in the cell.

As a second strategy for labeling new PG incorporation, we used D-alanine-alkyne (D-Ala-alk) (*Kuru et al., 2012*; *Siegrist et al., 2013*). This probe can be incorporated through the activity of PG transpeptidases (*Figure 4—figure supplement 1*). To verify that D-Ala-alk is incorporated into the cell wall and to determine the position(s) at which it is incorporated, we purified sacculi from wild-type (LSH100) cells labeled for six doublings for analysis. D-Ala-alk was detected in only pentapeptide monomers and tetra-pentapeptide dimers, indicating that D-Ala-alk is exclusively incorporated at the pentapeptide position (*Figure 4D* and *Figure 4—figure supplement 4*).

## PG synthesis is enriched at both negative Gaussian curvature and the major helical axis area

To visualize new PG incorporation, we labeled HJH1 with either MurNAc-alk or D-Ala-alk for 18 min (approximately 12% of the doubling time). AF555-azide was conjugated to the alkyne groups using click chemistry and cells were counterstained with WGA-AF488. Cells were imaged using 3D SIM (*Figure 5* and *Figure 5—video 1*). As expected, labeling was seen on the boundary of the cell but not in the cytoplasmic area (*Figure 5D and H*). For both metabolic probes, PG synthesis appeared

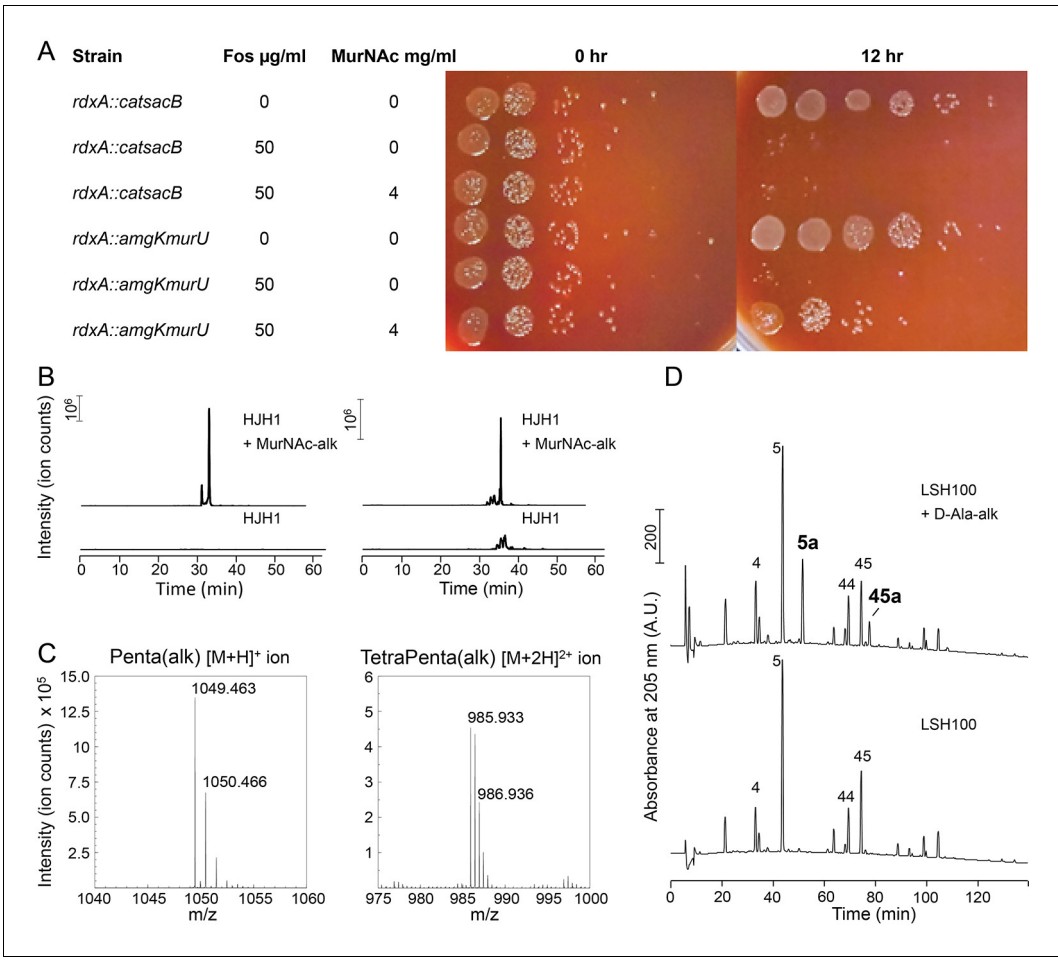

**Figure 4.** Validation of PG metabolic probes. (**A**) 10-fold dilutions showing LSH108 (*rdxA::catsacB*) or HJH1 (*rdxA::amgKmurU*) treated with 50 μg/ml fosfomycin or untreated and with or without 4 mg/ml MurNAc supplementation, from one representative of three biological replicates. (**B** and **C**) Verification of MurNAc-alk incorporation into pentapeptides (left column) and tetra-pentapeptides (right column) by HPLC/MS/MS. (**B**) Extracted ion chromatograms (EICs) for the ion masses over the HPLC elution for unlabeled (lower EIC) and labeled (top EIC) sacculi. (**C**) Spectra of the ions observed during LC-MS for the MurNAc-alk pentapeptide (left, non-reduced, predicted [M+H]+ ion *m/z* = 1049.452) and MurNAc-alk tetra-pentapeptide dimer (right, non-reduced, predicted [M+2H]2+ ion *m/z* = 985.920). (**D**) Verification of D-Ala-alk incorporation into pentapeptides and tetra-pentapeptides. HPLC chromatograms of labeled (top) and unlabeled (bottom) sacculi. The main monomeric and dimeric muropeptides are labeled (4, disaccharide tetrapeptide; 5, disaccharide pentapeptide; 44, bis-disacccharide tetratetrapeptide; 45, bis-disacharide tetrapentapeptide). D-Ala-alk-modified muropeptides (top, 5a and 45a) are present only in the sample from labeled cells and were confirmed by MS analysis of the collected peak fractions. 5a, alk-labeled disaccharide pentapeptide (neutral mass: 1036.448); 45a, alk-labelled bis-disaccharide tetrapentapeptide (neutral mass: 1959.852). Data (**B**, **C**, and **D**) are from one replicate.

The online version of this article includes the following figure supplement(s) for figure 4:

**Figure supplement 1.** Schematic of PG synthesis and incorporation of PG metabolic probes.

**Figure supplement 2.** The MIC of fosfomycin in *H. pylori* is 25 μg/ml.

**Figure supplement 3.** Detected MurNAc-alk labeled muropeptides.

**Figure supplement 4.** Detected D-Ala-alk labeled muropeptides.

to be excluded from the poles, dispersed along the sidewall, and present at septa. However, D-Ala-alk septal labeling appeared much brighter compared to MurNAc-alk septal labeling, indicating at least some difference between incorporation and/or turnover of the two probes. To discover if this labeling difference is due to curvature-biased transpeptidation rates, we also attempted labeling with dimers D-alanine-D-alanine-alkyne and D-alanine-alkyne-D-alanine, which is presumably

**Table 1.** MurNAc-alk incorporation into PG

| Muropeptide (non-reduced) | Theoretical neutral mass | MurNAc-alk labeled *H. pylori* | | | | Control *H. pylori* | | |
| | | Observed ion (charge) | Rt[*] (min) | Calculated neutral mass | | Observed ion (charge) | Rt[*] (min) | Calculated neutral mass |
|---|---|---|---|---|---|---|---|---|
| Di | 696.270 | 697.289 (1+) | 20.3 | 696.282 | | 697.290 (1+) | 20.4 | 696.283 |
| *Alk*-Di | 734.286 | 735.307 (1+) | 30.5 | 734.300 | | -[†] | - | - |
| Tri | 868.355 | 869.375 (1+) | 15.8 | 868.368 | | 869.374 (1+) | 15.8 | 868.367 |
| *Alk*-Tri | 906.371 | 907.392 (1+) | 25.8 | 906.385 | | - | - | - |
| Tetra | 939.392 | 940.411 (1+) | 20.4 | 939.404 | | 940.412 (1+) | 20.4 | 939.405 |
| *Alk*-Tetra | 977.408 | 978.428 (1+) | 30.4 | 977.421 | | - | - | - |
| Penta | 1010.429 | 1011.449 (1+) | 22.9 | 1010.442 | | 1011.449 (1+) | 22.8 | 1010.442 |
| *Alk*-Penta | 1048.445 | 1049.464 (1+) | 32.9 | 1048.457 | | - | - | - |
| TetraTri | 1789.736 | 895.889 (2+) | 33.4 | 1789.762 | | 895.888 (2+) | 33.3 | 1789.761 |
| *Alk*-TetraTri | 1827.752 | 914.898 (2+) | 39.2 | 1827.781 | | - | - | - |
| TetraTetra | 1860.774 | 931.407 (2+) | 35.0 | 1860.799 | | 931.407 (2+) | 34.9 | 1860.799 |
| *Alk*-TetraTetra | 1898.789 | 950.416 (2+) | 39.7 | 1898.817 | | - | - | - |
| TetraPenta | 1931.811 | 966.926 (2+) | 35.8 | 1931.837 | | 966.925 (2+) | 35.7 | 1931.835 |
| *Alk*-TetraPenta | 1969.826 | 985.934 (2+) | 39.9 | 1969.853 | | - | - | - |

[*] Rt, retention time.

[†] -, not detected. Muropeptides detected (confirming incorporation) via LC-MS analysis of MurNAc-alk labeled versus control PG digests. The control cells displayed no evidence of any MurNAc-alk incorporation.

incorporated predominantly through PG precursor biosynthesis in the cytoplasm, but no signal was detected (data not shown) (*Liechti et al., 2014*).

To quantify any curvature-based enrichment (expressed throughout as relative concentration vs. Gaussian curvature) of new cell wall synthesis, we used the fluorescent WGA signal to generate 3D cell surface reconstructions of hundreds of individual, non-septating cells labeled with MurNAc-alk, D-Ala-alk, or cells that were mock-labeled as a control. The Gaussian curvature was calculated at every location on the reconstructed 3D surface of the cell. Because the absolute amount of synthesis (or other signals of interest) can vary between cells, and because the level of illumination throughout the field of view is non-uniform, we set the average PG synthesis signal for each individual cell to one. We measured each cell's curvature-dependent PG synthesis signal intensity relative to that average value, normalized by the amount of that curvature present on the surface, since there is more surface area associated with positive Gaussian curvature than negative (*Figure 6A*).

As a tool to facilitate understanding and interpretation of these relative enrichment plots, we generated a synthetic cell surface with the same geometric properties as the average wild-type cell (*Figure 3*), applied a variety of example intensity distributions, and generated curvature enrichment plots. We began with a uniform baseline signal (*Figure 6—figure supplement 1*, 'uniform - low') and in each case added 25% extra signal intensity to specific geometries. In the enrichment profiles, a relative concentration value of one indicates that the average signal intensity at that curvature is the same as the average across the cell surface. Values greater than one indicate curvatures where normalized signal is enriched compared to average and values less than one indicate curvatures where normalized signal is depleted compared to average. These simulations illustrate the interrelated nature of the relative enrichment plots. Because there is more cell surface area with positive Gaussian curvature, adding 25% signal to this region (*Figure 6—figure supplement 1*, 'enriched at major axis') increases the average signal more than adding 25% signal at zero or negative Gaussian curvature. Thus by increasing the signal at positive curvature, the relative concentration decreases at the rest of the cell surface even though the absolute signal at these geometries remains the same. A similar change in relative concentration occurs with an increase in signal at zero or negative curvature (*Figure 6—figure supplement 1*, 'enriched at zero' and 'enriched at minor axis', respectively), but because there is less surface area with these curvatures, the magnitude of this change is lower. To further illustrate the implications of the interrelated nature of these plots, we added both signal

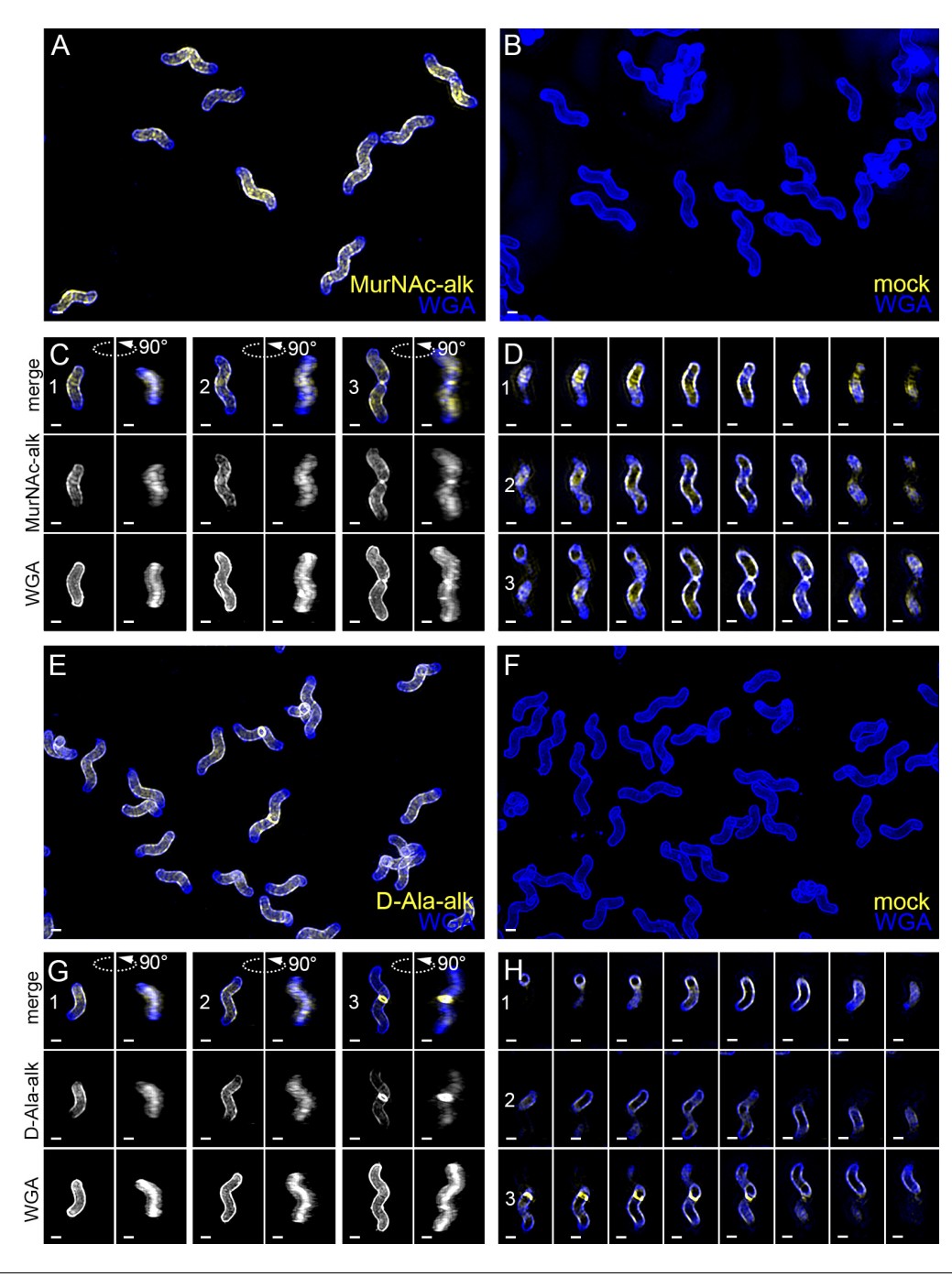

**Figure 5.** New cell wall growth appears dispersed along the sidewall, excluded from poles, and present at septa. 3D SIM imaging of wild-type cells labeled with an 18 min pulse of MurNAc-alk (**A–D**, yellow) or 18 min pulse of D-Ala-alk (**E–H**, yellow) counterstained with fluorescent WGA (blue). Color merged maximum projection of 18 min MurNAc-alk (**A**), D-Ala-alk (**E**), or mock (**B, F**) labeling with fluorescent WGA counterstain. (**C, G**) Top-down (left) and 90-degree rotation (right) 3D views of three individual cells, including a dividing cell at the right. Top: color merge; middle: 18 min MurNAc-alk (**C**) or D-Ala-alk (**G**); bottom: fluorescent WGA. (**D, H**) Color merged z-stack views of the three cells in (**C, G**), respectively (left to right = top to bottom of the cell). Numbering indicates matching cells. Scale bar = 0.5 µm. The representative images are selected from one of three biological replicates.

The online version of this article includes the following video for figure 5:

**Figure 5—video 1.** Volumetric rendering and z-slices of the example cells in *Figure 5*.
https://elifesciences.org/articles/52482#fig5video1

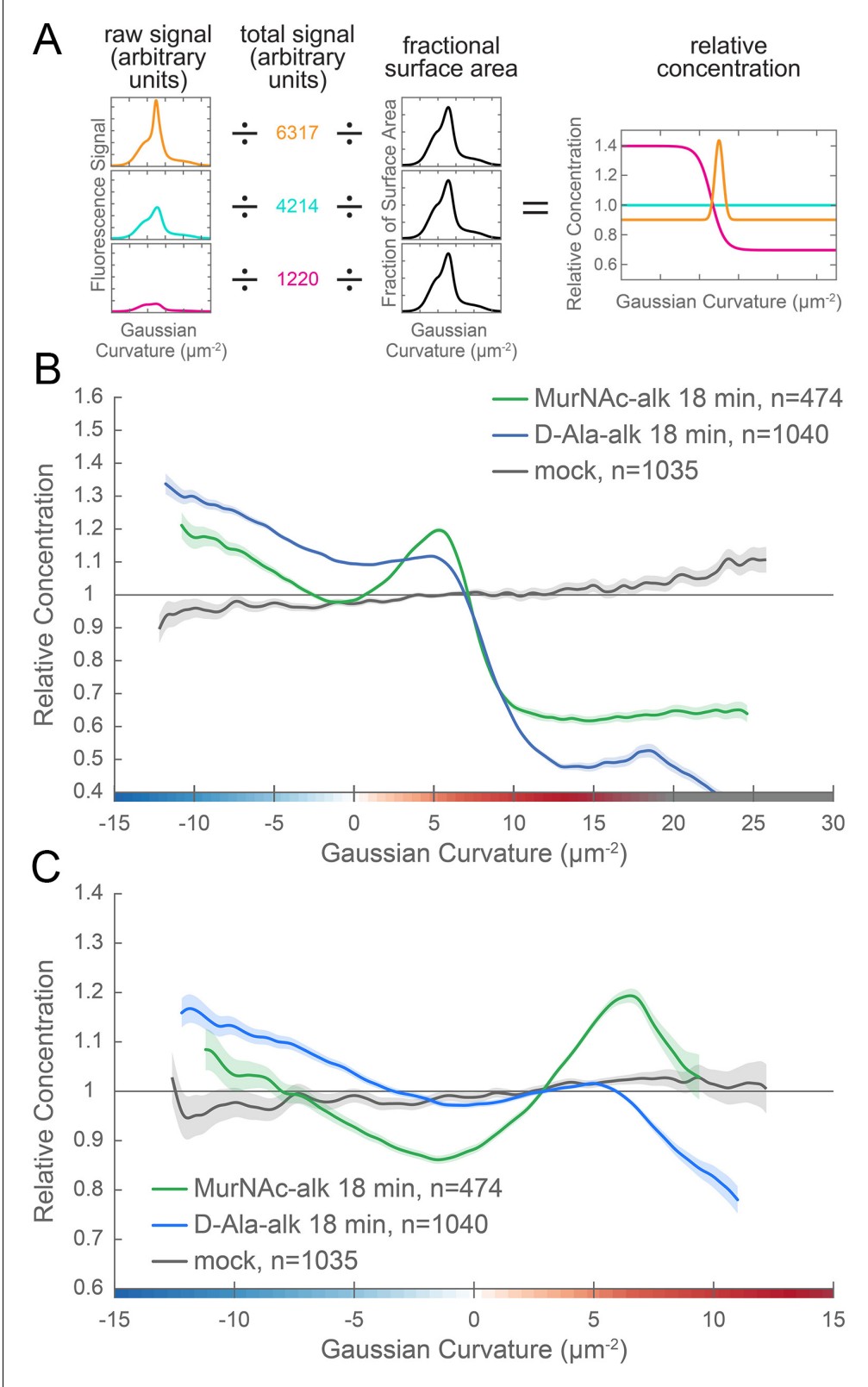

**Figure 6.** New cell wall growth is excluded from the poles and enriched at negative Gaussian curvature and the major axis area. (**A**) The calculation of relative concentration for a specific probe involves two steps of normalization. First, the raw signal is summed up in bins defined by the Gaussian curvature at the surface. Then, this raw signal is normalized by dividing by the sum of the raw signal at all Gaussian curvatures (total signal). This

*Figure 6 continued on next page*

*Figure 6 continued*

normalizes for changes in total signal, fluorophore brightness, imaging conditions, etc. The second step is to divide by the fractional surface area, or amount of surface area contributed by each Gaussian curvature bin. This distribution is dependent on the observed shape of the cell. Following these two normalization steps, one has the concentration of the probe of interest relative to a uniformly distributed null model. For illustration, we have shown this graphical equation for three noise-free cells that have the same geometry, but different relative signal abundances. In the experimental data presented in the main text, the single cell relative concentration profile is averaged over hundreds of cells, each with their own unique geometry. Whole surface (B) and sidewall only (C) surface Gaussian curvature enrichment of relative concentration of new cell wall growth (y-axis) vs. Gaussian curvature (x-axis) derived from a population of computational cell surface reconstructions of MurNAc-alk (green), D-Ala-alk (blue) 18 min pulse-labeled, and mock-labeled (gray) cells. 90% bootstrap confidence intervals are displayed as a shaded region about each line. The represented data are pooled from three biological replicates. The online version of this article includes the following figure supplement(s) for figure 6:

**Figure supplement 1.** For eight different example distributions (rows with brief labels to the left), five pieces of data are shown.

**Figure supplement 2.** Curvature enrichment analysis of biological replicates of MurNAc-alk-, D-Ala-alk-, and mock-labeling.

---

with a monotonic decline profile (*Figure 6—figure supplement 1*, 'monotonic decline') and signal enriched at the major axis (*Figure 6—figure supplement 1*, 'enriched at major axis') to one cell surface (*Figure 6—figure supplement 1*, 'monotonic decline and major axis'). By adding extra signal at the major axis area, the average concentration increases significantly, causing the rest of the relative concentrations to decrease compared to the monotonic decline profile alone. As these simulations demonstrate, relative enrichment plots must be considered holistically. The key features of interest are the overall increases, decreases, and peaks in the curves, along with the curvatures at which these occur.

We performed relative concentration enrichment analysis separately with the entire cell surface and with the sidewall only (poles removed) from the PG synthesis data. We then averaged the single cell measurements across more than 100 cells pooled from three biological replicates to obtain a profile of enrichment or depletion as a function of surface curvature. Curvature enrichment analysis of whole cell surfaces revealed that for both metabolic probes, signal was largely absent from the poles, as seen by the drop-off of relative enrichment at curvatures above 10 $\mu m^{-2}$ (*Figure 6B*). To focus on the curvature enrichment pattern along the sidewall, we repeated the analysis after first computationally removing the poles. Looking at sidewall curvature alone, MurNAc-alk was enriched at two sites. At negative curvature, enrichment increases as curvature becomes more negative. At positive curvature, enrichment peaks near 6 $\mu m^{-2}$ and then begins to decrease at higher curvatures (*Figure 6C*, green). D-Ala-alk showed peaks of enrichment aligning with those of MurNAc-alk (*Figure 6C*, blue), but the magnitude of the peak at positive curvature was reduced. The mock labeling control showed minimal curvature bias and is on average 3.6% of the D-Ala-alk signal and 4.5% of the MurNAc-alk signal (*Figure 6B and C*, gray and *Figure 6—figure supplement 2B*). This demonstrates that the fluorescent signal in the mock labeling is independent of geometry. Thus the non-specific signal should contribute negligibly to the PG synthesis enrichment profiles. Biological replicates are shown in *Figure 6—figure supplement 2A*.

## MreB is enriched at negative Gaussian curvature

The cytoskeletal protein MreB has been shown in rod-shaped organisms to preferentially localize to negative Gaussian curvatures near to and below zero and help direct PG synthesis (*Bratton et al., 2018*; *Ursell et al., 2014*). It has been reported that MreB is not essential in *H. pylori* and that treatment with the MreB inhibitor A22 does not alter cell shape (*Waidner et al., 2009*), though growth inhibition only occurred at concentrations well above those used to select for A22 resistance in other organisms (*Gitai et al., 2005*; *Ouzounov et al., 2016*; *Srivastava et al., 2007*; *Wu et al., 2011*). Since multiple attempts to knock out *mreB* in wild-type LSH100 were unsuccessful, we generated IM4, a merodiploid strain with a second copy of *mreB* at a neutral intergenic locus (McGee locus *Langford et al., 2006*) (*Figure 7—figure supplement 1A*) for comparative transformation experiments. To verify that both LSH100 and IM4 are readily transformable, we performed parallel

transformations with a *ccmA::CAT* deletion cassette. LSH100 and IM4 showed similar transformation efficiencies ($2.4 \times 10^{-4}$ and $1.2 \times 10^{-4}$, respectively) (*Figure 7A*). We transformed LSH100 and IM4 with an *mreB::CAT* deletion cassette (*Figure 7A* and *Figure 7—figure supplement 1A*) and obtained *mreB* targeting transformants in strain IM4 at a frequency of $2.3 \times 10^{-4}$. The CAT resistance cassette integrated into *mreB* at either the native locus or the McGee locus (19 and 5 of 24 clones tested, respectively) (*Figure 7—figure supplement 1B*). In contrast, we obtained two colonies after transformation of LSH100 (frequency of $6.7 \times 10^{-7}$). Sequencing revealed that an amplification event at the *mreB* locus occurred for each of these clones, such that an uninterrupted copy of *mreB* was present in addition to a copy of *mreB::CAT* (*Figure 7—figure supplement 1D*). Western blotting revealed that MreB was produced at wild-type levels in clone #2, but only a faint band was observed for clone #1 (*Figure 7—figure supplement 1C*). In clone #1, the terminal four amino acids were replaced due to the recombination event (GFSE to FLAN). One of the four epitopes used to generate the anti-MreB antibody includes the four terminal amino acids (*Nakano et al., 2012*), likely explaining the discrepancy between the sequencing results and western blot detection. While we requested the previously published *mreB* mutant strains (*Waidner et al., 2009*), they could not be revived from frozen stocks. We thus conclude that MreB is essential in LSH100 and perhaps all *H. pylori* strains.

We investigated MreB localization to determine if an altered curvature preference might account for the PG synthesis pattern we observed. Immunofluorescence labeling with 3D SIM imaging revealed that MreB is present at the cell periphery as many individual foci and some short arcs that appear to be oriented approximately circumferentially and excluded from the poles (*Figure 7B,D and E* and *Figure 7—video 1*). Only sparse foci were seen with immunofluorescence using the pre-immune serum (*Figure 7C*). Curvature enrichment analysis of non-dividing cells confirmed that MreB localization is depleted at the poles (*Figure 7—figure supplement 2*). Regardless of whether the poles were included in the analysis, we observed that as Gaussian curvature became more negative, relative MreB concentration increased monotonically (*Figure 7F* and *Figure 7—figure supplement 2*). Biological replicates are shown in *Figure 7—figure supplement 3A*. This echoes the enrichment of PG synthesis at negative Gaussian curvature; as Gaussian curvature became more negative (below $-2 \ \mu m^{-2}$), relative PG synthesis increased monotonically. Preimmune serum signal was 36.4% of the MreB signal (*Figure 7—figure supplement 3B*), but did not show a curvature preference (*Figure 7E*, gray). Thus, MreB may promote the enhanced PG synthesis observed at negative curvature.

## The bactofilin CcmA forms filaments, bundles, and lattices in vitro

We reasoned that another cytoskeletal element might promote the higher relative PG synthesis observed at the major axis area. While both coiled-coil rich proteins (Ccrp) and the bactofilin homolog CcmA have been implicated in *H. pylori* cell shape (*Specht et al., 2011*; *Sycuro et al., 2010*; *Waidner et al., 2009*), only loss of CcmA, and not individual Ccrps, results in a drastic cell shape defect in our strain background (*Yang et al., 2019*); $\Delta ccmA$ cells are nearly straight. To verify CcmA's status as a cytoskeletal filament, we tested its ability to form higher-order structures in vitro. Negative staining of recombinant wild-type CcmA purified from *E. coli* revealed filaments of varying length, long helical bundles of filaments, and lattice structures (*Figure 8A–B* and *Figure 8—figure supplement 1A*). Fourier transform analysis of the lattice structures revealed a filament spacing of 5.5 nm (*Figure 8—figure supplement 2*), similar to that previously observed for *C. crescentus* BacA lattices (5.6 nm) (*Vasa et al., 2015*). While BacA forms orthogonal lattices, the CcmA lattices are skewed (acute angle = 71.5°; obtuse angle = 106.2°).

To begin to assess the importance of higher-order structures and localization for CcmA cell shape functions, we constructed two point mutant variant proteins, located in the predicted hydrophobic core of the protein (I55A and L110S) (*Shi et al., 2015*). Homologous residues (75 and 130, respectively) were shown to be important for polar localization of the bactofilin BacA in *C. crescentus* (*Vasa et al., 2015*). While both proteins could be expressed and purified from *E. coli* (*Figure 8—figure supplement 1D*), the recombinant proteins either fail to form any higher order structures under any buffer condition tested (I55A; *Figure 8C*) or form no lattice structures and many individual filaments in addition to bundles that are straighter, narrower, and shorter than those of wild-type CcmA in vitro (L110S; *Figure 8D* and *Figure 8—figure supplement 1B*). When expressed as the sole copy of *ccmA* in *H. pylori*, both mutant proteins could be detected in whole cell extracts

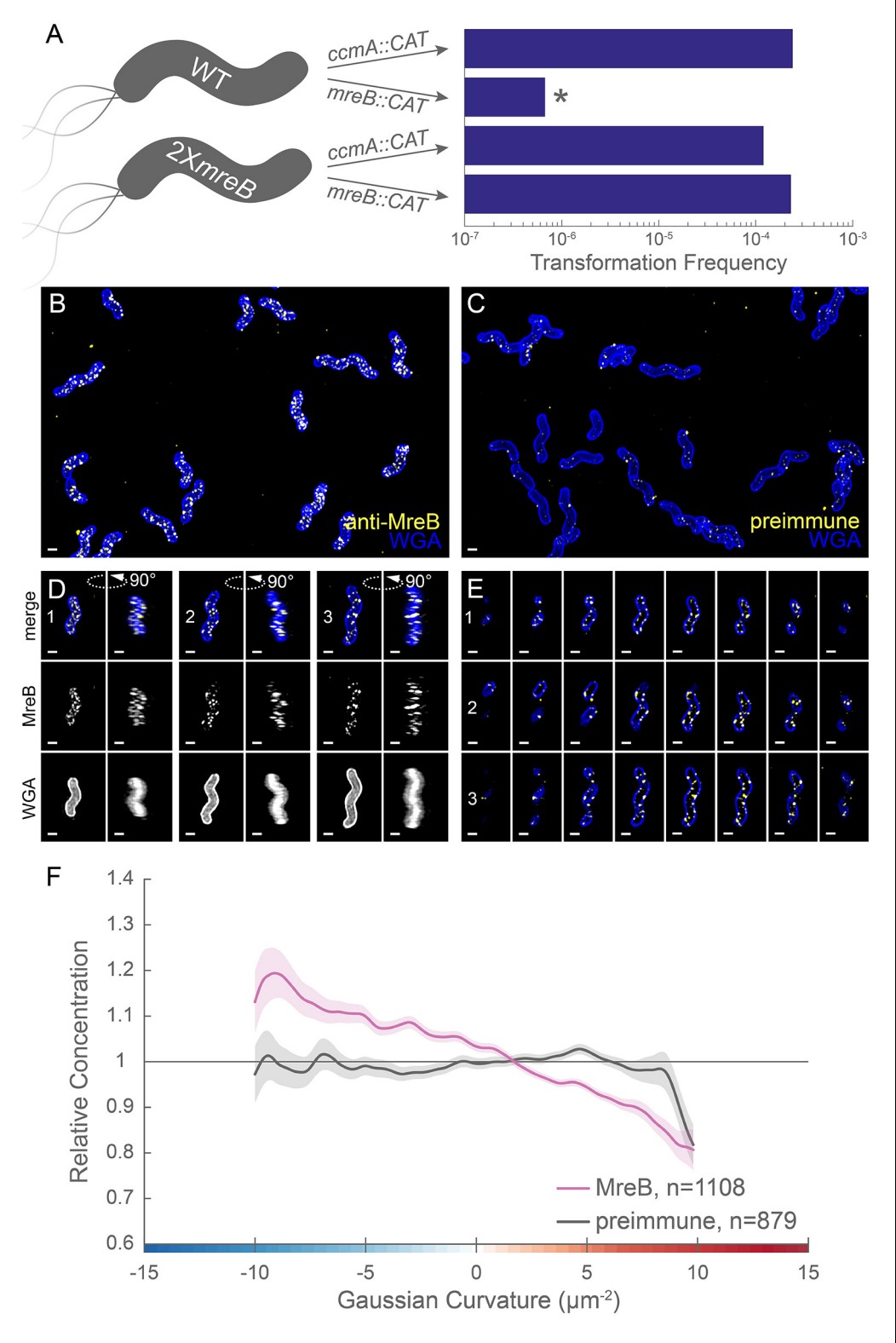

**Figure 7.** MreB is essential in LSH100 and is present as small foci enriched at negative Gaussian curvature. (**A**) Schematic of transformation experiment testing MreB essentiality in LSH100 (WT) and IM4 (2X*mreB*) (left) and corresponding transformation frequencies (right). *=two recombinant clones with *mreB* duplication (see *Figure 7—figure supplement 1* for details). 3D SIM imaging of wild-type cells immunostained with anti-MreB (**B**, **D**, **E**, yellow) or preimmune serum (**C**, yellow) and counterstained with fluorescent WGA (blue). (**B**, **C**) Color

*Figure 7 continued*

merged maximum projections (D) Top-down (left) and 90-degree rotation (right) 3D views of three individual cells. Top: color merge; middle: anti-MreB; bottom: fluorescent WGA. (E) Color merged z-stack views of the three cells in (A). (left to right = top to bottom of the cell). Numbering indicates matching cells. Scale bar = 0.5 μm. (F) Sidewall only surface Gaussian curvature enrichment plots for a population of cells immunostained with anti-MreB (pink), or preimmune serum (gray). Smooth line plot (solid line) of relative MreB concentration (y-axis) vs. Gaussian curvature (x-axis) derived from a population of computational cell surface reconstructions with poles excluded. 90% bootstrap confidence intervals are displayed as a shaded region about each line. The representative images are selected from one of three biological replicates and the data shown in (F) are pooled from the three biological replicates.

The online version of this article includes the following video and figure supplement(s) for figure 7:

**Figure supplement 1.** MreB is essential in the G27 derivative LSH100.
**Figure supplement 2.** MreB enrichment decreases with increasing positive Gaussian curvature.
**Figure supplement 3.** Curvature enrichment analysis of biological replicates of MreB.
**Figure 7—video 1.** Volumetric rendering and z-slices of the example cells in *Figure 7*.
https://elifesciences.org/articles/52482#fig7video1

---

(*Figure 8E*). The I55A variant showed lower steady-state protein levels than wild-type, while the L110S variant consistently showed higher steady-state protein levels than wild-type. In both cases, the mutant strains displayed a morphology indistinguishable from a *ccmA* null strain (*Figure 8F* and *Figure 8—figure supplement 1C*), suggesting that formation of higher-order structures by CcmA may be necessary for cell shape-determining functions.

## CcmA localization to positive curvature correlates with cell wall synthesis, CcmA polymerization, and helical cell shape

To determine the subcellular localization of CcmA, we performed immunofluorescence of HJH1 cells expressing a 2X-FLAG epitope tag at the native locus under endogenous control as the sole copy of CcmA (*Figure 9A,C and D* and *Figure 9—video 1*). As shown previously (*Blair et al., 2018*), helical morphology is retained upon addition of the 2X-FLAG tag to the wild-type protein. Wild-type CcmA was observed at the cell boundary as puncta and short arcs and was largely absent from the center of the cell, indicating an association with the cell membrane (*Figure 9D* and *Figure 9—video 1*). Puncta were in some cases present as lines of dots roughly parallel to the helical (long) axis of the cell, but were also found distributed along the cell surface. Immunofluorescence was also performed on cells expressing wild-type or polymerization defective CcmA (CcmA$^{I55A}$ and CcmA$^{L110S}$) using antisera raised against *H. pylori* CcmA (*Figure 9B,E–J* and *Figure 9—video 1*). Immunostaining with CcmA preimmune serum showed background signal in the interior of wild-type and mutant cells (*Figure 9—figure supplement 1*). In contrast to cells expressing the wild-type version of CcmA, the mutant CcmA proteins localized as puncta at the center with minimal signal at the cell boundary (*Figure 9G–J*).

To determine if wild-type CcmA localization corresponds to the peak of higher relative PG synthesis at the major axis area, we performed curvature enrichment analysis of CcmA-2X-FLAG immunofluorescence images of non-dividing cells. CcmA was depleted at the poles (*Figure 10—figure supplement 1*, gold). With or without the poles, we saw a marked preference for the positive helical axis area (*Figure 10* and *Figure 10—figure supplement 1*, red line and shaded box) that overlapped with the positive curvature enrichment peaks of MurNAc-alk and D-Ala-alk (*Figure 10*). The wild-type (no FLAG) negative control was 28.9% of the CcmA-FLAG signal (*Figure 10—figure supplement 2B*). While the negative control showed a small peak at 5 μm$^{-2}$, the magnitude of the CcmA-FLAG peak was far greater (*Figure 10A* and *Figure 10—figure supplement 1*). Biological replicates are shown in *Figure 10—figure supplement 2A*. We also performed curvature enrichment analysis on cells expressing wild-type, I55A, and L110S CcmA immunostained with anti-CcmA. Wild-type had a similar major axis area peak as CcmA-2X-FLAG (*Figure 10—figure supplement 3A*, gold), with a lower magnitude due to a lower signal to noise ratio and an enrichment of background (preimmune) staining at negative Gaussian curvature (*Figure 10—figure supplement 3A*, dotted gray). Preimmune signal was 33.0% of the anti-CcmA signal in wild-type (*Figure 10—figure supplement 3B*). There was no distinguishable curvature preference for I55A or L110S CcmA compared to

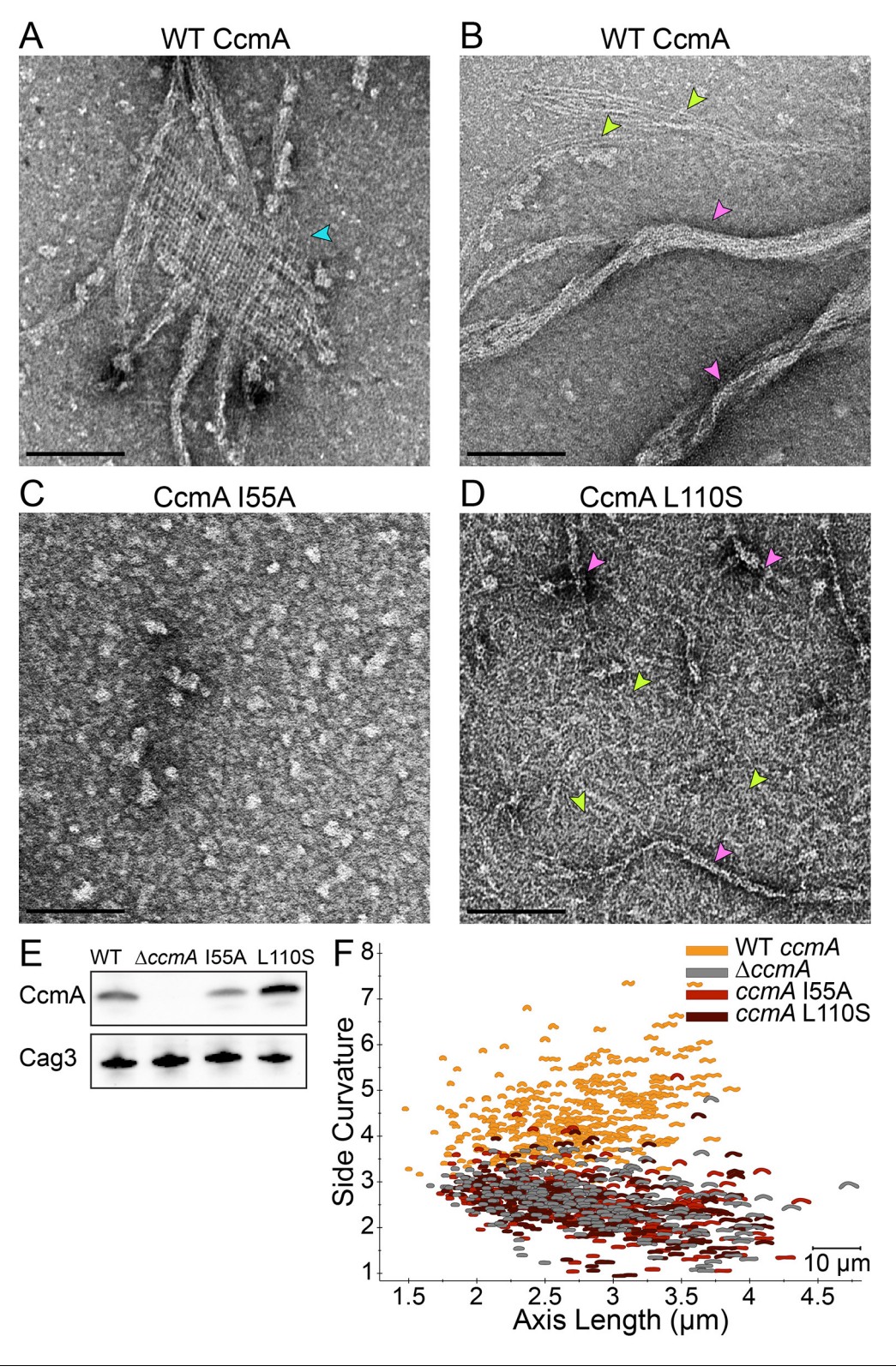

**Figure 8.** Amino acid substitution mutations in CcmA cause altered polymerization in vitro and alter cell shape in vivo. (A–D) Negatively stained TEM images of purified CcmA. Scale bars = 100 nm, with representative images from one of three biological replicates. Wild-type CcmA lattices (A) (blue arrows) and helical bundles (B) (pink arrows), which are comprised of individual filaments (lime green arrows). (C) The I55A variant does not form

*Figure 8 continued on next page*

*Figure 8 continued*

ordered structures in vitro. (D) CcmA$^{L110S}$ filament bundles (pink arrows) and individual filaments (lime green arrows). (E) Immunoblot detection of CcmA expression (top) in *H. pylori* lysates using Cag3 as loading control (bottom); representative of four experiments. (F) Scatterplot displaying axis length (x-axis) and side curvature (y-axis) of wild-type (gold), Δ*ccmA* (gray), *ccmA$^{I55A}$* (red), and *ccmA$^{L110S}$* (dark red) strains. Data are representative of two biological replicates. Wild-type, n = 346; Δ*ccmA*, n = 279; *ccmA$^{I55A}$*, n = 328; and *ccmA$^{L110S}$*, n = 303.
The online version of this article includes the following figure supplement(s) for figure 8:

**Figure supplement 1.** CcmA lattices and bundles.
**Figure supplement 2.** Fourier transform of CcmA lattices shows regular alignment and spacing.

preimmune serum (*Figure 10—figure supplement 3A*, red and dark red vs. dotted light pink and dotted mauve, respectively), indicating that these proteins are unable to localize preferentially to positive Gaussian surface curvature. Preimmune signal was 50.6% and 26.7% of the anti-CcmA signal in I55A and L110S, respectively (*Figure 10—figure supplement 3B*).

To ascertain the impact of deleting *ccmA* on MreB localization and cell wall synthesis patterning, we performed immunostaining for MreB and 18 min MurNAc-alk and D-Ala-alk pulse labeling on Δ*ccmA* cells (JTH6, *amgK murU* Δ*ccmA*, *Figure 11A and B* and *Figure 11—figure supplement 1A and B*; dark pink, dark green, and dark blue, respectively). In Δ*ccmA* cells, MreB is present as small foci (*Figure 11—figure supplement 2* and *Figure 11—video 1*). New cell wall labeling with Mur-NAc-alk is present as dispersed sidewall labeling with some subtle circumferential banding, while labeling with D-Ala-alk is present as clear circumferential bands along the length of the sidewall (*Figure 11—figure supplement 3* and *Figure 11—video 1*). MreB curvature preference appears largely similar in both wild-type (HJH1, *amgK murU*, light pink) and Δ*ccmA* with poles excluded (JTH6, *amgK murU* Δ*ccmA*, dark pink) (*Figure 11A*). When poles are included in the analysis, MreB curvature preference differs more between wild-type and Δ*ccmA*, though the general pattern of enrichment at negative Gaussian curvature remains (*Figure 11—figure supplement 1A*). In contrast, MurNAc-alk and D-Ala-alk patterning change with loss of CcmA; there is greater relative enrichment at low magnitude negative Gaussian curvature in Δ*ccmA* cells (dark green and dark blue) compared to wild-type cells (light green and light blue). Additionally, in Δ*ccmA* cells the enrichment at positive Gaussian curvature is both less pronounced and shifted to lower Gaussian curvature than that of wild-type (*Figure 11B* and *Figure 11—figure supplement 1B*). There is a small peak for MreB at approximately 3 μm$^{-2}$, however interpretation of the MreB peak is complicated by the presence of a peak at the same curvature range for the preimmune signal. For Δ*ccmA*, mock signal was 2.8% of the D-Ala-alk signal, 0.6% of the MurNAc-alk signal, and preimmune signal was 34.6% of anti-MreB signal (*Figure 11—figure supplement 1C*, dotted and solid dark blue and dotted and solid dark pink, respectively). These data suggest that proper localization of CcmA to the major helical axis may be required for promoting extra cell wall synthesis at the major axis area and patterning helical cell shape.

## Discussion

Bacterial cell shape is driven by patterning the cell wall. Maintenance of a cylindrical rod form in a variety of bacteria relies on the action of the actin-like protein MreB, which helps to pattern PG synthesis along the sidewall (*Typas et al., 2012*; *Zhao et al., 2017*). Detailed analysis of MreB localization in the Gram-negative straight-rod *E. coli* indicates that centerline straightness and diameter uniformity rely on MreB curvature enrichment (*Bratton et al., 2018*; *Ursell et al., 2014*), which may result from circumferential motion about the cell (*Wong et al., 2019*). One working model is that MreB localization and cell wall synthesis are enhanced at cell wall dimples (negative Gaussian curvature), cylindrical regions (zero Gaussian curvature), and limited at cell wall bulges (positive Gaussian curvature). This pattern minimizes local curvature as growth progresses (*Figure 11C*, left). While such a growth pattern is at odds with maintaining areas of negative and positive Gaussian curvature required for curved- and helical-rod shapes, MreB is present in many bacteria with these shapes. To be able to maintain curvature in the presence of MreB, the curved-rod shaped Gram-negative Proteobacteria *Caulobacter crescentus* and *Vibrio cholerae* appear to limit relative levels of PG synthesis at negative curvatures through the action of long, cell-spanning cytoskeletal filaments (CreS and

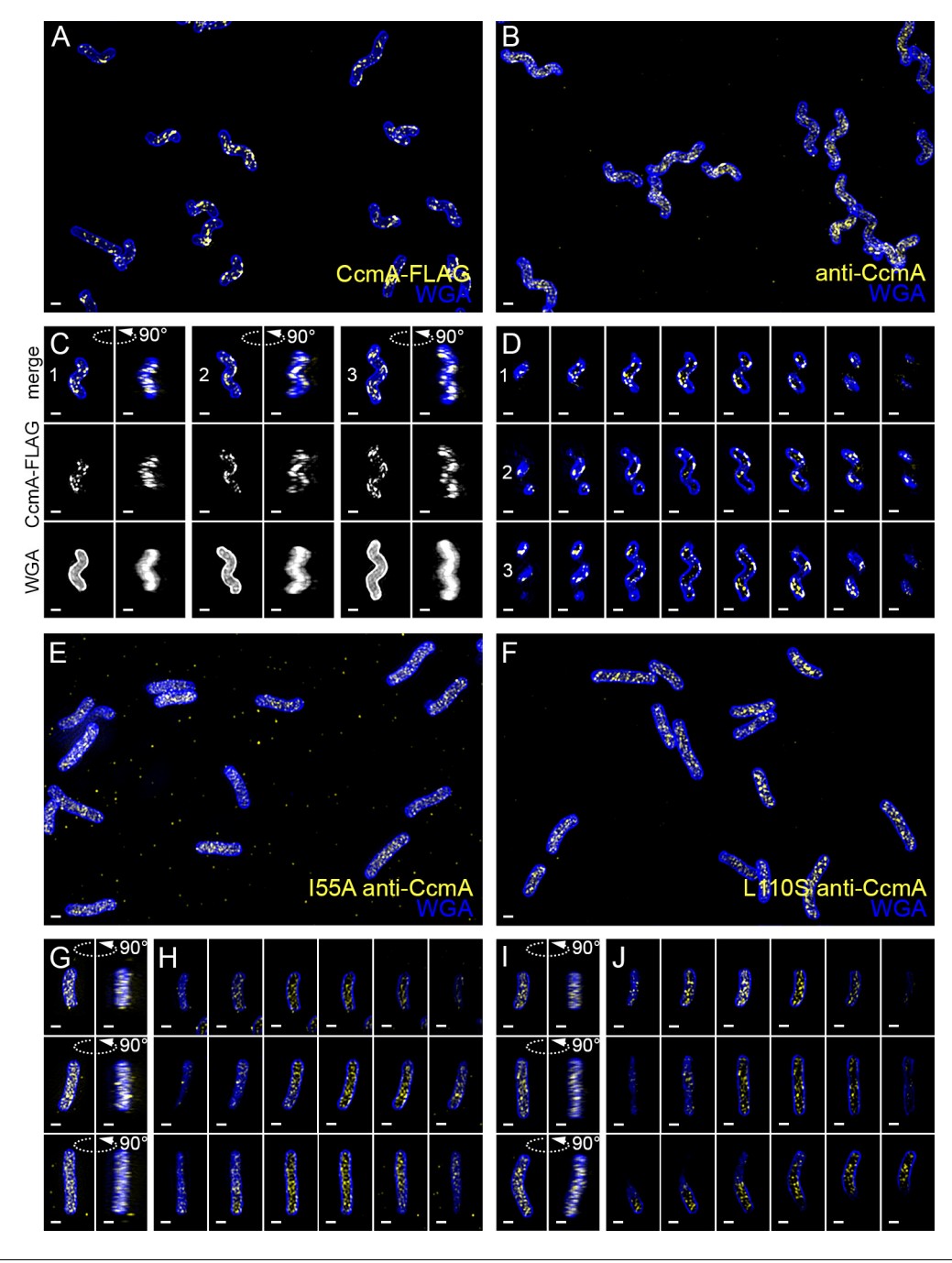

**Figure 9.** Wild-type CcmA appears as short foci on the side of the cell, but CcmA mutants I55A and L110S appear as foci in the interior of the cell. 3D SIM imaging of CcmA-FLAG cells immunostained with M2 anti-FLAG (A, C, D, yellow) or wild-type or CcmA amino acid substitution mutant cells immunostained with anti-CcmA (B, E–J, yellow); cells counterstained with fluorescent WGA (blue). (A) Color merged maximum projection of CcmA-FLAG immunostained with anti-FLAG and counterstained with fluorescent WGA. (B) Color merged field of view of wild-type cells immunostained with anti-CcmA and counterstained with fluorescent WGA. (C) Top-down (left) and 90-degree rotation (right) 3D views of three individual CcmA-FLAG cells. Top: color merge; middle: anti-FLAG; bottom: fluorescent WGA. (D) Color merged z-stack views of the three CcmA-FLAG cells in (C). (left to right = top to bottom of the cell). Numbering indicates matching cells. (E, F) Color merged field of view of I55A or L110S CcmA, respectively, immunostained with anti-CcmA and counterstained with fluorescent WGA. Top-down (left) and 90-degree rotation (right) 3D views of three individual I55A (G) or L110S (I) cells. (H, J) Color merged z-stack

*Figure 9 continued on next page*

*Figure 9 continued*

views of the three I55A cells in (**G**) or L110S cells in (**I**), respectively (Left to right = top to bottom of the cell). Scale bar = 0.5 μm. The representative images are selected from one of three biological replicates.

The online version of this article includes the following video and figure supplement(s) for figure 9:

**Figure supplement 1.** There is low signal in the no-FLAG and preimmune serum controls.
**Figure 9—video 1.** Volumetric rendering and z-slices of the example cells in *Figure 9* and three example WT cells immunostained with anti-CcmA and counterstained with fluorescent WGA.
https://elifesciences.org/articles/52482#fig9video1

---

CrvA) that preferentially localize to the minor axis (negative Gaussian curvature) and enable cells to increase relative synthesis rates on the opposite side of the wall (positive Gaussian curvature) (*Bartlett et al., 2017*; *Cabeen et al., 2009*). We propose that the helical Proteobacterium *H. pylori* uses different mechanisms than *C. crescentus* and *V. cholerae* to maintain the even higher levels of negative and positive Gaussian curvature required for its helical cell shape; *H. pylori* leverages the bactofilin CcmA, which localizes preferentially to the major helical axis area, to promote synthesis at positive Gaussian curvatures on the sidewall and supplements the MreB-associated enhanced

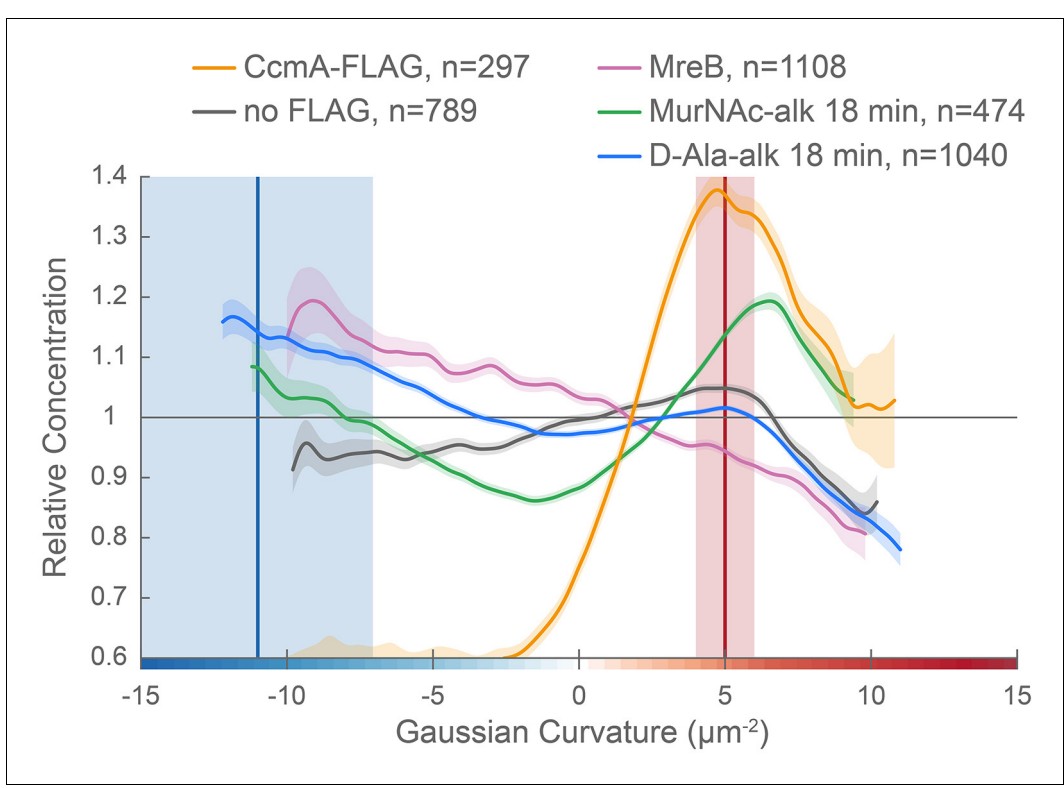

**Figure 10.** CcmA curvature preference correlates with the peak of new PG incorporation at the major axis area and MreB curvature preference correlates with new PG enrichment at negative Gaussian curvature. Overlay of sidewall only surface Gaussian curvature enrichment of relative concentration (y-axis) vs. Gaussian curvature (x-axis) from a population of computational cell surface reconstructions with poles excluded of CcmA-FLAG (gold), no-FLAG control (gray), MreB (pink, from *Figure 7F*), MurNAc-alk (green, from *Figure 6C*), and D-Ala-alk (blue, from *Figure 6C*). The represented data are pooled from three biological replicates. Blue and red vertical lines and shaded regions indicate the average ±1 standard deviation Gaussian curvature at the minor and major helical axis, respectively.

The online version of this article includes the following figure supplement(s) for figure 10:

**Figure supplement 1.** CcmA is excluded from the poles.
**Figure supplement 2.** Curvature enrichment analysis of biological replicates of CcmA-FLAG.
**Figure supplement 3.** CcmA mutants are not enriched at positive Gaussian curvature.

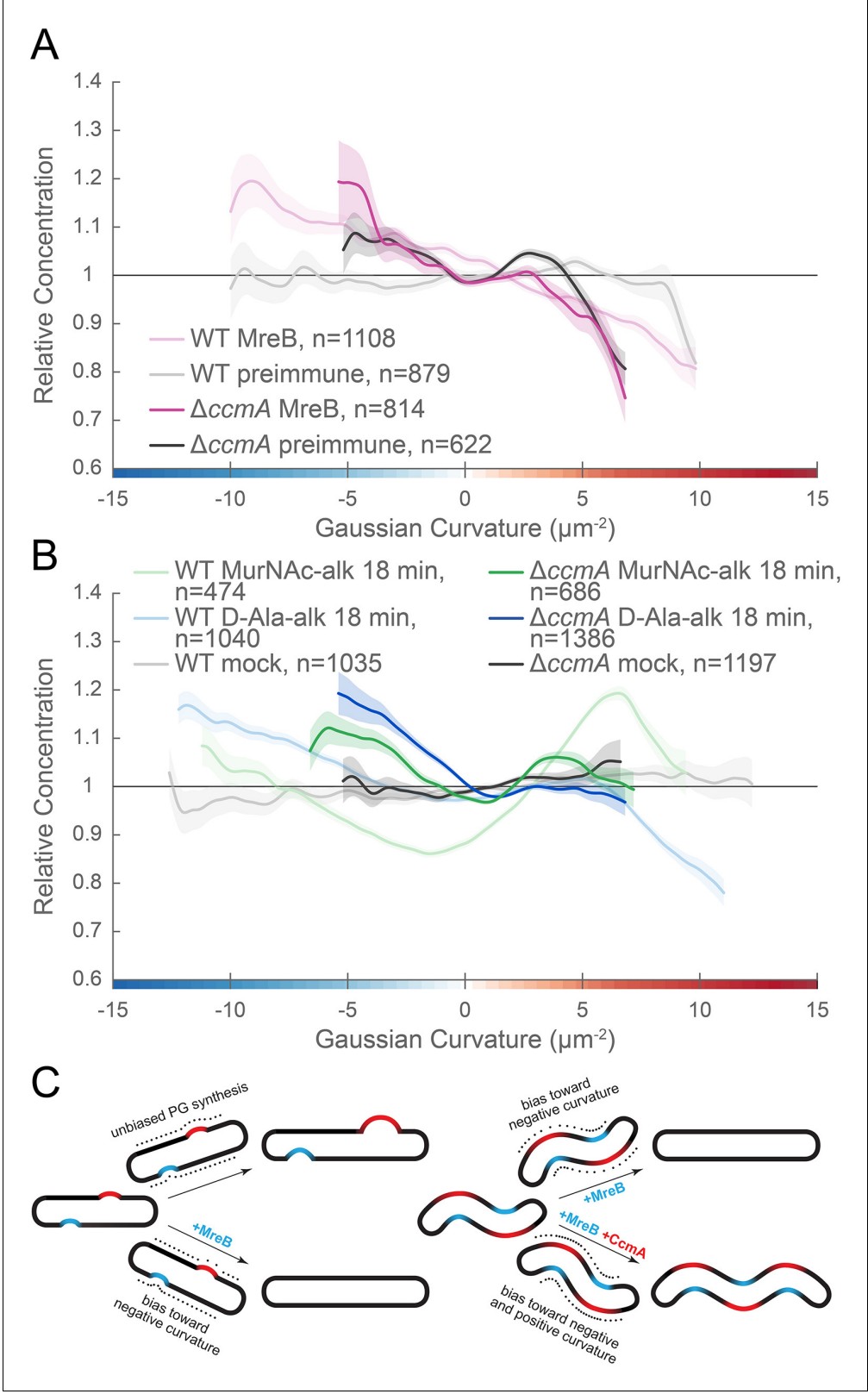

**Figure 11.** MreB and CcmA contribute to cell wall synthesis patterning. (A, B) Sidewall only Gaussian curvature enrichment of relative concentration (y-axis) vs. Gaussian curvature (x-axis) from a population of computational cell surface reconstructions of HJH1 (*amgK murU*) and JTH6 (*amgK murU ΔccmA*) cells immunostained with (A) anti-MreB (HJH1, light pink; JTH6, dark pink) or preimmune serum (HJH1, light gray; JTH6, dark gray) or (B) 18 min

*Figure 11 continued on next page*

*Figure 11 continued*

MurNAc-alk (HJH1, light green; JTH6, dark green) or D-Ala-alk (HJH1, light blue; JTH6, dark blue) pulse-labeled or mock-labeled (HJH1, light gray; JTH6, dark gray) cells. 90% bootstrap confidence intervals are displayed as a shaded region about each line. The represented data are pooled from three biological replicates. (**C**) Model of the contribution of synthesis patterning to rod and helical shape maintenance. Dots indicate different densities of cell wall synthesis that can decrease or propagate non-zero Gaussian curvature. Colored shading indicates local regions of positive (red) and negative (blue) Gaussian curvature.

The online version of this article includes the following video and figure supplement(s) for figure 11:

**Figure supplement 1.** Cell wall synthesis patterning but not MreB curvature preference is altered by loss of CcmA.

**Figure supplement 2.** MreB is present as small foci along the sidewall in Δ*ccmA*.

**Figure supplement 3.** New cell wall growth appears as diffuse labeling and circumferential bands dispersed along the sidewall, excluded from poles, and present at septa in Δ*ccmA*.

**Figure 11—video 1.** Volumetric rendering and z-slices of the example cells in *Figure 11—figure supplements 2* and *3*.

https://elifesciences.org/articles/52482#fig11video1

---

synthesis that is enriched at negative Gaussian curvatures (the minor helical axis) (*Figure 11C*, right). Adding the contribution of CcmA to the PG synthesis patterning allows *H. pylori* to maintain curvatures in the presence of MreB-associated PG synthesis.

To probe cell wall synthesis patterns in *H. pylori*, we used distinct metabolic probes to label the sugar (MurNAc-alk) and peptide (D-Ala-alk) portions of the polymer. While both probes indicate enhanced synthesis at the major and minor helical axes relative to the rest of the sidewall, there were considerable differences in enrichment peak magnitudes between the MurNAc-alk and D-Ala-alk probes. Modified D-alanine is thought to be incorporated into the cell wall through the action of synthesis-associated D-D-transpeptidases and cell wall-modifying L-D-transpeptidases, potentially complicating interpretation of this label. *H. pylori* does not have any known functional L-D-transpeptidases and no detectable 3–3 crosslinks, a hallmark of L-D-transpeptidase activity (*Costa et al., 1999*; *Sycuro et al., 2010*). Thus, signal from D-Ala-alk likely reports on D-D-transpeptidase activity. It is possible that D-D-transpeptidation may also occur separately from synthesis to promote cell wall remodeling or that the rates of synthesis-associated transpeptidation activity may vary on different sides of the cell. We only observed D-Ala-alk incorporation at the penta position (*Figure 4D* and *Figure 4—figure supplement 4*). *H. pylori* has a pentapeptide-rich cell wall and it is unclear if *H pylori* actively regulates pentapeptide trimming. However, the cell shape determining protein Csd3/HdpA has been shown to have weak pentapeptide carboxypeptidase activity in vitro (*Bonis et al., 2010*). Pentapeptides can also be trimmed via transpeptidase-mediated hydrolysis (*Ghuysen, 1991*). Curvature-biased trimming by either mechanism could also contribute to the difference between the D-Ala-alk and MurNAc-alk curvature enrichment profiles. The MurNAc probes have none of these complications as they are embedded in the glycan.

We provide the first example of MreB curvature enrichment analysis in a curved- or helical-rod bacterium and show that enrichment at negative Gaussian curvature is retained, even across the broad range of curvatures represented on the *H. pylori* sidewall. While there has been a report of MreB being non-essential in *H. pylori* (*Waidner et al., 2009*), the mutated strains could not be revived from frozen stocks when requested. In our strain, we could only knock out *mreB* if we first supplied the cells with a second copy of *mreB* at separate locus, indicating that MreB is functional and important. We propose that MreB promotes the peak of PG synthesis we observed at negative Gaussian curvature given its preference for this curvature in *H. pylori* and its role in localizing PG synthesis activity in other organisms. To enable maintenance of high sidewall curvature in the presence of the MreB-driven straight-rod cell growth pattern, we suggest that *H. pylori* augments the default rod pattern by means of enhanced growth at the major axis area that is independent of MreB (*Figure 11C*).

A major outstanding question is how *H. pylori* enhances PG synthesis activity at the major axis area. Our 3D analysis establishes that the average Gaussian curvature along the major axis is distinct from that along the minor axis (5 vs. $-11\ \mu m^{-2}$, respectively) and that the major axis is on average 70% longer than the minor axis in the strain used here. Cytoskeletal elements can form higher-order

structures that reach a sufficient size scale to be able to sense surface curvature, providing a potential mechanism for targeting synthesis to a specific range of positive Gaussian curvature. The bactofilin CcmA is the only non-essential cytoskeletal protein we have identified in our strain background that makes an indispensable and non-redundant contribution to helical shape maintenance. In contrast to the cell spanning filaments CreS in *C. crescentus* and CrvA in *V. cholerae*, which reside at the minor axis, we show that CcmA is present in cells as numerous puncta that have a preference for the major axis area. We propose that CcmA acts to enhance synthesis on its preferred cell face by promoting PG synthesis locally (at positive Gaussian curvature). In support of this hypothesis, the bactofilins BacA and BacB in *C. crescentus* recruit the PG synthase PBPC to assist in stalk elongation, indicating that they help recruit PG synthesis (*Kühn et al., 2010*). Additionally, our group recently showed that CcmA co-purifies with Csd5 and MurF, an enzyme involved in PG precursor synthesis (*Blair et al., 2018*), and separately that both CcmA and MurF are within the top 20 mass spec hits of a Csd7 immunoprecipitation (*Yang et al., 2019*). Furthermore, we demonstrate that in the absence of CcmA, similarly to in wild-type, MreB is still enriched at negative Gaussian curvature, but that MurNAc-alk and D-Ala-alk synthesis patterning shift to more closely resemble the MreB curvature enrichment profile. In $\Delta ccmA$, synthesis at negative Gaussian curvature makes a much more significant contribution to the overall synthesis pattern than does synthesis at positive curvature, as seen by the greater relative concentration at Gaussian curvature values below 0 $\mu m^{-2}$. The MurNAc-alk and D-Ala-alk signals do show a subtle peak at low magnitude positive Gaussian curvature (approximately 3 $\mu m^{-2}$), however the peak is far less prominent (greatly reduced peak to trough distance). Given that there is still some curvature in $\Delta ccmA$ cells, it is not necessarily surprising that there is still some enrichment at positive Gaussian curvature. CcmA is one of a suite of proteins required for helical cell shape maintenance; it is possible that other cell shape proteins can influence PG synthesis to promote some limited curvature in the absence of CcmA, consistent with multiple complementary mechanisms being required for helical shape maintenance.

It is possible that CcmA may also help promote localized crosslink trimming, as loss of CcmA results in an increased degree of crosslinking in the sacculus (*Sycuro et al., 2010*). Crosslink trimming may help promote synthesis but could also play some other role in helical shape maintenance. CcmA dynamics could also influence its ability to promote cell shape. While CcmA does not require a nucleotide cofactor for polymerization, it may be mobile through coupling with the motion of PG synthesis machinery. In other organisms, MreB filaments travel in a roughly circumferential path around the cell and we expect MreB to behave similarly in *H. pylori*.

Loss of CcmA results in cells with highly diminished cell curvature and without significant helical twist. Beyond helping promote curvature by localized PG synthesis, it is possible that CcmA also helps generate twist. We observed helical bundles of filaments in vitro by TEM. These bundles are far longer than the foci we see by immunofluorescence, but foci within the cell may consist of short twisted filament bundles and/or skewed lattices. While it remains unclear how filament or lattice twist would be coupled to cell wall twist, the bactofilin LbbD modulates helical pitch in the spirochete *Leptospira biflexa* (*Jackson et al., 2018*). Both CcmA point mutant variants show altered or no polymerized structures under a variety of buffer conditions in vitro and fail to localize to the cell envelope in vivo. It is still unclear which structures are relevant and if altering higher-order structures abolishes CcmA function by disrupting protein-protein interactions and/or CcmA localization.

Overall, our results are consistent with a model in which MreB-patterned straight-rod shape is the default pattern for *H. pylori* cells and helical shape is facilitated by adding major axis area PG synthesis via CcmA to augment straight-rod cell wall patterning. The enrichment of new cell wall synthesis to both negative Gaussian curvature, as expected for straight-rod shape, and to the major axis area indicates one mechanism for achieving helical shape, but it is not apparent how this growth pattern on its own could be sufficient for helical shape maintenance. The lower relative amount of synthesis at Gaussian curvatures corresponding to the sides of the cell body in comparison to the major and minor axis areas is both unexpected and counterintuitive; it suggests additional mechanisms may be required to maintain helical shape. Indeed, the noted difference between enrichment of D-Ala-alk and MurNAc-alk suggests that spatially-coordinated cell wall modification occurs. Curvature-dependent differences in crosslinking could alter cell wall mechanical properties and PG density; perhaps the PG at the side of the cell is less dense, thus requiring less PG synthesis during growth. Furthermore, our labeling strategy allowed us to determine the curvature bias of new PG insertion, but spatially-regulated turnover of old PG may also contribute to cell wall homeostasis. We also do not

know if super-twisting of the cell wall occurs during growth: does PG on the major axis remain at the major axis as the cell grows?

We employed sophisticated computational tools to demonstrate that *H. pylori* must achieve a much broader distribution of sidewall Gaussian curvature than the curved-rod bacteria *C. crescentus* and *V. cholerae* and that it uses distinct mechanisms to achieve these curvatures. In elucidating the spatial patterning of new cell wall synthesis, we have revealed one of the downstream mechanisms of *H. pylori*'s cell shape-determining program.

# Materials and methods

## Key resources table

| Reagent type (species) or resource | Designation | Source or reference | Identifiers | Additional information |
|---|---|---|---|---|
| Antibody | Monoclonal ANTI-FLAG M2 antibody produced in mouse | Sigma | Cat# F1804, RRID:AB_262044 | IF(1:200) |
| Antibody | Goat anti-Mouse IgG (H+L) Highly Cross-Adsorbed Secondary Antibody, Alexa Fluor 488 | Invitrogen | Cat# A-11029, RRID:AB_2534088 | IF(1:200) |
| Antibody | Goat anti-Rabbit IgG (H+L) Cross-Adsorbed Secondary Antibody, Alexa Fluor 488 | Invitrogen | Cat#: A-11008; RRID: AB_143165 | IF(1:200) |
| Antibody | Polyclonal rabbit αCcmA | (*Blair et al., 2018*) | | IF (1:200); WB (1:10,000) |
| Antibody | Polyclonal rabbit αMreB (*H. pylori*) | (*Nakano et al., 2012*) | | IF (1:500); WB (1:25,000) |
| Commercial assay, kit | Click-iT Cell Reaction Buffer Kit | Invitrogen | Cat# C10269 | |
| Chemical compound, drug | Alexa Fluor 555 Azide, Triethylammonium Salt | Invitrogen | Cat# A20012 | |
| Chemical compound, drug | D-Ala-alk ((R)—2-Amino-4-pentynoic acid) | Boaopharma | Cat# B60090 | |
| Chemical compound, drug | MurNAc-alk | (*Liang et al., 2017*) | | |
| Chemical compound, drug | MurNAc | Sigma | Cat# A3007 | |
| Chemical compound, drug | Wheat Germ Agglutinin, Alexa Fluor 488 Conjugate | Invitrogen | Cat# W11261 | |
| Chemical compound, drug | Wheat Germ Agglutinin, Alexa Fluor 555 Conjugate | Invitrogen | Cat# W32464 | |
| Other | ProLong Diamond Antifade Mountant | Invitrogen | P36961 | |

## Cultures and growth

*H. pylori* (LSH100 and derivatives, *Table 2*) was grown on horse blood (HB) agar plates (*Humbert and Salama, 2008*) incubated at 37°C under micro-aerobic conditions in either 90% air, 10% $CO_2$ (dual-gas) or in 10% $CO_2$, 10% $O_2$, 80% $N_2$ (tri-gas). For resistance marker selection, HB agar plates were supplemented with 15 µg/ml chloramphenicol, 25 µg/ml kanamycin, or 30 mg/ml sucrose, as appropriate. Liquid *H. pylori* cultures were grown shaking in Brucella broth (BD Biosciences, Sparks, MD) supplemented with 10% heat-inactivated fetal bovine serum (Gemini Bio-Products, West Sacramento, CA) (BB10) at 37°C in tri-gas conditions. For plasmid selection and maintenance, *E. coli* cultures were grown in lysogeny broth (LB) or agar supplemented with 100 µg/ml ampicillin or as described at 37°C.

**Table 2.** Strains used in this study.

| Strain | Genotype/description | Construction | Reference |
|---|---|---|---|
| LSH100 | Wild-type: mouse-adapted G27 derivative | - | *Lowenthal et al., 2009* |
| LSH141 (Δcsd2) | LSH100 csd2::cat | - | *Sycuro et al., 2010* |
| TSH17 (Δcsd6) | LSH100 csd6::cat | - | *Sycuro et al., 2013* |
| LSH108 | LSH100 rdxA::aphA3sacB | - | *Sycuro et al., 2010* |
| HMJ_Ec_pLC292-KU | E. coli TOP10 pLC292-KU | Transformation of TOP 10 with pLC292-KU | This study |
| HJH1 | LSH100 rdxA::amgKmurU | Integration of pLC292-KU into LSH108 | This study |
| IM4 | LSH100 mcGee:mreB | Integration of pIM04 into LSH100 | This study |
| JTH3 | LSH100 ccmA:2X-FLAG:aphA3 | - | *Blair et al., 2018* |
| JTH5 | LSH100 ccmA:2X-FLAG: aphA3 rdxA::amgKmurU | Natural transformation of HJH1 with JTH3 genomic DNA | This study |
| KGH10 | NSH57 ccmA::catsacB | - | *Sycuro et al., 2010* |
| LSH117 | LSH100 ccmA::catsacB | Natural transformation of LSH100 with KGH10 genomic DNA | This study |
| SSH1 | LSH100 ccmA$^{I55A}$ | Natural transformation with ccmA I55A PCR product | This study |
| SSH2 | LSH100 ccmA$^{L110S}$ | Natural transformation with ccmA L110S PCR product | This study |
| LSH142 (ΔccmA) | LSH100 ccmA::cat | - | *Sycuro et al., 2010* |
| JTH6 | LSH100 rdxA::amgK murU ccmA::cat | Natural transformation of HJH1 with LSH142 genomic DNA | This study |

## AmgK MurU strain constuction

AmgK and MurU-encoding sequences were PCR amplified from expression plasmid pBBR-KU (*Liang et al., 2017*) using primers AmgK_BamHI_F and MurU_HindIII_R (*Table 3*). The a*mgK murU* amplification product and plasmid pLC292 (*Terry et al., 2005*) were digested with BamHI-HF and HindIII-HF (New England BioLabs, Ipswich, MA) at 37℃ for 1 hr and cleaned up with the QIAquick PCR Purification Kit (Qiagen, Valencia, CA) according to manufacturer instructions. Insert and vector were then ligated with T4 ligase (New England BioLabs) for 10 min at room temperature, inactivated at 65℃ for 20 min, and stored at −20℃. 1 µl of the ligation mixture was transformed into OneShot TOP10 competent cells (Invitrogen, Carlsbad, CA) according to manufacturer instructions. Cells were plated on LB-ampicillin plates and incubated overnight at 37℃. Colonies were screened by colony PCR using primers AmgK_BamHI_F and MurU_HindIII_R. Plasmid pLC292-KU was purified from the resulting clone, HMJ_Ec_pLC292-KU, using the QIAprep Spin Miniprep Kit (Qiagen) according to manufacturer instructions. Recipient *H. pylori* containing a *aphA3sacB* cassette at the *rdxA* locus (LSH108 *Sycuro et al., 2010*) were transformed with the purified plasmid. Transformants were selected on sucrose plates and kanamycin sensitivity was verified. Genomic DNA was purified using the Wizard Genomic DNA Purification Kit (Promega, Fitchburg, WI) and insertion of *amgK murU* at *rdxA* was verified by PCR amplifying and sequencing the locus using primers RdxA_F1P1 and RdxA_dnstm_RP2. The resulting confirmed strain was named HJH1. JTH6, Δ*ccmA* with *amgK murU* was generated by natural transformation of HJH1 with genomic DNA from LSH142 and selection on chloramphenicol plates. Deletion of *ccmA* was confirmed by PCR.

## *mreB* merodiploid strain construction and quantitative transformation assays

To generate the *mreB* merodiploid strain IM4, the promoter of the operon containing *mreB* and a 5' KpnI site was amplified from LSH100 genomic DNA using primers O#9 ProMreB (KpnI_5') and O#10 ProMreB_R. The *mreB* coding sequence with a 3' XhoI site was PCR amplified using primers O#11

**Table 3.** Primers used in this study.

| Primer name | Sequence (5' to 3') |
| --- | --- |
| AmgK_BamHI_F | GATAGGATCCTGACCCGCTTGACGGCTA |
| MurU_HindIII_R | GTATAAGCTTTCAGGCGCGCTCGC |
| RdxA_F1P1 | CAATTGCGTTATCCCAGC |
| RdxA_dnstm_RP2 | AAGGTCGCTTGCTCAATC |
| O#9 ProMreB (KpnI_5') | TATTGGTACCCGCTTGATGTATTCATCAAAG |
| O#10 ProMreB_R | GATTAATTTGCTAAAAATCATAAAATAAACTCCTTGTTTTG |
| O#11 ProMreB_F | CAAAACAAGGAGTTTATTTTATGATTTTTAGCAAATTAATC |
| O#12 ProMreB (XhoI_3') | TATTCTCGAGTTATTCACTAAAACCCACAC |
| O#36 pMcGee-Insert-F | CTGCCTCCTCATCCTCTTCATCCTC |
| O#45 MreBC-seq-F2 | GCACCTATTTTGGGGTTTGAAACC |
| O#47 MreB-seq-F2 | CATTGAGCGCTGGTTTTAAGGCGGTC |
| O#28 MreBseq-F3 | CGATCGTGTTAGTCAAAGGGCAGGGC |
| O#37 pMcGee-Insert-R | GGTGTACAAACATTTAAAGGTAGAG |
| O#68 McGee-1F | CATTTCCCCGAAAAGTGCCACGAGCTCGAAGGAGTATTGATGAAAAAGG |
| O#69 McGee-1R | CTAGAGCGGCCCCACCGCGGCCATCATTAACATCATTATCG |
| O#70 MCS-kan-F | CTCGAGGGGGGGGCCCGGTACCCACAGAATTACTCTATGAAGC |
| O#71 MCS-kan-R | CCATTCTAGGCACTTATCCCCTAAAACAATTCATCCAGTAA |
| O#72 McGee-2F | TTACTGGATGAATTGTTTTAGGGGATAAGTGCCTAGAATGG |
| O#73 McGee-2R | CGGATATTATCGTGAGATCGCTGCAGACTGGGGGGAAACTCATGGG |
| O#74 McGee-R6K-F | CCCATGAGTTTCCCCCCAGTCTGCAGCGATCTCACGATAATATCCG |
| O#75 McGee-R6K-R | GTAACTGTCAGACCAAGTTTACTGCGGCCGCGCAAGATCCGGCCACGATGCG |
| O#76 R6K-amp-F | CGCATCGTGGCCGGATCTTGCGCGGCCGCAGTAAACTTGGTCTGACAGTTAC |
| O#77 R6K-amp-R | CCTTTTTCATCAATACTCCTTCGAGCTCGTGGCACTTTTCGGGGAAATG |
| O#78 MCS fragment | CCGCGGTGGGGCCGCTCTAGAACTAGTGGATCCCCCGGG CTGCGGAATTCGCTTATCG |
| O#79 McGee-MCS-F | CGATAATGATGTTAATGATGGCCGCGGTGGGGCCGCTCTAG |
| O#80 McGee-MCS-R | GCTTCATAGAGTAATTCTGTGGGTACCGGGCCCCCCCTCGAG |
| Csd1F | GAGTCGTTACATTAATGTGCATATCT |
| G1480_DnStrmP2 | AAGGGTGCAATAACGCGCTAA |
| MreB_start_F | ATGATTTTTAGCAAATTAATCGG |
| MreB_cat_up_R | CACTTTTCAATCTATATCCGTGCCTCCGCCAATATC |
| C1 | GATATAGATTGAAAAGTGGAT |
| C2 | TTATCAGTGCGACAAACTGGG |
| Cat_mreB_dn_F | AGTTTGTCGCACTGATAAACTGAAATTGGCG |
| MreB_end_R | TTATTCACTAAAACCCACACGGCTGA |
| FabZ_up_F | GCTATCCCATGCTATTGATAGAC |
| Cat_mid_R | GTCGATTGATGATCGTTGTAACTCC |
| MreB_mid_dn_F | GATCAAAGCATCGTGGAATACATCC |
| Supp2_junc1_R_mid | AATTTGCTAAAAATCACTAA |
| MreB_up | AATACCAGCAACTTTTCAAAA |
| Supp1_Junction1_R | ATTTGCTAAAAACACACGGC |
| Catout | CCTCCGTAAATTCCGATTTGT |
| McGee_187 | GCGAGTATTACCACAAGTTTTC |
| CcmA SDM mi R | AGACTAGATTGGATCATTCCCTATTTATTTTCAATTTTCT |
| CcmA SDM mi F | ATAAAGAAAGGAGCATCAGATGGCAATCTTTGATAACAAT |

*Table 3 continued on next page*

*Table 3 continued*

| Primer name | Sequence (5' to 3') |
| --- | --- |
| CcmA SDM up R | ATTGTTATCAAAGATTGCCATCTGATGCTCCTTTCTTTAT |
| CcmA SDM dn F | AGAAAATTGAAAATAAATAGGGAATGATCCAATCTAGTCT |
| CcmA SDM dn R | GCTCATTTGAGTGGTGGGAT |
| SDM 155A F | ATTCTAAAAGCACGGTGGTGgcCGGACAAACCGGCTCGGTAG |
| SDM 155A R | CTACCGAGCCGGTTTGTCCGgcCACCACCGTGCTTTTAGAAT |
| SDM L110S F | TGGTGGAAAGGAAGGGGATTtcGATTGGGGAAACTCGCCCTA |
| SDM L110S R | TAGGGCGAGTTTCCCCAATCgaAATCCCCTTCCTTTCCACCA |

ProMreB_F and O#12 ProMreB (XhoI_3'). These products were joined using PCR SOEing (*Horton, 1995*). A modified Bluescript SK vector, pDCY40, containing the RK6 origin and *aphA3* flanked by two 550 bp segments of DNA from a previously characterized neutral locus (McGee locus) located between HPG27_186 and HPG27_187 (*Langford et al., 2006*). pDCY40 was constructed using isothermal assembly (*Gibson et al., 2009*) of six pieces amplified using primers O#68 McGee-1F, O#69 McGee-1, O#70 MCS-kan-F, O#71 MCS-kan-R, O#72 McGee-2, O#73 McGee-2R, O#74 McGee-R6K-F, O#75 McGee-R6K-R, O#76 R6K-amp-F, O#77 R6K-amp-R, O#78 MCS fragment, O#79 McGee-MCS-F, and O#80 McGee-MCS-R. The PCR SOEing product and pDCY40 were digested with KpnI and XhoI and ligated to generate vector pIM04DY containing the promotor-*mreB* fusion with flanking McGee locus sequences. pIM04DY was transformed into Chung competent DH5αλpir cells and selected on LB plates with 50 µg/ml ampicillin and 0.2% glucose. The pIM04DY insert was sequence confirmed using primers O#36 pMcGee-Insert-F, O#45 MreBC-seq-F2, O#47 MreB-seq-F2, O#28 MreBseq-F3, and O#37 pMcGee-Insert-R. Linear DNA was PCR amplified from pIM04DY using primers O#73 McGee-2R and O#68 McGee-1F. LSH100 was transformed with this PCR product and kanamycin resistant clones were verified by Sanger sequencing. IM4 was generated by back-crossing LSH100 with genomic DNA from one of these verified clones.

*ccmA::CAT* linear DNA was PCR amplified from LSH142 (ΔccmA) genomic DNA (*Sycuro et al., 2010*) using primers csd1F and G1480_DnStrmP2. *mreB::CAT* linear DNA was generated using previously published methods (*Sycuro et al., 2010*). Briefly, PCR products were amplified from LSH100 genomic DNA using primers MreB_start_F and MreB_cat_up_R for the upstream fragment and Cat_mreB_dn_F and MreB_end_R for the downstream fragment. The CAT cassette was amplified from LSH123 (Δcsd5) genomic DNA (*Sycuro et al., 2012*) using primers C1 and C2. These products were annealed using PCR SOEing (*Horton, 1995*). For transformations, LSH100 and IM4 were grown up to mid-log phase in liquid. $4.5 \times 10^5$ cells in liquid were spotted onto plates, allowed to dry, and were incubated three hours prior to transformation. Each transformation was performed in triplicate. 300 ng of either *mreB::CAT* or *ccmA::CAT* linear DNA was mixed with each cell patch. Transformations were incubated overnight and then each cell patch was resuspended in BB10, serially diluted, and spread on non-selective plates for colony counts and chloramphenicol plates for selection of transformants. Colonies were counted after six days. Plates without colonies after six days were incubated for three weeks to allow any slowly growing colonies to arise. Genomic DNA was purified from the two transformants of LSH100 (clone 1 and 2) with *mreB::CAT*. Sanger sequencing was performed on recombinant clone 1 and 2. For sequencing clone1, sequencing template was PCR amplified from genomic DNA using primers FabZ_up_F and Cat_mid_R and sequenced using primers Supp1_Junction1_R and MreB_up. Additional sequencing template for clone 1 was PCR amplified using primers MreB_mid_dn_F and Cat_mid_R and sequenced using primer MreB_mid_dn_F. For sequencing clone 2, template was PCR amplified from genomic DNA using primers Supp2_junc1_R_mid and MreB_up and sequenced using primers Supp2_junc1_R_mid and MreB_up. Additional sequencing template was PCR amplified using primers MreB_mid_dn_F and Cat_mid_R and sequenced using primers MreB_mid_dn_F and Cat_mid_R. Genomic DNA was purified from eight transformants per transformation of IM4 with *mreB::CAT*. PCR with primers Catout, MreB_up, and McGee_187 was used to determine which copy of *mreB* in each clone was disrupted.

### *ccmA* point mutation strain construction

Strains containing CcmA amino acid substitution mutations were created based on previously published methods (*Sycuro et al., 2010*). Briefly, PCR products were amplified from pKB69H (I55A) or pKB72D (L110S) using primers CcmA SDM mi F and CcmA SDM mi R (*Table 3*). Those products were annealed using PCR SOEing (*Horton, 1995*) to fragments amplified from WT *H. pylori* flanking the CcmA locus using primers Csd1F and CcmA SDM up R (upstream fragment, 810 bp flanking) and CcmA SDM dn F and CcmA SDM dn R (downstream fragment, 540 bp flanking). PCR product was transformed into a *catsacB ccmA* knockout strain LSH117 (LSH100 naturally transformed with KGH10 [*Sycuro et al., 2010*] genomic DNA) and colonies resistant to sucrose and susceptible to chloramphenicol were validated using PCR and Sanger sequencing. Single clones of colonies containing correct mutations were used for all experiments.

### Fosfomycin rescue with MurNAc

Overnight liquid cultures of HJH1 and parent strain LSH108 grown to an optical density at 600 nm ($OD_{600}$) of 0.3–0.5 $OD_{600}$/ml were diluted in BB10, BB10 containing fosfomycin, or BB10 containing fosfomycin and MurNAc to yield cultures at 0.002 $OD_{600}$/ml, with 50 µg/ml fosfomycin, or 50 µg/ml fosfomycin and 4 mg/ml MurNAc, as appropriate. Cultures were grown shaking in 5 ml polystyrene tubes. Samples were taken initially and after 12 hr. 10 µl of culture was diluted into 30 µl of BB10 and a 10-fold dilution series was performed from this initial dilution. 4 µl of each dilution for each experimental condition was spotted on plates and plates were incubated 5–6 days. One biological replicate is defined as beginning with a new overnight liquid culture.

### Synthesis and characterization of MurNAc-alk

MurNAc-alk was synthesized and characterized as previously described (*Liang et al., 2017*) and underwent multiple rounds of purification using our previously-described autopur preparatory HPLC purification strategy until no more than 5% *N*-hydroxysuccinimide (NHS) remained in the product as judged by H NMR, chemical shift 2.6 ppm. The final MurNAc-alk product was then solubilized in DMSO or water (200 mg/ml) for subsequent bacterial PG labeling experiments.

### PG preps and analysis for D-Ala-alk and MurNAc-alk

330 ml of liquid cultures were grown for six doublings to 1 $OD_{600}$/ml with 100 µg/ml D-alanine-alk ((R)−2-Amino-4-pentynoic acid, Boaopharma, Woburn, MA), 62.5 µg/ml MurNAc-alk, or no additions. Cells were harvested and sacculi were purified as previously described (*Blair et al., 2018*). Briefly, cells were harvested by centrifugation at 4°C, resuspended in PBS, and added dropwise to boiling 8% SDS. SDS was then removed by ultracentrifugation and washing. Then sacculi were resuspended in 900 µl of 10 mM Tris HCl with 10 mM NaCl pH 7.0 and 100 µl of 3.2 M imidazole pH 7.0 and incubated with 15 µl α-amylase (10 mg/ml) (Sigma, St. Louis, MA) for 2 hr at 37°C and 20 µl Pronase E (10 mg/ml) (Fisher Scientific, Pittsburgh, PA) for 1 hr at 60°C. 500 µl of 8% SDS was added and samples were boiled for 15 min. SDS was again removed by ultracentrifugation and washes with water. The purified PG was suspended in 20 mM sodium phosphate pH 4.8 (D-Ala-alk samples) or 20 mM ammonium formate pH 4.8 (MurNAc-alk samples) and incubated overnight with 10 µg of cellosyl (kind gift from Hoechst, Frankfurt am Main, Germany) at 37°C on a Thermomixer at 900 rpm. Following this incubation, the samples were placed in a dry heat block at 100°C for 10 min and centrifuged at room temperature for 15 min at 16,000 × g. The supernatant was retrieved. D-Ala-alk labeled digests were reduced with sodium borohydride (Merck KGaA, Darmstadt, Germany) and separated by RP-HPLC, peaks collected and analyzed using offline electrospray mass spectrometry as previously described (*Bui et al., 2009*).

MurNAc-alk labeled digests (non-reduced) were analyzed via injection onto a capillary (0.5 × 150 mm) ACE Ultracore 2.5 super C18 column (Hichrom, Lutterworth, UK). The LC-MS instrument configuration comprised a NanoAcquity HPLC system (Waters, Milford, MA) and QTOF mass spectrometer (Impact II, Bruker, Billerica, MA). Buffer A was 0.1% formic acid (VWR, Lutterworth, UK) in water (VWR). Buffer B was 0.1% formic acid in acetonitrile (VWR). RP-HPLC conditions were as follows: 0% buffer B for 3 min, 1.5% B at 20 min, 3.0% B at 35 min, 15% B at 45 min, 45% B at 50 min, followed by 2 min at 85% B and finally 15 min re-equilibration at 0% B. The flow rate was 0.02 ml/min and the capillary column temperature was set at 35°C.

MS data was collected in positive ion mode, 50–2000 m/z, with capillary voltage and temperature settings of 3200 V and 150℃ respectively, together with a drying gas flow of 5 L/min and nebulizer pressure of 0.6 Bar. The resulting MS spectral data was analyzed using Compass DataAnalysis software (Bruker).

## 18 min pulses with D-Ala-alk and MurNAc-alk

400 µl of HJH1 overnight liquid cultures in BB10 grown to 0.3–0.5 OD$_{600}$/ml was added to a 5 ml polystyrene round bottom tube and equilibrated in the 37℃ Trigas incubator for 15 min before addition of the metabolic probe. 8 µl of a 200 mg/ml MurNAc-alk stock in DMSO or water (final concentration = 4 mg/ml) or 4 µl of a 100 mM stock of D-Ala-alk ((R)−2-Amino-4-pentynoic acid, Boaopharma) in water was added to the culture. The culture was incubated for 18 min and growth was arrested by adding 4 µl of 10% sodium azide and placing cultures on ice for 5 min. Cells were transferred to a 1.5 ml microcentrifuge tube, pelleted in a microcentrifuge for 5 min at 5000 rpm, and resuspended in 1 ml Brucella broth. Paraformaldehyde was added to a final concentration of 4%. Cells were fixed at room temperature for 45 min, pelleted, and resuspended in 70% ethanol. Cells were permeabilized on ice for 30 min, pelleted, and resuspended in PBS. Cell suspension density was normalized between samples using a hemocytometer and cells were spun onto clean glass coverslips at 500 rpm for 5 min in a Hettich Rotana 460R swinging bucket centrifuge. Click chemistry was performed on coverslips using the Click-iT Cell Reaction Buffer Kit (Invitrogen) according to manufacturer instructions (without BSA washes) with 8 µg/ml Alexa Fluor 555 Azide (Invitrogen). Coverslips were washed two times with 0.05% Tween-20 in PBS (PBST) for 10 min each and were then stained with 30 µg/ml WGA-Alexa Fluor 488 (Invitrogen) in PBS for 30 min at room temperature. Coverslips were washed an additional four times in PBST and mounted on slides with Prolong Diamond antifade (Invitrogen). Slides were cured for a week before imaging. One biological replicate is defined as beginning with a new overnight liquid culture.

## Immunofluorescence (CcmA-FLAG, CcmA, MreB)

Overnight liquid cultures in BB10 grown to 0.3–0.5 OD$_{600}$/ml were fixed at room temperature for 45 min with 4% paraformaldehyde. Cells were pelleted in a TOMY TX-160 micro centrifuge for 5 min at 5000 rpm and resuspended in 0.1% Triton X-100 in PBS for one hour at room temperature to permeabilize the cells. Cells were then pelleted in an Eppendorf microfuge at 2400 rpm for 10 min and resuspended in PBS. Cell suspension density was normalized using a hemocytometer and cells were spun onto clean glass coverslips at 500 rpm for 5 min in a Hettich Rotana 460R swinging bucket centrifuge. Coverslips were stained with 30 µg/ml WGA-Alexa Fluor 555 (Invitrogen) in PBS for 30 min at room temperature, washed four times with 0.05% Tween-20 in PBS (PBST) for 10 min each, blocked for two hours with 5% goat serum (Sigma) in PBST at room temperature, and then incubated overnight at 4℃ in primary antibody in 5% goat serum PBST. Mouse anti-FLAG M2 (Sigma, RRID:AB_262044), rabbit anti-CcmA (*Blair et al., 2018*), and CcmA preimmune serum were used at a 1:200 dilution. Rabbit anti-MreB and MreB preimmune serum (a gift from Dr. Hong Wu and Dr. Kouichi Sano *Nakano et al., 2012*) were used at a 1:500 dilution. After primary antibody incubation, coverslips were washed four times in PBST and incubated with 1:200 Alexa Fluor 488 anti-mouse (A-11029, Invitrogen, RRID:AB_2534088) or 1:200 Alexa Fluor 488 anti-rabbit (A-11008, Invitrogen, RRID:AB_143165) in PBST for 45 min at room temperature. After secondary antibody incubation, coverslips were washed four times in PBST and mounted on slides with Prolong Diamond antifade (Invitrogen). Slides were cured for a week before imaging. For CcmA-FLAG immunofluorescence, strain JTH5 was used. JTH5 was generated by natural transformation of HJH1 with genomic DNA from JTH3 (*Blair et al., 2018*) and selection on kanamycin blood plates. HJH1 was used as the corresponding no-FLAG control, as well as for the anti-MreB and MreB preimmune immunofluorescence. Wild-type LSH100 (*Lowenthal et al., 2009*) was used for anti-CcmA and CcmA preimmune immunofluorescence. One biological replicate is defined as beginning with a new overnight liquid culture.

## 3D structured illumination imaging

Slides for cell surface curvature profiles for LSH100, Δ*csd2*, and Δ*csd6* were imaged on a DeltaVision OMX V4 BLAZE 3D microscope (GE Healthcare Life Sciences, Chicago, IL) equipped with Photometrics Evolve 512 emCCD cameras and an Olympus UPlanApo 100x/1.42 oil objective with oil matched

for the sample refractive index. 512 × 512 pixel images were collected with three msec exposure and 170 EMCCD gain using a 100 mW 488 nm laser with 10% transmission. Z-plane images were acquired with 125 nm spacing. The remaining SIM microscopy was performed on a DeltaVision OMX-SR equipped with PCO scientific CMOS cameras, 488 nm and 568 nm lasers, and an Olympus 60x/1.42 U PLAN APO oil objective with oil matched for the sample refractive index. 512 × 512 pixel Z-plane images with 125 nm spacing and 3 µm thickness were collected. For HJH1 D-Ala-alk samples, images were collected with 5% 488 and 15% 568 laser power for 20 msec and 100 msec exposures, respectively. For JTH6 D-Ala-alk samples, images were collected with 5% 488 and 30% 568 laser power for 20 msec and 100 msec exposures, respectively. For MurNAc-alk samples, images were collected with 10% 488 and 15% or 2% 568 laser power for 2 msec and 80 msec exposures, respectively. For anti-FLAG immunostained samples, images were collected with 10% 488 and 10% 568 laser power and 40 msec and 25 msec exposure, respectively. For HJH anti-MreB immunostained samples, images were collected with 10% 488 and 10% 568 laser power and 70 msec and 25 msec exposure, respectively. For JTH6 α-MreB immunostained samples, images were collected with 20% 488 and 20% 568 laser power, respectively, and 25 msec exposure. For anti-CcmA immunostained samples, images were collected with 15% 488 and 15% 568 laser power and 30 msec and 40 msec exposure, respectively. Images were processed using included Softworx software. Figures were generated by opening files in Fiji (*Schindelin et al., 2012*), adjusting brightness and contrast, and assembling in Adobe Photoshop. Intensity scaling of maximum projection and Z-slice images are equal for all samples within a set (D-Ala-alk and mock; MurNAc-alk and mock; anti-FLAG M2; anti-MreB and preimmune serum; and anti-CcmA and preimmune serum), with the exception of the I55A CcmA anti-CcmA and preimmune images, which were brightened in comparison to other anti-CcmA and preimmune images to compensate for the reduced expression of I55A CcmA. Intensity scaling is equal for I55A CcmA anti-CcmA and preimmune images.

## 3D reconstructions and curvature enrichment

3D cell surfaces were generated from the 3D-SIM OMX software reconstructions using existing software (*Bartlett et al., 2017*; *Bratton et al., 2018*) with parameters optimized for the difference in imaging modality and file formats. This method minimizes the difference between the observed image and a forward convolution model of the true intensity distribution and the microscope's transfer function. While the images generated by 3D-SIM are not precisely equal to the convolution of the true intensity distribution, we consider the observed images as if they had been generated with an effective blurring function that we parameterize as a 3D Gaussian blur. For each individual cell, the reconstruction algorithm returns the 3D shape of the cell as a collection of vertex positions {$V_i$} and a collection of faces defining which vertices are connected to each other. These faces and positions allow us to calculate geometric properties including the volume, surface area, local principal curvatures, etc. (*Bratton et al., 2018*; *Rusinkiewicz, 2004*). The Gaussian curvature at any point on the surface is the product of the principal curvatures and is therefore independent of the sign convention chosen for the principal curvatures. Following reconstruction, each cell surface undergoes a visual inspection quality control step. To estimate the diameter of each cell, we use the distance from each surface point to its nearest centerline point as a proxy for the local radius. The cell diameter is then the weighted average of twice the local radius, weighted by the surface area represented by each vertex.

In addition to the geometric properties of the surface, we calculate the intensity of a secondary fluorophore at the coordinates of the surface, for example D-Ala-alk, MurNAc-alk, or immunofluorescence. For each individual cell, the average surface concentration was calculated as the surface area weighted sum of the fluorescence at the surface divided by the total surface area of that cell. This normalization sets the concentration scale for the enrichment analysis; a value of one is the same concentration as if all the intensity was uniformly spread on the surface, concentrations greater than one are enriched and concentrations less than one are depleted. When considering the entire cell surface, the normalization included all surface vertices. When only considering the sidewalls of the cell, we first removed all the vertices in the polar regions. These regions were defined as all the points on the surface whose nearest centerline point was closer to the pole than 0.75 of the cell diameter (*Figure 1B*). Following normalization, we calculated the geometric enrichment in each individual cell by averaging the concentration across all the vertices of a particular Gaussian curvature. This enrichment profile was then averaged across the entire population of cells. We truncate the

analysis to Gaussian curvatures which have sufficient representation (>4e-4). For error estimation, we report 90% confidence intervals from bootstrap analysis across cells and plot this interval, along with the mean, using cubic smoothing splines (*Figure 6*, lines). Each sample is the composite dataset from three biological replicates.

We approximated the total fluorescent signal from each cell including the contributions from inside the cell and surface intensities. This total signal is a good proxy for the selectivity of the labeling experiments. As a first step, the entire z-stack was summed to make a 2D projection. A thresholded, binary mask of each cell was generated using Otsu's method on the color channel used to generate the computation cell surface reconstruction and dilated by three pixels to make sure that we captured all the intensity in the cell. The total intensity in the corresponding pixels of the other color channel were added together to calculate the total intensity in the cell. To normalize for effects of cell size, this total intensity was divided by the number of pixels in the mask, resulting in the total fluorescence signal/cell.

The MATLAB scripts used to reconstruct cell surfaces and perform the geometric enrichment analyses are publicly available under a BSD 3-clause license at https://github.com/PrincetonUniversity/shae-cellshape-public and archived at https://doi.org/10.5281/zenodo.3627045 and http://arks.princeton.edu/ark:/88435/dsp01h415pd457.

## Determining helical fits of 3D centerlines

To examine the eight helical parameters of each cell's centerline, we adapted the helical fitting algorithm from *Nievergelt (1997)*. The first step in the routine is to estimate a right-cylindrical surface on which all the data lie. This is defined by four parameters, three of which define a vector parallel to the helical axis ($X_a$, $Y_a$, $Z_a$) and the fourth is the cylinder diameter ($D$). The subsequent steps determine the remaining four parameters that define a point on the helix ($X_o$, $Y_o$, $Z_o$) and the helical pitch ($P$). The algorithm takes advantage of the speed of singular value decomposition (SVD) by framing the best fit as a linear algebra problem. The modifications that we made to the algorithm were in a preconditioning step as well as steps 2 and 3. The center of mass of the data was subtracted off from all the observations and then added back into $X_0$, $Y_0$, and $Z_0$. For our real cells the two smallest singular values in step 2.3 are sometimes of similar magnitude and are both checked to see which right-singular vector is more consistent with a cylinder. The use of SVD instead of eigenvalue decomposition does not retain the right-handed convention of space forcing us to switch step 2.4 to an eigenvalue decomposition. In estimating the pitch of the helix in step 3.2, the algorithm by Nievergelt did not support helical data that covered more than one helical turn. This type of data presents a phase wrapping issue. To solve this issue, we first sorted the data by its projected position along the helical axis. We assumed that the relative phase difference between any two subsequent points was close to zero and calculated an absolute phase at each point by summing the relative phase differences along the whole curve. This then allowed us to calculate the relative slope of the helical phase. Here we again had to break from Nievergelt's SVD approach and used simple linear regression to retain the right-handed convention of space.

For each cell that was independently reconstructed, we estimate the best fit helical parameters for the centerline. Because we do not consider the orientation and offset of the helix to be shape parameters, we do not present any statistics on them. To estimate if the best fit helix was consistent with the centerline, we calculated the root mean squared deviation (RMSD) between the observed centerline coordinates and the best fit helix. One third (402/1137) of the cells had centerlines consistent with single helix. From the one third of the population that matched a single helix, we generated synthetic helical rods with the same helical parameters as each individual cell. From these, we compared the simulated and reconstructed cells in terms of their surface area, volume, volume of the convex hull, and Euclidean distance from pole to pole. If any of the parameters from the simulated cell deviated from the measured value by more than 10%, we excluded that cell from the analysis. In the end, we were left with almost 20% (231/1137) of the wild-type cells that were consistent with our model that cell shape is close to a spherocylinder wrapped around a helical centerline.

Synthetic cells were generated using two major components, a helical centerline and a cylindrical coordinate system about that centerline. In cylindrical coordinates (R, θ, L), a cylinder with hemispherical endcaps has a simple form of a constant radius in the cylinder region and parabolic dependence in the endcaps. We then wrap the coordinate system around a helical axis by calculating the Frenet-Serret frame at each point of the helical centerline from the local tangent, normal, and their

cross-product, the binormal. This wraps a fixed angular coordinate θ around the centerline, generating the helical rod surface of interest. However, these surfaces are still in a rectangular format, meaning that they are stored as three matrices {x,y,z} each as a function of the (θ, L). This surface is resampled into a triangular approximation of the surface with approximately equilateral triangles using the surface reconstruction tools that we have previously developed (*Bartlett et al., 2017*; *Bratton et al., 2018*). Some geometric parameters, including the Gaussian curvature at each point on the surface and the surface area and the volume of the cells, can be calculated for both real cell reconstructions and the synthetic cells (*Figure 3C–F*, *Figure 3—figure supplement 1C-E*, and *Figure 3—figure supplement 2*, left column). For these, we defined the pole surface area as the surface within 0.75 cell diameters of the end. Because of their intrinsic unwrap coordinate system, synthetic cells have defined surface helical axes, which allows us to compute the length of the major and minor axes as well as the Gaussian curvature at these axes. Since the decrease in local diameter near the pole changes both the curvature and the length of the helical axis, we calculate the major and minor axis lengths and Gaussian curvatures from the central 50% of the cell, where the measurements are not influenced by the poles (*Figure 3G–H* and *Figure 3—figure supplement 2*, middle and right columns). Decreasing the total length of the cell proportionally decreases the both the sidewall portion of the cell (including surface curvature properties) and the length of the major and minor axes, retaining the same ratio of major axis to minor axis length. As shown in *Figure 3—figure supplement 2A* (center and right columns), the length of the cell has negligible influence on the distribution of surface curvatures and the ratio of major to minor axis length, further validating our aggressive threshold for removing the ends of the cells for these measurements.

The MATLAB scripts used to fit helical centerlines are publicly available under a BSD 3-clause license at https://github.com/PrincetonUniversity/shae-cellshape-public and archived at https://doi.org/10.5281/zenodo.3627045 and http://arks.princeton.edu/ark:/88435/dsp01h415pd457.

## Purification of recombinant 6His-CcmA and variants

Plasmids containing N-terminal 6-histidine fusions to WT CcmA (pKB62) and CcmA containing point mutations were generated using site directed mutagenesis primers (*Table 3*) to generate CcmA I55A (pKB69H; primers SDM I55A F and SDM I55A R) and CcmA L110S (pKB72D; primers SDM L110S F and SDM L110S R). Plasmids were transformed into *E. coli* protein production host BL21. Strains were grown in liquid culture overnight at 37°C in LB with 0.2% glucose and 100 µg/ml ampicillin. The next day, cells were diluted 1/1000 into fresh media without glucose, grown to mid-log (0.5–0.75), chilled on ice for 15 min, then induced for protein expression by adding 1.0 mM IPTG. Flasks were transferred to room temperature and incubated with shaking for 3.5–4 hr. Cells were harvested by centrifugation and either used immediately for protein purification or frozen at $-80°C$. For purification, cells were resuspended in 2/5 culture volume of lysis buffer (25 mM Tris pH 8.0, 2 M urea, 500 mM NaCl, 2% glycerol, 0.5 mg/ml lysozyme) supplemented with ¼ EDTA-free protease inhibitor tablet (Pierce, Waltham, MA) and 2 U Benzonase nuclease (EMD Millipore, Burlington, MA) and incubated at room temperature with gentle rolling for 1 hr. After lysing, cells were sonicated at 20% power with 15 s pulses until all cells were lysed. Lysates were cleared at 5000 x g at 4°C, then applied to equilibrated TALON metal affinity resin (TaKaRa, Shiga, Japan) and incubated for 2 hr at room temperature with gentle rolling. The protein bound to resin was washed twice with wash buffer (25 mM Tris pH 8.0, 2 M urea, 500 mM NaCl, 2% glycerol, 7.5 mM imidazole), and proteins eluted from the resin using 25 mM Tris pH8.0, 2 M urea, 500 mM NaCl, 2% glycerol, 250 mM imidazole). Fractions were analyzed by SDS-PAGE for purity and yield. Protein concentration was determined using a Nanodrop 1000 (Thermo Fisher Scientific, Waltham, MA) using the Protein A280 program. One biological replicate is defined as beginning with a new overnight liquid culture.

## Immunoblotting *H. pylori* extracts

Whole cell extracts were prepared by harvesting 1.0 $OD_{600}$ of log phase (0.3–0.7 $OD_{600}$/ml) *H. pylori* liquid culture by centrifugation for 2 min at max speed in a microcentrifuge and resuspending in 2x protein sample buffer (62.5 mM Tris pH 8, 2% SDS, 0.02% bromophenol blue, 20% glycerol) or Lämmli buffer at 10.0 $OD_{600}$/ml and boiled for 10 min. Whole cell extracts were separated on 4–15% gradient BioRad TGX gels or 4–15% mini-PROTEAN TGX Stain-Free gels (used according to manufacturer instructions) by SDS-PAGE and transferred onto PVDF membranes using the BioRad Turbo-

transfer system according to the manufacturer's instructions (BioRad, Hercules, CA). Membranes were blocked for 2 hr at room temperature with 5% non- fat milk in TBST (0.5 M Tris, 1.5 M NaCl, pH 7.6, 0.05% Tween 20). Membranes were incubated with primary antibody for 2 hr at room temperature or overnight at 4°C with 1:10,000 anti-CcmA primary antibody, 1:20,000 dilution for $\alpha$-Cag3 (*Pinto-Santini and Salama, 2009*), or 1:25,000 dilution for anti-MreB, in TBST. Six washes with TBST over a 30 min period were followed by a 1 hr incubation at room temperature with horseradish peroxidase-conjugated anti-rabbit immunoglobulin G (Santa Cruz Biotechnology, Dallas, TX) antibody at 1:20,000 dilution in TBST. After six washes with TBST over a 30 min period, antibody detection was performed with ECL Plus (Pierce) detection kit or Immobilon Western Chemiluminescent HRP substrate (Millipore), following the manufacturer's protocol and imaged with the BioRad Gel Documentation System. One biological replicate is defined as beginning with a new liquid culture.

## 2D *H. pylori* quantitative cell shape analysis

Phase-contrast microscopy was performed on cells grown in shaken liquid culture until mid-log phase ($OD_{600}$ 0.3–0.6), fixed in a 4% PFA/PBS + 10% glycerol solution, and mounted on glass slides. Resulting images were acquired using a Nikon TE 200 microscope with a 100X oil-immersion objective and Nikon CoolSNAP HQ CCD camera controlled by MetaMorph software (MDS Analytical Technologies, Sunnyvale, CA). Images were thresholded using the ImageJ software package. Quantitative analysis of thresholded images were used to measure both side curvature and central axis length with the CellTool software package as described previously (*Sycuro et al., 2010*). One biological replicate is defined as beginning with a new liquid culture.

## Transmission electron microscopy

For TEM, 10 µM WT, I55A, or L110S CcmA was dialyzed overnight at 4°C against 25 mM Tris pH 8. The proteins were applied to glow-discharged carbon-coated grids and negatively stained with 0.75% uranyl acetate. Images were acquired with JEOL 1400 transmission electron microscope using a Gatan UltraScan 1000xp camera with 2K $\times$ 2K resolution.

## Acknowledgements

This research was supported in part by US National Institutes of Health R01 AI136946 (NRS), U01 CA221230 (CLG and NRS), T32 CA009657 (KMB), T32 GM95421 (SRS), T32 GM008550 (KED), GM113172 (MSV), the FHCRC Cellular Imaging and Genomics and Bioinformatics Shared Resources of the NCI Center Support Grant P30 CA015704, the Stanford Imaging Award Number 1S10OD01227601 from the National Center for Research Resources (NCRR), the Life Sciences Research Foundation (EK), and the Wellcome Trust grant 101824/Z/13/Z (WV).This work was supported by the National Science Foundation Graduate Research Fellowship Program under Grant No. DGE-0718124 (JAT) and DGE-1256082 (JAT and KMB), the Department of Defense (DoD) through the National Defense Science and Engineering Graduate Fellowship (NDSEG) Program (JAT), and the GO-MAP Graduate Opportunity Program Research Assistantship Award (GOP Award) (SRS). This work was supported by National Science Foundation PHY-1734030 (BPB and JWS), the Glenn Centers for Aging Research (BPB), and National Institutes of Health NIH R21 AI121828 (BPB and JWS).

The opinions, findings, and conclusions or recommendations expressed in this material contents are solely the responsibility of the authors and do not necessarily represent the official views of the NCRR, the National Institutes of Health, the Department of Defense, or the National Science Foundation. The authors have no conflicts of interest to report.

We would like to thank Laura Sycuro, Desirée Yang, and Irina Mavrodi for strain construction; Dr. Cintia Santiago for assistance with NAM purification and compound shipment; Patrina Pellett (GE Healthcare) for assistance with OMX imaging; the David Baker Lab (University of Washington) for OMX access and use; and Sloan Siegrist (University of Massachusetts Amhurst) for the D-alanine-D-alanine-alkyne and D-alanine-alkyne-D-alanine reagents. We kindly thank Dr. Hong Wu and Dr. Kouichi Sano (Osaka Medical College) for the anti-MreB and corresponding preimmune sera used in this study; Anson Chan (University of British Columbia) for consultation on CcmA mutant design; and Zachary Jones for assistance in synthesizing the MurNAc sugars.

## Additional information

### Funding

| Funder | Grant reference number | Author |
|---|---|---|
| National Institutes of Health | R01 AI136946 | Nina R Salama |
| National Institutes of Health | U01 CA221230 | Catherine L Grimes<br>Nina R Salama |
| National Institutes of Health | T32 CA009657 | Kris M Blair |
| National Institutes of Health | T32 GM95421 | Sophie R Sichel |
| National Institutes of Health | T32 GM008550 | Kristen E DeMeester |
| National Institutes of Health | P30 CA015704 | Nina R Salama |
| National Center for Research Resources | Stanford Imaging Award Number 1S10OD01227601 | Nina R Salama |
| Wellcome | 101824/Z/13/Z | Waldemar Vollmer |
| National Science Foundation | DGE-0718124 | Jennifer A Taylor |
| National Science Foundation | DGE-1256082 | Jennifer A Taylor<br>Kris M Blair |
| Department of Defense | National Defense Science & Engineering Graduate Fellowship (NDSEG) | Jennifer A Taylor |
| Graduate Opportunities and Minority Achievement Program | Graduate Opportunity Program Research Assistantship Award | Sophie R Sichel |
| National Science Foundation | PHY-1734030 | Benjamin P Bratton<br>Joshua W Shaevitz |
| Glenn Centers for Aging Research | | Benjamin P Bratton |
| National Institutes of Health | R21 AI121828 | Benjamin P Bratton<br>Joshua W Shaevitz |
| National Institutes of Health | GM113172 | Michael S VanNieuwenhze |

The funders had no role in study design, data collection and interpretation, or the decision to submit the work for publication. The opinions, findings, and conclusions or recommendations expressed in this material contents are solely the responsibility of the authors and do not necessarily represent the official views of the NCRR, the National Institutes of Health, the Department of Defense, or the National Science Foundation.

### Author contributions

Jennifer A Taylor, Conceptualization, Formal analysis, Funding acquisition, Investigation, Visualization, Methodology; Benjamin P Bratton, Conceptualization, Software, Formal analysis, Funding acquisition, Investigation, Visualization, Methodology; Sophie R Sichel, Conceptualization, Formal analysis, Funding acquisition, Investigation, Methodology; Kris M Blair, Conceptualization, Funding acquisition, Investigation, Methodology; Holly M Jacobs, Investigation, Methodology; Kristen E DeMeester, Resources, Funding acquisition, Investigation, Visualization, Methodology; Erkin Kuru, Conceptualization, Methodology; Joe Gray, Formal analysis, Investigation, Visualization, Methodology; Jacob Biboy, Formal analysis, Investigation, Methodology; Michael S VanNieuwenhze, Supervision, Funding acquisition, Methodology; Waldemar Vollmer, Formal analysis, Supervision, Funding acquisition, Methodology; Catherine L Grimes, Conceptualization, Resources, Supervision, Funding acquisition, Methodology; Joshua W Shaevitz, Conceptualization, Supervision, Funding acquisition, Methodology; Nina R Salama, Conceptualization, Supervision, Funding acquisition, Project administration

## Author ORCIDs

Benjamin P Bratton (iD) https://orcid.org/0000-0003-1128-2560
Joe Gray (iD) http://orcid.org/0000-0003-2338-0301
Jacob Biboy (iD) http://orcid.org/0000-0002-1286-6851
Joshua W Shaevitz (iD) https://orcid.org/0000-0001-8809-4723
Nina R Salama (iD) https://orcid.org/0000-0003-2762-1424

## Decision letter and Author response

Decision letter https://doi.org/10.7554/eLife.52482.sa1
Author response https://doi.org/10.7554/eLife.52482.sa2

## Additional files

### Supplementary files

• Transparent reporting form

### Data availability

The MATLAB scripts used to reconstruct cell surfaces and perform the geometric enrichment analyses are publicly available under a BSD 3-clause license at https://github.com/PrincetonUniversity/shae-cellshape-public and archived at https://doi.org/10.5281/zenodo.3627045 and http://arks.princeton.edu/ark:/88435/dsp01h415pd457.

The following datasets were generated:

| Author(s) | Year | Dataset title | Dataset URL | Database and Identifier |
|---|---|---|---|---|
| Bratton BP, Nguyen J | 2020 | PrincetonUniversity/shae-cellshape-public: Support for SIM data, visualization tools for quality control, and calculating total intensity of individual cells | https://doi.org/10.5281/zenodo.3627045 | Zenodo, 10.5281/zenodo.3627045 |
| Taylor JA, Bratton BP, Sichel SR, Blair KM, Jacobs HM, DeMeester KE, Kuru E, Gray J, Biboy J, VanNieuwenhze MS, Vollmer W, Grimes CL, Shaevitz JW, Salama NR | 2019 | Distinct cytoskeletal proteins define zones of enhanced cell wall synthesis in Helicobacter pylori | http://arks.princeton.edu/ark:/88435/dsp01h415pd457 | DataSpace, 88435/dsp01h415pd457 |

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

## Appendix 1

# Selecting a subset of wild-type cells whose geometry is consistent with the four parameter model of helical-rod shape

We generated a set of simulated helical cells based on the three-dimensional reconstructions of the wild-type population shown in *Figure 2*. Inputs to this simulation are the measured pole-pole cell lengths along the curved centerlines (*Figure 3A and C*, gray); the diameters of the cells (*Figure 3A and D*, purple); the helical pitches of the centerlines (*Figure 3A and E*, pink); and the helical diameters of the centerlines (*Figure 3A and F*, green). To determine the helical pitch and radius from each reconstructed cell, we borrowed heavily from previous algorithms designed to calculate the best fit helix to a set of observations (*Nievergelt, 1997*). We modified these algorithms to accommodate helices longer than one helical repeat and to allow the pitch to be a signed value, with positive pitches corresponding to right-handed helices and negative to left-handed ones. Not all centerlines fit well to a single helical fit as some centerlines have kinks or variable pitch along their long axis. We calculated the relative error of the helical fit as the root mean squared deviation (RMSD) of the error in the fit to the RMSD between two subsequent points along the centerline. This relative error is unitless; we set a threshold value of two for satisfactory fits (*Figure 3—figure supplement 1A* and *Figure 3—video 1*). About one quarter of the centerlines had a good fit to a single helix (402/1137). Wild type *H. pylori* cells have been shown to be right handed (*Yoshiyama and Nakazawa, 2000*). Our algorithm finds that 96% of the cells with satisfactory fits are right handed (387/402). Infrequently (15/402), the algorithm returned a left-handed helix as the best fit. Upon visual inspection, none of these centerlines were globally left-handed and were thus discarded.

From the four calculated 3D shape parameters, we generated synthetic cells to mimic the original wild-type population. Just as we ignored cells whose centerlines were not well fit by a single helix, we also removed cells whose simulated counterpart differed from the real cell reconstruction by more than 10% in surface area, volume, volume of the convex hull, or Euclidean distance from pole to pole. For roughly 20% of the total wild-type population (231/1137), the observed geometry of the cell was consistent with the simple four parameter model (see Materials and methods and *Figure 3—figure supplement 1A and B*). It is not reasonable to look at the distribution of helical parameters for centerlines that do not have satisfactory fits. The distribution of cell lengths, cell diameters, and surface curvatures for the entire population and the population subset are closely matched (*Figure 3—figure supplement 1C–E*), indicating that the subset adequately represents the population. Both wild-type and synthetic cells share a multimodal distribution of Gaussian curvatures with peaks around 5 $\mu m^{-2}$ and between −5 and −10 $\mu m^{-2}$. However, there is a notable difference in the widths and magnitudes of these peaks between the wild-type and corresponding synthetic cells, consistent with the fact that, unlike real cells, the synthetic cell surfaces are perfectly smooth.

Using this subset of simulated cells, we then proceeded to characterize the major and minor helical axes. Because we simulated these cells based on a model of a cylinder wrapped and twisted about a helical axis, they inherently have a natural unwrap helical coordinate system (*Figure 3B* and *Figure 3—figure supplement 1A* and *2*). We chose to set the unwrap angle of the major helical axis to 0° and the minor helical axis to 180° allowing us to measure the relative length of the major to minor helical axes as well as measure the average Gaussian curvature along the helical axes. The average Gaussian curvature at the major axis is 5 ± 1 $\mu m^{-2}$, and the average Gaussian curvature at the minor axis is −11 ± 4 $\mu m^{-2}$. There was substantially more variation in the average curvature at the minor axis than at the major axis (*Figure 3H*).

