## [Decision Letter]

**Acceptance summary:**

The paper provides compelling evidence that the helical shape of *Helicobacter pylori* is established through the complementary action of two cytoskeletal polymers, the bactofilin protein CcmA and the actin-like protein MreB. Because in *Helicobacter*, these scaffolds preferentially accumulate at sites of opposite gaussian curvature (positive CcmA, negative, MreB) their balanced action directs cell wall synthetic complexes so as to counteract each other spatially along the cell axis. The work thus sets an important framework to elucidate how this opposing spatial action of the polymers finally generates a helical shape of the cell wall.

**Decision letter after peer review:**

[Editors’ note: the authors submitted for reconsideration following the decision after peer review. What follows is the decision letter after the first round of review.]

Thank you for submitting your work entitled "Distinct cytoskeletal proteins define zones of enhanced cell wall synthesis in *Helicobacter pylori*" for consideration by *eLife*. Your article has been reviewed by two peer reviewers, and the evaluation has been overseen by a Reviewing Editor and a Senior Editor. The reviewers have opted to remain anonymous.

Our decision has been reached after consultation between the reviewers. Based on these discussions and the individual reviews below, we regret to inform you that your work will not be considered further for publication in *eLife*. As you will see from the individual reviews, both reviewers found interest in the study and acknowledged a real technical tour de force, developing new tools to study PG synthesis in *H. pylori*. However, they also raised a number of significant concerns that preclude publication of the work at this time. Currently, the reviewers do not feel that the work proves that MreB is really fulfilling on negative-curvature a similar role as CcmA is fulfilling on positive curvature. It is also not clear how these polymers respectively influence each other and whether there is a balance between their activity/abundance that dictates whether the cell commits to helicity or rod shape. Last, a controversy about the function of MreB in *H. pylori* PG synthesis remains to be resolved.

Reviewer #2:

Taylor and colleagues have done beautiful work studying the pattern of peptidoglycan synthesis in *Helicobacter pylori* using two distinct clickable probes. The work is thorough and well performed. The authors try to correlate their patterns of labeling with the role of two cytoskeletal proteins in *H. pylori*, MreB and CcmA. This part is also very well done but fails to make the functional connection between the two aspects of this work. My major comments:

1) Regarding the labeling with the clickable probes, the authors do raise the limitations of using alkyne-D-alanine compared to alkyne-MurNAc. I agree with their assessment. Indeed, D-Ala-alk could be incorporated directly in the periplasm through transpeptidase activity although this has not been shown to happen in *H. pylori*. Alternatively, it could be incorporated through the precursors. The authors did not test that by analyzing the precursor pool composition in presence of D-Ala-Alk. If used for precursor synthesis, in that case, it would reflect direct peptidoglycan synthesis and help with the interpretation of the results. In fact, the authors could have used a 3-azido-D-Ala to co-label in the same cells the sugar and the peptide moiety of the PG and compare it in the same cells.

2) The authors discuss the surprising fact that synthesis rarely occurs in zero Gaussian curvature. The way the data was acquired may be misleading. The authors did a pulse of 18 minutes to see where the probe is being incorporated. It is possible that there is simply more effective turnover in zero Gaussian curvature and therefore less label retained. The authors should also do chase experiments where they labeled the entire PG layer and then chase in media without alkyne versions of the MurNAc or D-Ala. If there is really less synthesis in zero Gaussian curvature, then the label should be retained as with the poles. This would also help in interpreting the D-Ala-alk data since its incorporation is the result of synthesis vs. D,D-carboxypeptidase activity. In a chase experiment, it would only reveal where on the cell wall is D,D-carboxypeptidase activity enriched.

3) The authors did a marvelous job studying the role of CcmA on the morphology of *H. pylori* and how it affects the positive Gaussian curvature. They also show that MreB is enriched in negative Gaussian curvature. But the authors do not actually show how the absence of CcmA or MreB actually impacts the pattern of probe incorporation. Does it become completely random on the lateral wall? This is crucial. In particular, it is very important to compare a *ccmA* mutant to the two point *ccmA* mutants. The pattern could be radically different in the absence of *ccmA* compared to the two mutants despite a similar cell shape of the strains.

4) The authors discuss that MreB has been shown to be not essential in *H. pylori* and although they haven't been able to reproduce the data in their strain, the work on MreB showed that MreB was not required for cell shape in *H. pylori*. Furthermore, A22, a known MreB inhibitor, had minimal effect of *H. pylori* morphology. Hence, despite the nice data on MreB localization, there is no evidence in *H. pylori* that MreB is involved in PG synthesis and cell shape regulation. Thus, MreB preferred localization to negative Gaussian curvature might be just a general characteristic of MreB conserved in *H. pylori* but independent of PG synthesis. The authors' model of cell shape regulation and pattern of PG synthesis is not supported by the existing evidence.

5) Please, provide with gels showing the purity of the recombinant MreB and CcmA (and its mutants). Since, the authors have a CcmA-flag version, they should localize it by cryo-EM by antibody staining. It would be very reassuring to observe that in the bacterium CcmA is organized as polymers.

Reviewer #3:

General assessment:

Overall an interesting collection of approaches (genetics, biochemistry, fluorescent microscopy, advanced 3D imaging and analysis) and results, clearly written, which together lead to a slightly disappointing, rather descriptive report. The manuscript presents three main results: the localization of newly inserted PG, that of the cytoskeletal protein MreB and, the most convincing part, the demonstration of the polymerization properties of CcmA and its subcellular accumulation to distinct locations in the cell. All the localization experiments rely on a post-processing analysis previously described and a bit oversell in the first part of the manuscript. The results concerning the PG and MreB localization present strong limitations (described below), that restrains their interpretation, hence their importance and the conclusions and model concerning MreB should be tuned down. The results on CcmA are at the contrary convincing, interesting, and are shedding light on the helical morphogenesis of *H. pylori*. However, the absence of mechanism explaining how this protein controls the curvature limits the overall impact of the findings, and probably its interest for a broad audience.

Summary of substantive concerns:

1) The authors show that +/- Gaussian curvatures (Gc) are different in an helix than in a rod or curved shape (Figure 2-3).

The corresponding sections of the Results are required because this is the basis of all the subsequent image analyses to localize CcmA, MreB and labelled-PG. However it is presented in a way putting forward the obvious and masking its weaknesses. The authors are largely emphasizing the existence of huge +/- Gc as if it was revealed by their approach ("we show that the helical centerline (…) dictate surface curvatures of considerably higher + and - Gaussian curvatures than those present in straight or curved-rod bacteria" in the Abstract). But this is an obvious consequence of the *Helicobacter* cell geometry, whose helicity is not a discovery (Figure 2) and the whole simulation part (Figure 3) is just stating the obvious that curvature depends on the helical diameter and pitch of the helix. A helical object as *H. pylori* is expected to have these strong positive and negative Gc as the application of their method to *Staphylococcus aureus* would predictably show a peak of Gc ~ +4 µm^-2^ because it is a sphere of 1µm diameter. They however do not comment much on the limit of this method when applied to *H. pylori*: indeed this method was originally designed to detect small, non obvious fluctuations of the seemingly straight rod *E. coli*, but when applied here to *H. pylori*, only the major – and again known – +/- Gc due to the rod torsion are really visible. It is unclear to me if *H. pylori* cells are just more "regular" in their curvature or, as I suspect, these huge +/- curvatures due to the cell torsion is somehow masking smaller variations. My guess is that if they were focusing their analysis on the side (meaning excluding the major and minor helical axes) of the cells they may see the kind of fluctuations they observed with *E. coli* cells. That said, I am not utterly convinced of the accuracy of the method to discriminate such small curvatures neither of their relevance, thus I am not suggesting adding such analysis. Especially since the important finding of the study concerns the localization of CcmA to large, unambiguously curved, regions.

Thus, to summarize, albeit useful for the subsequent imaging analysis because it shows that their 3D reconstruction and calculus of +/- GC matches what would be expected for a helical object, this whole part should be rephrase, shorten, less emphasizing the obvious and mainly placed as supplementary.

2) Using clickable D-Ala and an engineered mutant to allow incorporation of a clickable modified PG precursor, the authors labelled PG insertion (Figure 4). The depletion from the pole is unambiguous (Figure 5-6), but the other claims are less convincing. First, the use of 90% Bootstrap confident intervals give the false impression of highly reproducible experiments while the Figure 6—figure supplement 1 reveals the important variability between the 3 replicates and how misleading is this representation. Plotting standard deviations to the mean (not the variations between replicates but between all the pooled values) would certainly be less impressive. Regarding this, please let the reader make their own judgment and avoid using opinionated wording such as "clearly" (subsection “PG synthesis is enriched at both negative Gaussian curvature and the major helical axis area”) or "highly reproducible" (especially in the figure title and when it is not!).

Next, looking at Figure 6—figure supplement 1, it seems that for Gc below -8 and above +8, the variability is so important compared to the relative increase or decrease of concentration that it precludes interpretation. From this, I would agree that there is an enrichment at high positive Gc (~6µm²) but it seems difficult to conclude for negative curvatures: is there an increase for higher negative Gc or is that a depletion at low negative Gc (-2µm²)? The interpretation is even complicated by the fact that the two labeling approaches give different patterns (possibly because of PG maturation) with D-Ala showing virtually no enrichment for both +6 and -2µm² and an enrichment at lowest negative Gc. Importantly, and as commented by the authors in the Discussion, the patterning shows minimal labelling at the side wall (meaning the regions not along the major and minor helical axes, thus, the regions with GC close to 0), while shape maintenance certainly request a higher synthesis rate at this location than along the minor helical axis. Thus, it is likely that in fine the labeling patterns do not properly reflect the CW synthesis pattern, which undermine the results and makes interpretation difficult and speculative.

Altogether, except for the depleted poles (no surprise here considering the literature) and the 20% relative increase of labelling at high Gc (6µm²) (if we ignore the D-Ala result), the other findings are not very solid, and difficult to interpret.

3) Of a lesser importance, but I have been puzzled by the labelling experiment: albeit the strategy is sound and obviously efficient (Figure 5), the authors certainly have a good reason not to use the much easier fluo-DAA labelling (developed by one of the co-authors, E. Kuru). Is it because the approach (modeling, 3D reconstruction) requires working on fixed cells and that implies postponing the labelling step? Or because they wanted to acquire all the data using an identical procedure for fair comparison (and fixed cells being imposed by immuno-fluorescence labelling of the proteins)? This reason for this strategy may be briefly mentioned in the corresponding result section.

4) The authors claim that MreB is enriched at negative Gc (and excluded from poles). Again, and for the same reasons mentioned in point 2, the results are not as strong as the authors are trying to convince us ("highly reproducible" in subsection “MreB is enriched at negative Gaussian curvature” and Figure 7—figure supplement 3 title). Although the reproducibility seems better here than for the PG localization (but again the SD to the mean seems more appropriate than the bootstrap) there is still a large experiment to experiment variability in enrichment for Gc below -7 and above +7 (Figure 7—figure supplement 3). Once the poles, clearly depleted of MreB as reported previously in the literature for other bacteria, and the highly variable areas (for higher +/- Gc) are removed, the trends of MreB accumulation is weak (+/-10%). Surprisingly, another monotonically increasing concentration (this time with increasing Gc) is reported, presenting an enrichment similar to that of MreB (+/-10%; Figure 10—figure supplement 1): the negative control (no Flag) of Figure 10. This control is nonetheless described as showing "negligible curvature preference", suggesting that a +/-10% enrichment is not significant. Thus, MreB is not significantly enriched either.

5) The authors suggest that MreB is playing an active role in CW synthesis. Although this would not surprise most of the MreB crowd, this hypothesis is contradicting previous findings from Waidner et al., 2009, claiming that MreB is not essential and not required for *H. pylori* CW synthesis and shape (in fairness, a point mentioned by the authors). If the authors want to claim a role for MreB in CW control in the present study, they need to address this discrepancy (request strains from PLG, or make depletion strains, or show an effect of A22 on cell shape). As an alternative, considering the weakness of their evidence on this topic, they may prefer to dampen their conclusions, remove MreB from the model, and refocus their study toward CcmA.

6) In their model, MreB increased frequency of localization at negative Gc should translated into increased synthesis along the minor helical axis, which would exactly counteract the maintenance of the helical shape by trying to restore the rod shape. In such a model, while it is easy to imagine that CcmA promote synthesis and deformation along the major helical axis, it is really unclear how the small axis would be maintained and why MreB would fail to restore the rod. Thus, in addition to be lightly supported by their data, this hypothesis does not enlighten our understanding of the building of *H. pylori* helical shape.

[Editors’ note: further revisions were suggested prior to acceptance, as described below.]

Thank you for re-submitting your article "Distinct cytoskeletal proteins define zones of enhanced cell wall synthesis in *Helicobacter pylori*" for consideration by *eLife*. Your article has been reviewed by two peer reviewers, a reviewer previously involved in the original review and a new reviewer. The evaluation has been overseen by Tâm Mignot as a Reviewing Editor and Anna Akhmanova as the Senior Editor. The reviewers have opted to remain anonymous.

In this re-submission, Taylor and colleagues describe a quantitative characterization of the relationship between two cytoskeletal polymers – MreB and CcmA – and the localization of PG metabolic activity and subsequent cell shape in the helical pathogen *Helicobacter pylori*. Their major contributions are (1) development of a quantitative framework for describing and analyzing *H. pylori* cell shape; (2) demonstration of the relationship between local Gaussian curvature of the cell surface and the local incorporation of 2 metabolic probes for PG; (3) the same for the relationship between local Gaussian curvature and localization of cytoskeletal proteins MreB and CcmA; (4) demonstration that MreB is (contrary to a prior report) essential for *H. pylori* growth; (5) characterization of the in vitro polymerization of CcmA; (6) functional characterization of two point mutants of CcmA that fail to polymerize in vitro; and (7) demonstration of the impact of loss of CcmA function on PG probe incorporation and MreB localization. Collectively these results lead to a model wherein two cytoskeletal systems work to spatially modulate cell wall metabolic activity at regions of different Gaussian curvature to elicit helical cell shape.

Overall the authors went to extreme length to answer the concerns raised by the previous reviewers in a convincing way. There are nevertheless key interpretations issues that would need be addressed before the work can be accepted for publication:

1) The paper now shows convincingly that MreB is essential for growth of *H. pylori*, which likely is linked to its function in PG synthesis. The model proposes that the recognition of regions of negative curvature by MreB and positive curvature by CcmA underlies the helicity of the *H. pylori* cell. However, although it is understood that these mechanisms are likely contributing mechanisms, the authors still do not discuss clearly why the action of MreB acting at negative gaussian curvature does not counteract helicity toward Rod restoration. This question is especially important given that a study by Wollrab et al. (bioRxiv 716407; doi:10.1101/716407) questions the MreB negative curvature recognition in *E. coli* cells. If MreB is indeed localizing to -GC regions in the *Helicobacter* cell, how do the authors reconcile this with the Wollrab et al. study?

The authors seem to rule out mutual exclusion mechanisms (ie MreB occupies cylindrical parts that CcmA does not occupy) because in subsection “CcmA localization to positive curvature correlates with cell wall synthesis, CcmA polymerization, and

helical cell shape” they conclude that MreB localization is the same in *∆ccmA* as in WT. However, there appears to be a difference wherein the relative concentration increases more for MreB at negative curvatures in *∆ccmA* than it does in WT and (as the authors actually note later in that section) there is a slight peak of MreB enrichment at ~3 µm^-2^ in the mutant but not in WT. There also appears to be a dip in concentration around 10-15 µm^-2^. Importantly, the differences in MreB distribution look to be of a similar magnitude as the differences in D-Ala-alk incorporation between WT and *∆ccmA* but the D-Ala-alk incorporation pattern is described as different.

2) For the biochemical characterization of CcmA and mutant forms of CcmA, it would be important to note in the text (not just the Materials and methods) that the proteins are purified in denatured form and refolded to induce/monitor polymer formation. This is important because it does somewhat complicate interpretation of the mutants. It could be that the I55A or L110S mutations influence the ability of CcmA to fold properly during renaturation, rather than having a direct effect on polymerization capacity. The punctate cytoplasmic signal by IF could be consistent with either effect. Related to this, the authors say that the mutants "failed to form any higher order structure under any buffer condition tested". What conditions were tested? Did they test whether the mutants that cannot polymerize were soluble when purified?

---

## [Author Response]

[Editors’ note: what follows is the authors’ response to the first round of review.]

We appreciate the reviewers’ support for our work and the concerns that they raise. Both reviewers were concerned with two major issues which we have extensively modified in the main text and discussed below. In short, there seems to be (1) conceptual misunderstanding about the enrichment analysis and how we calculate the relative concentration of signals and (2) disagreement between us and the literature about the essentiality of MreB in *Helicobacter pylori*. In regard to the conceptual misunderstanding, we add to Figure 6 a schematic showing how the arithmetic is performed as well as theoretical example cells showing different relative concentration profiles. In regard to the essentiality of MreB, we have added substantial experimental evidence that at least in our strain, the G27 derivative LSH100, MreB is essential. Additionally, we have reached out to the authors of the study mentioned (Waidner et al., 2009) and they have been unable to provide us with the plasmid they used to delete MreB nor have they been able to regrow any of their MreB deletion strains from their freezer stocks.

Reviewer #2:Taylor and colleagues have done beautiful work studying the pattern of peptidoglycan synthesis in Helicobacter pylori using two distinct clickable probes. The work is thorough and well performed. The authors try to correlate their patterns of labeling with the role of two cytoskeletal proteins in H. pylori, MreB and CcmA. This part is also very well done but fails to make the functional connection between the two aspects of this work. My major comments:1) Regarding the labeling with the clickable probes, the authors do raise the limitations of using alkyne-D-alanine compared to alkyne-MurNAc. I agree with their assessment. Indeed, D-Ala-alk could be incorporated directly in the periplasm through transpeptidase activity although this has not been shown to happen in H. pylori. Alternatively, it could be incorporated through the precursors. The authors did not test that by analyzing the precursor pool composition in presence of D-Ala-Alk. If used for precursor synthesis, in that case, it would reflect direct peptidoglycan synthesis and help with the interpretation of the results. In fact, the authors could have used a 3-azido-D-Ala to co-label in the same cells the sugar and the peptide moiety of the PG and compare it in the same cells.

The reviewer’s concerns about the D-Ala-alk incorporation mechanism do not alter the conclusions we have presented. We have been successful incorporating D-Ala-alk into *H. pylori* PG in vitro with the transpeptidase PBP4 from *Staphylococcus aureus* (unpublished results), showing that this transpeptidation mechanism can be used for incorporation as it is in other bacteria. It is possible that D-Ala-alk can be incorporated through both precursor synthesis and transpeptidation, but unlikely that it would be solely incorporated through precursor synthesis. However, neither case negates the conclusions as presented. With respect to the suggested dual D-Ala, MurNAc labeling, we have attempted labeling with azido-modified probes, but the click reaction results in high levels of non-specific signal, making these probes unusable in our hands. While future work exploring in more detail all possible modes of incorporation of these precursors would be interesting, we do not feel such analysis is necessary for our conclusions.

2) The authors discuss the surprising fact that synthesis rarely occurs in zero Gaussian curvature. The way the data was acquired may be misleading. The authors did a pulse of 18 minutes to see where the probe is being incorporated. It is possible that there is simply more effective turnover in zero Gaussian curvature and therefore less label retained. The authors should also do chase experiments where they labeled the entire PG layer and then chase in media without alkyne versions of the MurNAc or D-Ala. If there is really less synthesis in zero Gaussian curvature, then the label should be retained as with the poles. This would also help in interpreting the D-Ala-alk data since its incorporation is the result of synthesis vs. D,D-carboxypeptidase activity. In a chase experiment, it would only reveal where on the cell wall is D,D-carboxypeptidase activity enriched.

We accidentally omitted the word “relative” in our statement regarding synthesis at the sidewall, which made it sound like we were commenting on low absolute rather than low relative PG incorporation on the sides of the cells. We apologize for the confusion and have changed the text. Please see reviewer 3 item #2 for a detailed discussion of the enrichment plot analysis. In brief, a relative concentration signal at the y=1 line does not imply a lack of synthesis. Rather, it indicates that the area concentration of synthesis is the same at that geometry as the average synthesis everywhere.

In the figure legend for Figures 6, the text "(>1 is enriched; <1 is depleted)”, while correct, may have led the reviewers to overemphasize the importance of when the relative concentration is greater than the average concentration rather than looking holistically at the shape of the relative enrichment curves. We have thus removed this text from the figure legend.

As the reviewer notes, D-Ala-alk and MurNAc-alk signal is a function of incorporation plus turnover. We are interested in PG turnover but assessing turnover rates will require extensive additional work and the development of additional sophisticated image analysis tools and thus is beyond the scope of this work. Given the clarification that there indeed is substantial total synthesis at near-zero GC (though less relative synthesis), we feel that these experiments are not necessary for the conclusions presented in this manuscript. Also, we would like to note that turnover of the D-Ala-alk would be difficult to interpret, due to a combination of both D,D-carboxypeptidase activity and muropeptide turnover.

Wording change:

“The lower relative amount of synthesis at Gaussian curvatures corresponding to the sides of the cell body in comparison to the major and minor axis areas is both unexpected and counterintuitive; it suggests additional mechanisms may be required to maintain helical shape.”

3) The authors did a marvelous job studying the role of CcmA on the morphology of H. pylori and how it affects the positive Gaussian curvature. They also show that MreB is enriched in negative Gaussian curvature. But the authors do not actually show how the absence of CcmA or MreB actually impacts the pattern of probe incorporation. Does it become completely random on the lateral wall? This is crucial. In particular, it is very important to compare a ccmA mutant to the two point ccmA mutants. The pattern could be radically different in the absence of ccmA compared to the two mutants despite a similar cell shape of the strains.

We agree with the reviewer that demonstration of an altered pattern of new cell wall incorporation in the absence of CcmA or MreB would further support our model. Unfortunately, *mreB* is essential in our strain, as further described in our response to Point 4 below and we have not yet been able to engineer a system to deplete *mreB* in our strain. We have, however, undertaken both MreB immunofluorescence and metabolic labeling using the D-Ala-alk probe in a *ccmA* null strain. A caveat of this experiment is that this strain has limited positive Gaussian curvature outside of the cell poles due to its nearly straight morphology. We found that the MreB signal enrichment pattern in *ccmA* cells resembled that in wildtype. In contrast, the cell wall probe enrichment pattern changed in *ccmA* cells compared to wildtype, showing lower relative enrichment at signal Gaussian curvatures representative of the major helical axis (5 µm^-2^) and exclusively mirroring the signal enrichment pattern of MreB, This result appears consistent with retention of MreB driven synthesis and loss of CcmA-dependent synthesis. These data have been added as below and Figure 11 A-B and Figure 11—figure supplement 1.

Wording change:

“To ascertain the impact of deleting *ccmA* on MreB localization and cell wall synthesis patterning, we performed immunostaining and 18-minute D-Ala-alk pulse labeling on Δ*ccmA* cells (JTH6, *amgK murU* Δ*ccmA,* Figure 11A and B, dark pink and dark blue, respectively). […] These data suggest that proper localization of CcmA to the major helical axis may be required for promoting extra cell wall synthesis at the major axis area and patterning helical cell shape.”

4) The authors discuss that MreB has been shown to be not essential in H. pylori and although they haven't been able to reproduce the data in their strain, the work on MreB showed that MreB was not required for cell shape in H. pylori. Furthermore, A22, a known MreB inhibitor, had minimal effect of H. pylori morphology. Hence, despite the nice data on MreB localization, there is no evidence in H. pylori that MreB is involved in PG synthesis and cell shape regulation. Thus, MreB preferred localization to negative Gaussian curvature might be just a general characteristic of MreB conserved in H. pylori but independent of PG synthesis. The authors' model of cell shape regulation and pattern of PG synthesis is not supported by the existing evidence.

In this revision, we demonstrate that in our strain background, MreB is indeed essential. Additionally, *H. pylori* MreB appears to be A22-resistant. Our result that MreB is essential for *H. pylori* growth result is in conflict with the published work of Waidner et al., 2009, who claim to have been able to delete MreB from *H. pylori* strains 26695, KE88-3887, and 1061. It is possible that MreB essentiality is a strain specific effect. However, we have been unable to obtain and verify the deletion of MreB in their strains since upon request, we were informed that they have been unable to regrow these strains from their freezer stocks. Therefore, we cannot comment on whether or not MreB was in fact knocked out in the published strains.

In our revised manuscript we show explicitly that MreB is essential in *H. pylori* strain LSH100 by testing our ability to delete MreB. If the gene is essential, we should only be able to grow up transformants if the cells have a redundant copy of the gene. We transformed both the wild-type strain and an MreB merodiploid strain that has a second copy of MreB at a neutral intergenic locus (McGee locus) with a linear fragment of DNA that would disrupt the MreB locus by inserting a chloramphenicol resistance (cat) cassette into MreB. In the merodiploid strain, transformants were obtained at a frequency of 2.3x10^-4^, but in the wild-type strain we obtained only two colonies (frequency = 6.7x10^-7^). In the merodiploid strain, the *cat* cassette was able to insert into either copy of *mreB*, demonstrating that *mreB* at the native locus can be readily disrupted in the presence of a second copy of *mreB*. The two transformants in the wild-type background contained duplications at the *mreB* locus, resulting in both a disrupted copy of *mreB* and a nearly full length intact copy of *mreB*, further indicating the essentiality of MreB. We furthermore demonstrated that the wild-type strain is readily transformable by disrupting the non-essential single copy gene CcmA with a *cat* cassette (transformation frequency = 2.4x10^-4^). This additional data is discussed in the main text in the Results section “MreB is enriched at negative Gaussian curvature” and Figure 6 and Figure 6—figure supplement 1.

Consistent with Waidner et al., 2009, our strain of *H. pylori* is resistant (no effect on cell growth) to high A22 concentrations (>10ug/mL) (unpublished results).These concentrations have previously been used to isolate *mreB* point mutants that are insensitive to A22 in other species. *H. pylori*’sresistance to A22 does not imply anything about the essentiality of MreB, but it does prevent us from using A22 to disrupt MreB activity.

With respect to whether MreB plausibly contributes to PG synthesis and cell shape regulation in *H. pylori*, we point out that the core rod elongation complex described in *E. coli* (RodA, Pbp2, MreC, MreB) appears to be conserved in *H. pylori*. Work from Ivo Boneca’s group has shown that Pbp2 and MreC are essential and depletion of these proteins leads to cell rounding and cell growth arrest (El Ghachi M et al., 2011, Mol Micro, 82(1):68-86), consistent with observations in *E. coli*. Moreover, their X-ray structure of a complex between *H. pylori* MreC and Pbp2 (Contreras-Martel C et al., 2017, Nat Commun, 8(1):776) reveals conservation of structure in the platform regulatory domain where *E. coli* Pbp2 suppressor mutations that relieve the requirements for MreC and RodZ, but not MreB for cell morphology map (Rohs PDA et al., 2018, PLOS Genetics, 14(10):e1007726). While MreB is not present in several rod-shaped organisms that have a polar growth mode, in this manuscript we show that *H. pylori* has a growth mode like that of *E. coli*; dispersed along the sidewall and largely excluding the poles. This observed cell wall synthesis pattern combined with our findings that MreB is essential and shows a localization of puncta dispersed along the sidewall do not contradict a function for MreB in PG synthesis and cell shape regulation in *H. pylori*.

Wording changes:

“It has been reported that MreB is not essential in *H. pylori* and that treatment with the MreB inhibitor A22 does not alter cell shape (Waidner et al., 2009), though growth inhibition only occurred at concentrations well above those used to select for A22 resistance in other organisms (Gitai et al., 2005; Ouzounov et al., 2016; Srivastava et al., 2007; Wu et al., 2011). […]While we requested the previously published *mreB* mutant strains (Waidner et al., 2009), they could not be revived from frozen stocks. We thus conclude that MreB is essential in LSH100 and perhaps all *H. pylori* strains.”

5) Please, provide with gels showing the purity of the recombinant MreB and CcmA (and its mutants). Since, the authors have a CcmA-flag version, they should localize it by cryo-EM by antibody staining. It would be very reassuring to observe that in the bacterium CcmA is organized as polymers.

Please note that we did not report purifying or studying recombinant MreB. However, we have happily included gels showing the purity of WT and mutant CcmA (Figure 8—figure supplement 1D). We have tried immunogold labeling of thin sections with traditional TEM sample prep methods to detect CcmA-FLAG but unfortunately did not see any distinctive labeling. Unfortunately, with thin sections and only a small surface of accessible antigen, this becomes a “needle in a haystack” problem. Electron cryotomography is not trivial to perform or interpret. As positive identification of putative densities will be challenging, success in this arena will merit a separate manuscript.

Reviewer #3:[…]Summary of substantive concerns:1) The authors show that +/- Gaussian curvatures (Gc) are different in an helix than in a rod or curved shape (Figure 2-3).The corresponding sections of the results are required because this is the basis of all the subsequent image analyses to localize CcmA, MreB and labelled-PG. However it is presented in a way putting forward the obvious and masking its weaknesses. The authors are largely emphasizing the existence of huge +/- Gc as if it was revealed by their approach ("we show that the helical centerline (…) dictate surface curvatures of considerably higher + and - Gaussian curvatures than those present in straight or curved-rod bacteria" in the Abstract). But this is an obvious consequence of the Helicobacter cell geometry, whose helicity is not a discovery (Figure 2) and the whole simulation part (Figure 3) is just stating the obvious that curvature depends on the helical diameter and pitch of the helix. A helical object as H. pylori is expected to have these strong positive and negative Gc as the application of their method to Staphylococcus aureus would predictably show a peak of Gc ~ +4 µm^-2^ because it is a sphere of 1µm diameter. They however do not comment much on the limit of this method when applied to H. pylori: indeed this method was originally designed to detect small, non obvious fluctuations of the seemingly straight rod *E. coli*, but when applied here to pylori, only the major – and again known – +/- Gc due to the rod torsion are really visible. It is unclear to me if H. pylori cells are just more "regular" in their curvature or, as I suspect, these huge +/- curvatures due to the cell torsion is somehow masking smaller variations. My guess is that if they were focusing their analysis on the side (meaning excluding the major and minor helical axes) of the cells they may see the kind of fluctuations they observed with *E. coli* cells. That said, I am not utterly convinced of the accuracy of the method to discriminate such small curvatures neither of their relevance, thus I am not suggesting adding such analysis. Especially since the important finding of the study concerns the localization of CcmA to large, unambiguously curved, regions.Thus, to summarize, albeit useful for the subsequent imaging analysis because it shows that their 3D reconstruction and calculus of +/- GC matches what would be expected for a helical object, this whole part should be rephrase, shorten, less emphasizing the obvious and mainly placed as supplementary.

We are unclear exactly what the reviewer is suggesting in this comment, and precisely what the reviewer feels is intuitive about the geometry of the surface of a helical rod. Here we try to summarize the objections that we think the reviewer is making: (a) The observation that *H. pylori* has a helical-rod morphology is not new; (b) the distribution of surface curvatures depends on the shape of the object; (c) the goal of our 3D reconstruction method is detect non-obvious deviations of a surface from an idealized geometry; (d) there is a side/surface to helical rod cells which is neither major nor minor helical axis and this side/surface may be similar to non-helical rod-like cells; (e) Action point: edit the text to be shorter and move it mainly to the supplement. We apologize if these were not the points the reviewer was attempting to make but will address them each in turn.

We respectfully disagree with reviewer #3 regarding these comments. The geometric points made herein are the first 3D description of *H. pylori* shape; facilitate clarity in a field struggling with using accurate spatial language; and are essential for interpretation of all the subsequent curvature enrichment plots, on which the conclusions of this manuscript depend. The computational simulations and validation were already included as supplementary figures; the corresponding section in the main text is one paragraph; and we already included the finer details of the computational simulations as an appendix.

a) We absolutely agree with the reviewer that there have been many previous reports of *H. pylori* as a helical-rod bacterium. Where this manuscript pushes that qualitative description forward is by quantitatively describing the 3D shape of *H. pylori* both in terms of surface curvatures and helical-rod parameters. Even our previous state of the art estimates of *H. pylori’s* 3D shape came from 2D images (Sycuro et al., 2010; Martinez et al., 2016), which has various known issues including a length-dependent bias in estimating pitch. As requested, we have updated the Abstract and text to clarify that the helicity of *H. pylori* and some amount of intuitable geometric properties are not a ‘discovery’ and to clarify the importance of the 3D population parameter measurements.

Wording changes:

“The helical centerline pitch and radius of wild-type *H. pylori* cells dictate surface curvatures of considerably higher positive and negative Gaussian curvatures than those present in straight- or curved-rod *H. pylori*.”

“Display of the Gaussian curvature, which is the product of the two principal curvatures, at each point on the meshwork shows the distinct curvatures on opposite sides of helical cells (Figure 1B).”

“Furthermore, prior shape parameter characterizations of *H. pylori* have been performed using 2D images (Martínez et al., 2016; Sycuro et al., 2013, 2012, 2010; Yang et al., 2019); measurement of pitch and helical radius from 2D images is subject to systematic errors for short cells (approximately <1.5 helical turns) depending on their orientation on the coverslip. Therefore, we also wished to determine *H. pylori* population shape parameters from our 3D dataset.”

b) We agree with the reviewer that the distribution of surface curvatures for different shapes are different, and that they would also be different for different sizes. As the reviewer mentions, this is a key point for our enrichment analysis. Before we ran the simulations, we were unable to intuit which helical parameters would be most important for changing the distribution of surface curvatures. It is possible that the reviewer’s intuitive sense of geometry is keener than ours. Following the reviewer’s suggestion, this will remain a supplementary figure.

There appears to be additional confusion about the content of Figure 3. While the supplements (Figure 3—figure supplement 3 and 4) do indeed discuss the effect on curvature of modulating the “core” helical shape parameters (within the biologically-relevant +/- 1.5 standard deviations of the LSH100 population average), this is not presented within the main Figure 3. In fact, as Figure 3 provides the first 3D measurement of the parameters of helical cell shape for *H. pylori*, it is an important contribution to the field. Figure 3C-F (3D length, diameter, helical pitch, and helical diameter) are, as stated in the figure legend, measurements from actual WT cell surfaces. Due to technical limitations of fitting the major and minor axis to actual cell surfaces with imperfections (i.e. *H. pylori* cells are not perfect helices), Figure 3G and H (major:minor axis length and Gaussian curvature at the surface helical axes) are calculated from “perfect” simulated cells, which we demonstrated accurately reflect the full population (Figure 3—figure supplement 1). As these are experimentally observed population measurements, we do not agree that these measurements are obvious and can be intuited. Furthermore, the connection between surface landmarks (major and minor helical axes) and Gaussian curvature are crucial for interpreting our curvature enrichment results in the context of cellular features. In supplementary figures (but not in main figures), we discuss the contribution of the core helical parameters to both the distribution of Gaussian side curvature and to the major:minor axis length ratio. While the reviewer may be able to intuit the contribution of modulating these parameters to the side curvature distribution and the major:minor axis length ratio (indeed being fundamentally mathematical in nature one expects a well-formed geometric intuition to predict general conclusions), we disagree that these conclusions are obvious and without value. We have been unable to find any publication that explores how modulating a helical rod shape would impact sidewall curvature or the major:minor axis ratio. Moreover, the helical shape field is hindered by imprecise or completely flawed geometric descriptions (please see for example Stahl et al., 2016, in which a Δ1228 strain is described as having “decreased pitch”, when one can see clearly from their Figure 1B that the authors actually meant that the strain has increased pitch). It is our intention that these helical shape parameter discussions will also help the field to use precise and correct geometric language.

c) We disagree with the reviewer’s assertion that our “method was originally designed to detect small, non obvious fluctuations of the seemingly straight rod *E. coli*, but when applied here to *H. pylori*, only the major – and again known – +/- Gc due to the rod torsion are really visible.” In fact, the first paper we published that used this method (Ursell et al., 2014) measured the 3D shape of both rod-shaped and sinusoidal cells with considerably more curvature than unperturbed *E. coli*. We have also shown that our 3D surface reconstruction achieves 30-nm accuracy in all three dimensions (Nguyen, Methods in Molecular Biology 2016) and thus we accurately probe Gaussian curvatures on small and large scales. Our 3D shape reconstruction method describes the shape of the surface, including small surface fluctuations. For example, most of the positive Gaussian curvature values along the sidewall occur near the major axis. But if there is an area of positive curvature at the minor axis, this surface is not discarded but rather contributes to the data at that curvature point.

d) We have provided an additional supplemental figure to help provide some graphical examples of what simulated additional signal at different types of geometric localizations would look like with our analysis platform (Figure 6—figure supplement 1; see more extensive discussion in our response to point 2). Because the Gaussian curvature goes from positive at the major helical axis to negative at the minor, there must be a region that has zero Gaussian curvature. The region of the surface which has zero Gaussian curvature is a very small ribbon which wraps around the cell (shown in white in Figure 1B). For the reviewer’s point about the ‘side’ of the cell which is neither major nor minor axis, we would like to refer the reviewer to Figure 6—figure supplement 1C, the row labeled ‘enriched at zero’. We think the reviewer may have an inaccurate mental picture that has much of the sidewall as zero Gaussian curvature and discontinuities at the major and minor axis. Rather, a more realistic picture is a smoothly varying Gaussian curvature as one goes around the helical unwrap theta dimension (Figure 3H).

e) Regarding the reviewer’s request that we move the majority of this discussion to the supplement, we already presented the finer details of some of the simulations as a supplemental appendix and presented most of these figure panels as supplemental figures in the original manuscript, leaving only measurements of actual *H. pylori* population parameters as a main figure.

2) Using clickable D-Ala and an engineered mutant to allow incorporation of a clickable modified PG precursor, the authors labelled PG insertion (Figure 4). The depletion from the pole is unambiguous (Figure 5-6), but the other claims are less convincing. First, the use of 90% Bootstrap confident intervals give the false impression of highly reproducible experiments while the Figure 6—figure supplement 1 reveals the important variability between the 3 replicates and how misleading is this representation. Plotting standard deviations to the mean (not the variations between replicates but between all the pooled values) would certainly be less impressive. Regarding this, please let the reader makes is own judgment and avoid using opinionated wording such as "clearly" (subsection “PG synthesis is enriched at both negative Gaussian curvature and the major helical axis area”) or "highly reproducible" (especially in the figure titles and when it is not!).Next, looking at Figure 6—figure supplement 1, it seems that for Gc below -8 and above +8, the variability is so important compared to the relative increase or decrease of concentration that it precludes interpretation. From this, I would agree that there is an enrichment at high positive Gc (~6µm²) but it seems difficult to conclude for negative curvatures: is there an increase for higher negative Gc or is that a depletion at low negative Gc (-2µm²)? The interpretation is even complicated by the fact that the two labeling approaches give different patterns (possibly because of PG maturation) with D-Ala showing virtually no enrichment for both +6 and -2µm² and an enrichment at lowest negative Gc. Importantly, and as commented by the authors in the discussion, the patterning shows minimal labelling at the side wall (meaning the regions not along the major and minor helical axes, thus, the regions with GC close to 0), while shape maintenance certainly request a higher synthesis rate at this location than along the minor helical axis. Thus, it is likely that in fine the labeling patterns do not properly reflect the CW synthesis pattern, which undermine the results and makes interpretation difficult and speculative.Altogether, except for the depleted poles (no surprise here considering the literature) and the 20% relative increase of labelling at high Gc (6µm²) (if we ignore the D-Ala result), the other findings are not very solid, and difficult to interpret.

From these comments (as well as other comments throughout from both reviewer 2 and 3), it is clear that there has been a fundamental misunderstanding of the relative concentration plots we present throughout the manuscript. We are grateful to the reviewers for revealing this issue and we have striven to modify the manuscript text to allow the reader to understand these curvature enrichment plots. We have added an additional supplemental figure (Figure 6—figure supplement 1) and modified text in various places to try to help clarify. In the response below, we discuss the curvature enrichment plot, which is relevant for both reviewers as well as the general audience, and then we address reviewer #3’s specific concerns about statistical analysis and interpretation of error bars.

The first step in generating the curvature enrichment profile, or a plot of the relative concentration as a function of Gaussian curvature, is to collect the raw fluorescence intensity data at each point on the reconstructed cell surface. We add together all the intensities that came from a particular curvature to generate a raw signal profile for a single cell. We then normalize these raw intensities by setting the average surface concentration to 1 for each cell. Numerically, this is performed by dividing both by the total signal in the cell and by the fractional surface area contributed to that cell by each Gaussian curvature bin. This division results in the final single cell enrichment profile: the concentration of the signal relative to a uniform distribution of the same total amount of signal. We have added a graphical depiction of these calculations to Figure 6 (Figure 6A). To generate the sidewall only plots, we perform the normalization after we computationally remove the poles. The resulting “without poles” relative concentration plot is different than the “with poles” plot due to changes in all three components: the raw signal, the total signal, and the fractional surface area. After the enrichment plot has been calculated for each cell, it is averaged across all the cells. The bootstrap confidence intervals of that mean value are calculated by bootstrap sampling the single cell normalized enrichment profiles.

Due to the normalization process, the relative concentration at each Gaussian curvature value is intimately related to the intensities at all of the other Gaussian curvatures. To clarify this point, we have provided eight different noise-free example cells with added signal intensity at different geometric distributions in Figure 6—figure supplement 1 and a paragraph in the main text discussing this figure. The first two have perfectly uniform labeling, one with a low concentration and one with 25% higher signal at all curvature points. The fourth column shows that these have identical relative concentration profiles exactly at the y=1 line.

Now we simply add 25% extra intensity to regions that are not the poles (pole exclusion, pink lines) without removing any signal from the rest of the cell. The addition of signal means that the average intensity is greater. After normalization, the relative concentration at the sidewall is above average and thus above 1. Even though the intensity at the poles in the raw signal is the same, because the average concentration increased, the relative concentration at the poles is below the *y*=1 line.

Consider next a set of signal increases at particular parts of the cell. The next three rows show the effect of increasing the intensity at the minor axis, zero Gaussian curvature, and the major axis by 25%. Along the minor axis (gold lines), there is very little total surface area. Thus, a 25% increase raises the average value only slightly and the relative concentration at the ‘excluded’ regions is only slightly below the y=1 line. Because of the greater proportion of sidewall represented, a 25% increase at zero Gaussian curvature results in a greater total signal increase and therefore “sinks” the other side of the curve more. This effect is even stronger for the 25% increase at the major axis.

To further illustrate the interrelated nature of the enrichment plots with the simulations, we added 25% signal to the major axis (blue lines), 25% signal at negative Gaussian curvature with a monotonically declining profile (red lines), and 25% signal both at the major axis and at negative Gaussian curvature with a monotonically declining profile (purple lines). By adding 25% extra signal at the major axis, the average signal increases, which then “sinks” the rest of the curve and influences how far the above the y=1 average signal line the plot sits at negative Gaussian curvature, even though there is still 25% extra signal added at negative Gaussian curvature (compare the monotonic decline alone (red) to the combination of monotonic decline and major axis (purple)).

With these points in mind, we return to our actual data for a more complicated example of the interrelated nature of this data. Please refer to Figure 6, in which we present in panel B the enrichment plots with poles included and in panel C the same cells, but with the poles removed before calculating the average surface intensity and rescaling the data. In B, the low values at the poles decrease the average compared to the average in C with the poles removed. As a result, in B the relative average intensity at the Gaussian curvatures corresponding to the non-pole sidewall increases compared to in C. Note, for example, that the “trough” for the 18-minute MurNAc-alk labeling now just barely dips below the average value and the “enrichment” at negative axis is at 1.2 with poles vs. at less than 1.1 without poles. For this reason, it is inappropriate to describe enrichments in terms of percentage of enrichment. Rather, the curve must be considered holistically.

When reviewer 3 comments that enrichment “seems difficult to conclude for negative curvatures: is there an increase for higher negative Gc or is that a depletion at low negative Gc (-2µm²)?”, the interrelated nature of the relative curvature enrichment has been missed. The answer to the question posed is “yes” – there is a relative increase for higher negative GC and “yes” – there is a relative depletion at near-zero GC. By definition, there is no way for this to be an either/or situation. What we can say is that there is above average signal intensity at negative GC and also above average signal intensity at positive GC, which dictates that the signal elsewhere (near-zero GC) is below average. The final row of our simulated supplement (purple line) shows a more complicated situation by augmenting the baseline with both the monotonic decline and enrichment at the major axis profiles. While this is, in some sense, the sum of the two rows above it (25% monotonic decline in red combined with 25% major axis enrichment in light blue), the relative weights of the two in the enrichment plot of the final row (purple line) are different because of the fraction of total surface area that each contributes.

We now return to the scientific conclusions we derived from our observed enrichment profiles. As stated by the reviewer: “Importantly, and as commented by the authors in the Discussion, the patterning shows minimal labelling at the side wall (meaning the regions not along the major and minor helical axes, thus, the regions with GC close to 0), while shape maintenance certainly request a higher synthesis rate at this location than along the minor helical axis. Thus, it is likely that in fine the labeling patterns do not properly reflect the CW synthesis pattern, which undermine the results and makes interpretation difficult and speculative.”

We agree that our statement in the Discussion is misleading due to the accidental omission of the word “relative” and apologize for the confusion this caused. We have corrected this omission. The reviewer statement that “shape maintenance certainly request a higher synthesis rate at this location [near-zero GC] than along the minor helical axis”, however, is unjustified. Shape maintenance does require that the total amount of surface area near zero GC be greater than that along the minor helical axis, which we show in the distribution of surface curvatures (Figure 2). However, this does not necessarily require that the rate of synthesis be greater and the total steady state amount of PG could include contributions from wall removal or recycling, synthesis of wall with differential material properties, localized stress generation, etc. The difference between the D-Ala-alk and MurNAc-alk curves, in fact, supports the idea that other mechanisms also contribute to helical shape maintenance. Even if shape maintenance occurs through synthesis of uniform material, it would not necessarily require differential rates of synthesis per unit area. One could very reasonably propose a model in which synthesis rates per unit area are uniform along the entire cell surface which results in more total material inserted where there is already more surface area. We find no support for the reviewer’s contention that “Thus, it is likely that in fine the labeling patterns do not properly reflect the CW synthesis pattern”. We are also somewhat confused by the reviewer’s seemingly contradictory statement about this in their point 3, “the [labeling] strategy is sound and obviously efficient (Figure 5)”, but again apologize if the misleading statement led to a mistaken interpretation of our data.

Reviewer 3 mentions “variability is so important compared to the relative increase or decrease of concentration that it precludes interpretation” when looking at the data from independent biological replicates. For the reasons detailed above, we never try to comment on a “percent enrichment” at various cellular locations. What we do comment on is (1) if there is a relative enrichment of signal and (2) at what GC this enrichment occurs. Even though there is some variation in the amount of enrichment seen, (1) enrichment of signal is still present at the minor and major axis areas (because as explained above, it matters where the curve is increasing/decreasing rather than where the line crosses the *y*=1 line) and (2) the biological replicate curves have “peaks” and “troughs” at consistent GC values. While great attention was paid to perform these experiments in as identical a manner as possible, these biological replicates were performed on separate days with separate batches of liquid cultures. We use these replicates to convince ourselves that the pattern of the enrichment (position and existence of peaks or overall trends) was not merely an artifact. Regarding the increased variability at the GC extremes, this is an inherent component of the analysis: there are fewer points in each cell with these values, so the measurements are noisier. This is reflected in the size of the confidence interval. We have taken the reviewer’s suggestion and altered the titles of Figure 6—figure supplement 2, Figure 7—figure supplement 4, and Figure 10—figure supplement 2, but contend that the existence and location of peaks and troughs in the enrichment profile are robust across the biological replicates.

Regarding error bars and statistical analysis of variability, we are confused by what the reviewer requests that we plot by the description “standard deviations to the mean.” We assume here that they are requesting a plot of the standard error of the mean as calculated by the standard deviation of the data divided by the square root of the number of samples. Our response is based on that assumption and we apologize if the reviewer had intended something else. Choosing appropriate error bars is discussed in detail in “Error bars in experimental biology” 10.1083/jcb.200611141. We would be happy to consider alternate ways of presenting the data and, historically, we have debated amongst ourselves and with other reviewers about the most appropriate way of displaying the data. We have chosen to represent in the main text error bars that reflect the precision of our measurement, and not the variability across different cells. We additionally provide supplemental figures to allow the reader to assess the variability in these average values across different biological replicates. Two reasons why we have chosen to avoid the ‘standard error of the mean’ approach and have instead chosen the bootstrap CI is (1) the SEM technique ignores within-cell correlations in the data, and (2) the SEM method assumes the data are normally distributed, an assumption that is not valid for our data. For a further discussion of CI, SEM and STD, please see “Misuse of standard error of the mean (sem) when reporting variability of a sample. A critical evaluation of four anaesthesia journals” https://doi.org/10.1093/bja/aeg087. That being said, if the reviewer is requesting a standard error of the mean plot, they are incorrect that “Plotting standard deviations to the mean (not the variations between replicates but between all the pooled values) would certainly be less impressive.” Because of the large number of cells in each sample, such SEM bars are thinner than the solid lines.

Wording changes:

“As a tool to facilitate understanding and interpretation of these relative enrichment plots, we generated a synthetic cell surface with the same geometric properties as the average wild-type cell (Figure 3), applied a variety of example intensity distributions, and generated curvature enrichment plots. […] The key features of interest are the overall increases, decreases, and peaks in the curves, along with the curvatures at which these occur.”

Figure 6 legend “(A) The calculation of relative concentration for a specific probe involves two steps of normalization. […] In the experimental data presented in the main text, the single cell relative concentration profile is averaged over hundreds of cells, each with their own unique geometry.”

“Biological replicates are shown in Figure 6—figure supplement 2.”

“Figure 6—figure supplement 2. Curvature enrichment analysis of biological replicates of MurNAc-alk-, D-Ala-alk-, and mock-labeling.”

“Biological replicates are shown in Figure 7—figure supplement 4.”

“Figure 7—figure supplement 4. Curvature enrichment analysis of biological replicates of MreB.”

“Biological replicates are shown in Figure 10—figure supplement 2A.”

“Figure 10—figure supplement 2. Curvature enrichment analysis of biological replicates of CcmA-FLAG.”

3) Of a lesser importance, but I have been puzzled by the labelling experiment: albeit the strategy is sound and obviously efficient (Figure 5), the authors certainly have a good reason not to use the much easier fluo-DAA labelling (developed by one of the co-authors, E. Kuru). Is it because the approach (modeling, 3D reconstruction) requires working on fixed cells and that implies postponing the labelling step? Or because they wanted to acquire all the data using an identical procedure for fair comparison (and fixed cells being imposed by immuno-fluorescence labelling of the proteins)? This reason for this strategy may be briefly mentioned in the corresponding result section.

Our preliminary experiments began with using various FDAAs with great success. However, we decided to use the D-Ala-alk for two reasons. (1) We can use brighter and more photostable fluorophores, which are superior for SIM and allow for more rapid image acquisition which adds up over the >1500 SIM images used to generate this manuscript. (2) D-Ala-alk is far more cost effective and easy to obtain in large quantities commercially, allowing us to use the same probe for both the peptidoglycan mass spectrometry experiments and microscopy experiments. We do not find this information to be sufficiently relevant to merit mention in the text of the manuscript.

4) The authors claim that MreB is enriched at negative Gc (and excluded from poles). Again, and for the same reasons mentioned in point 2, the results are not as strong as the authors are trying to convince us ("highly reproducible" in subsection “MreB is enriched at negative Gaussian curvature” and Figure 7—figure supplement 3 title). Although the reproducibility seems better here than for the PG localization (but again the SD to the mean seems more appropriate than the bootstrap) there is still a large experiment to experiment variability in enrichment for Gc below -7 and above +7 (Figure 7—figure supplement 3). Once the poles, clearly depleted of MreB as reported previously in the literature for other bacteria, and the highly variable areas (for higher +/- Gc) are removed, the trends of MreB accumulation is weak (+/-10%). Surprisingly, another monotonically increasing concentration (this time with increasing Gc) is reported, presenting an enrichment similar to that of MreB (+/-10%; Figure 10—figure supplement 1): the negative control (no Flag) of Figure 10. This control is nonetheless described as showing "negligible curvature preference", suggesting that a +/-10% enrichment is not significant. Thus, MreB is not significantly enriched either.

We refer to the thorough discussion of reviewer 3’s item #2 to address the concerns about the credibility of the enrichment of MreB, the inapplicability of interpreting the data as +/- 10%, and the variation at extreme GC values. We realize that our phrasing in the manuscript has once again caused confusion and have modified the language to clarify. The negative control has negligible curvature preference relative to the corresponding signal of interest (PG synthesis, CcmA, MreB). We acknowledge that there is actually a “characteristic” enrichment pattern for each negative control. This is most clear for the anti-CcmA preimmune serum, and it is unsurprising that antibodies in the serum might recognize something with a preference for specific GC values. In fact, this signal even contributes to the overall anti-CcmA curvature enrichment plot, as seen by the increase at negative GC that is not seen in the anti-FLAG CcmA-FLAG plots nor in the anti-FLAG WT control. In fact, if anything, the anti-FLAG and preimmune serum plots vs. the anti-FLAG plots strengthen our conclusions about the relative enrichment of PG synthesis signal at negative GC, as one can see the actual contribution of added signal at negative GC with the presence of an enriched signal at the major axis area to the shape of the enrichment plot.

The reviewer’s comments about comparing the negative control to the actual signal led us to further quantify the total signal in different conditions. To that end, we now provide histograms showing the amount of signal for both the negative control and the signal of interest (Figure 6—figure supplement 2B, Figure 7—figure supplement 4B, Figure 10—figure supplement 2B and 3B, and Figure 11—figure supplement 1C). We also have added text to discuss the negative controls in more detail.

Wording changes:

“The mock labeling control showed minimal curvature bias and is on average 3.6% of the D-ala-alk signal and 4.5% of the MurNAc-alk signal (Figure 6B and C, gray and Figure 6—figure supplement 2B).”

“Preimmune serum signal was 36.4% of the MreB signal (Figure 7—figure supplement 4B), but did not show a curvature preference (Figure 7E, gray)”

“Immunostaining with CcmA preimmune serum showed some background signal in the interior of wild-type and mutant cells (Figure 9 and Figure 9—figure supplement 2).”

“The wild-type (no FLAG) negative control was 28.9% of the CcmA-FLAG signal (Figure 10—figure supplement 2B). While the negative control showed a small peak at 5 µm^-2^, the magnitude of the CcmA-FLAG peak was far greater (Figure 10A and supplement 1A).”

“Preimmune signal was 33.0% of the anti-CcmA signal in wild-type (Figure 10—figure supplement 3).”

“Preimmune signal was 50.6% and 26.7% of anti-CcmA signal in I55A and L110S, respectively (Figure 10—figure supplement 3B).”

5) The authors suggest that MreB is playing an active role in CW synthesis. Although this would not surprise most of the MreB crowd, this hypothesis is contradicting previous findings from Waidner et al., 2009, claiming that MreB is not essential and not required for H. pylori CW synthesis and shape (in fairness, a point mentioned by the authors). If the authors want to claim a role for MreB in CW control in the present study, they need to address this discrepancy (request strains from PLG, or make depletion strains, or show an effect of A22 on cell shape). As an alternative, considering the weakness of their evidence on this topic, they may prefer to dampen their conclusions, remove MreB from the model, and refocus their study toward CcmA.

We have addressed these comments in our response to reviewer #2, item 4.

6) In their model, MreB increased frequency of localization at negative Gc should translated into increased synthesis along the minor helical axis, which would exactly counteract the maintenance of the helical shape by trying to restore the rod shape. In such a model, while it is easy to imagine that CcmA promote synthesis and deformation along the major helical axis, it is really unclear how the small axis would be maintained and why MreB would fail to restore the rod. Thus, in addition to be lightly supported by their data, this hypothesis does not enlighten our understanding of the building of H. pylori helical shape.

We appreciate that the reviewer understands our model that the MreB-based enrichment of synthesis at the minor axis area would, excluding other contributors, result in loss of helical shape and formation of a straight rod, as we illustrate in the top portion of the “helical shape maintenance” model in Figure 11C. We refer the reviewer to the bottom portion of the “helical shape maintenance” model for our proposed model that CcmA-promoted PG synthesis at the major helical axis counterbalances the MreB-patterned PG synthesis at negative Gaussian curvature. We do not claim that synthesis patterning is the only possible contributing mechanism, but we do propose and support with our data that MreB- and CcmA-biased PG insertion is one of the contributing mechanisms, and is in fact to date the best-described mechanism for *H. pylori* cell shape maintenance and the first mechanism showing where changes to the cell wall necessary for helical shape are occurring. Moreover, we present a novel mechanism for curvature maintenance; the two other proposed mechanisms for curvature maintenance demonstrate that a cell-spanning skeletal filament (CreS in *Caulobacter crescentus* and CrvA in *Vibrio cholerae*) on the minor axis/negative GC yields increased synthesis on the opposite face of the cell. Here, we show a new mechanism for promoting curvature: the cytoskeletal protein CcmA forms puncta at the major axis/positive GC and promotes synthesis at that face of the cell. As such, this paper is a marked step forward for understanding strategies for how cells can achieve a given shape.

[Editors’ note: what follows is the authors’ response to the second round of review.]

Overall the authors went to extreme length to answer the concerns raised by the previous reviewers in a convincing way. There are nevertheless key interpretations issues that would need be addressed before the work can be accepted for publication:1) The paper now shows convincingly that MreB is essential for growth of H. pylori, which likely is linked to its function in PG synthesis. The model proposes that the recognition of regions of negative curvature by MreB and positive curvature by CcmA underlies the helicity of the H. pylori cell. However, although it is understood that these mechanisms are likely contributing mechanisms, the authors still do not discuss clearly why the action of MreB acting at negative gaussian curvature does not counteract helicity toward Rod restoration. This question is especially important given that a study by Wollrab et al. (bioRxiv 716407; doi:10.1101/716407) questions the MreB negative curvature recognition in E. coli cells. If MreB is indeed localizing to -GC regions in the Helicobacter cell, how do the authors reconcile this with the Wollrab et al. study?

Thank you for pointing out this interesting preprint. The study by Wollrab et al. has two major observations that conflict with current wording and/or observations in our manuscript.

The first observation relates to the role of MreB in localizing the Rod PG synthesis complex. Previous published data highlights roles for MreB in *E. coli* and *B. subtilis* in promoting circumferential motion of the Rod complex and relative enrichment at negative Gaussian curvature. There is much debate in the field about the mechanisms driving the curvature localization bias of MreB and the Rod complex. Recent literature has invoked curvature-based constraints on MreB assembly dynamics or membrane interactions (Wang and Wingreen, 2013, *Biophys J;* Quint et al., 2016, *Biophys J*; Wong et al., 2019, *eLife*). The Wollrab study provides convincing data that in *E. coli* the PG transpeptidase binds PBP2 binds some cell envelope feature (likely cell wall) and a subset of PBP2 binding events show simultaneous or subsequent recruitment of MreB and other Rod complex members resulting in persistent circumferential motion. At several points in our manuscript we summarized prior literature as suggesting that MreB recruits side wall PG synthesis complexes. Given these new data, it seems more likely that rather than initiating Rod complex assembly, MreB plays roles in further assembly or activation of the Rod complex and in promoting or enhancing the persistent circumferential motion of Rod complexes. To accommodate this new understanding we have changed “help recruit” to “help maintain”. At other points in the manuscript we use the wording “helps direct”, which we think remains consistent with the available data. We have changed “recruits” to “helps to pattern” and changed “MreB-driven” to “MreB-associated”. To clarify that static measurements of enrichment at negative Gaussian curvature are not necessarily in conflict with observations that MreB tends to move circumferentially about the cell, we added “which may result from circumferential motion about the cell.”

The second observation relates to MreB enrichment at negative Gaussian curvature. The Wollrab study concludes that MreB does not have a curvature localization bias because curvature correlations are diminished if they remove cell poles (which have very high curvatures) and center line cell bends from their analysis. In our study, we retain a curvature localization bias for MreB even when we exclude the cell poles. Given the significant contribution of the poles to curvature, as discussed in the Wollrab study, we updated Figure 11 and supplements and now present the sidewall only plots in the main figure and moved the whole cell surface plots to the supplement. The Wollrab manuscript also raises concerns about bending induced by placing cells on an agarose pad and the influence this could have on MreB curvature enrichment. As we fix cells in liquid culture and then gently deposit cells on coverslips after fixation, these concerns do not apply to our experiments. Even with excluding the poles, we still see an enrichment of MreB at negative curvature in both wild-type and Δ*ccmA* cells. The different observations by our two studies could result from the fact that unlike *E. coli, H. pylori* sidewalls retain a much larger range of curvatures in wild-type cells and/or the fact that these two studies use different methods to measure curvature. The Wollrab study infers surface curvature from 2D cell outline contour curvature measurements while our study used 3D reconstructions of cell surfaces to measure curvature. Given these major differences in the experimental systems and analysis methods, we do not feel it is appropriate to directly comment on the Wollrab observations.

We do agree with the comment that the action of MreB at negative Gaussian curvature would be expected to counteract helicity toward rod restoration if there are not other contributors to cell wall patterning. As already discussed in the manuscript (and depicted the model in Figure 11), we suggest that CcmA promotion of synthesis at positive curvature may be one mechanism that counters the hypothesized rod-restoration activity of MreB. As pointed out in the paragraph, a major outstanding question is how CcmA or MreB promotes cell wall synthesis in *H. pylori* and will require further experiments beyond the scope of this paper. We also already specifically call out unexpected and counterintuitive implications of our finding of higher relative synthesis at both negative and positive curvatures. As indicated in this paragraph we think it likely that incorporation of measurements of PG turnover and changes in PG architecture will be necessary to resolve the new questions raised by our study.

To help further clarify our model, we made the following text updates:

“To be able to maintain curvature in the presence of MreB, the curved-rod shaped Gram-negative Proteobacteria Caulobacter crescentus and Vibrio cholerae appear to limit relative levels of PG synthesis at negative curvatures through the action of long, cell-spanning cytoskeletal filaments (CreS and CrvA) that preferentially localize to the minor axis (negative Gaussian curvature) and enable cells to increase relative synthesis rates on the opposite side of the wall (positive Gaussian curvature) (Bartlett et al., 2017; Cabeen et al., 2009).”

“*H. pylori* leverages the bactofilin CcmA, which localizes preferentially to the major helical axis area, to promote synthesis at positive Gaussian curvatures on the sidewall, and supplements the MreB associated enhanced synthesis that is enriched at negative Gaussian curvatures (the minor helical axis) (Figure 11A, right). Adding the contribution of CcmA to the PG synthesis patterning allows *H. pylori* to maintain curvatures in the presence of MreB-associated PG synthesis.”

“Overall, our results are consistent with a model in which MreB-patterned straight-rod shape is the default pattern for *H. pylori* cells and helical shape is facilitated by adding major axis area PG synthesis via CcmA to augment straight-rod cell wall patterning.”

“Furthermore, our labeling strategy allowed us to determine the curvature bias of new PG insertion, but spatially-regulated turnover of old PG may also contribute to cell wall homeostasis.

The authors seem to rule out mutual exclusion mechanisms (i.e. MreB occupies cylindrical parts that CcmA does not occupy) because in subsection “CcmA localization to positive curvature correlates with cell wall synthesis, CcmA polymerization, andhelical cell shape” they conclude that MreB localization is the same in ∆ccmA as in WT. However, there appears to be a difference wherein the relative concentration increases more for MreB at negative curvatures in ∆ccmA than it does in WT and (as the authors actually note later in that section) there is a slight peak of MreB enrichment at ~3 µm^-2^ in the mutant but not in WT. There also appears to be a dip in concentration around 10-15 µm^-2^. Importantly, the differences in MreB distribution look to be of a similar magnitude as the differences in D-Ala-alk incorporation between WT and ∆ccmA but the D-Ala-alk incorporation pattern is described as different.

At present our data do not support or rule out mutual exclusion mechanisms because we did not do colocalization studies. Our data indicate that both MreB and CcmA can occupy the same curvatures although at different frequencies. Interpreting the nuanced features of the MreB enrichment curve in Δ*ccmA* is complicated by the non-uniform enrichment of preimmune signal. The relative concentration of MurNAc-alk and D-Ala-alk signal at negative Gaussian curvature is markedly higher in Δ*ccmA* cells as compared to wild-type. As the plots show relative signal enrichment, this is consistent with the observed marked reduction in enrichment at positive Gaussian curvature. Now that we are able to include the MurNAc-alk data, the difference between labeling patterns in wild-type and Δ*ccmA* are even more pronounced. These data are consistent with MreB-patterned synthesis playing a dominant role in the overall distribution of PG synthesis, with some minimal enhanced synthesis at positive Gaussian curvature not patterned by CcmA. We have updated the text to clarify and expand upon these points in the Discussion:

“MreB curvature preference appears largely similar in both wild-type (HJH1, *amgK murU*, light pink) and ΔccmA with poles excluded (JTH6, *amgK murU* Δ*ccmA*, dark pink) (Figure 11A). […] There is a small peak for MreB at approximately 3 µm^-2^, however interpretation of the MreB peak is complicated by the presence of a peak at the same curvature range for the preimmune signal.”

“Furthermore, we demonstrate that in the absence of CcmA, similarly to in wild-type, MreB is still enriched at negative Gaussian curvature, but that MurNAc-alk and D-Ala-alk synthesis patterning shift to more closely resemble the MreB curvature enrichment profile[…] CcmA is one of a suite of proteins required for helical cell shape maintenance; it is possible that other cell shape proteins can influence PG synthesis to promote some limited curvature in the absence of CcmA, consistent with multiple complementary mechanisms being required for helical shape maintenance.”

2) For the biochemical characterization of CcmA and mutant forms of CcmA, it would be important to note in the text (not just the Materials and methods) that the proteins are purified in denatured form and refolded to induce/monitor polymer formation. This is important because it does somewhat complicate interpretation of the mutants. It could be that the I55A or L110S mutations influence the ability of CcmA to fold properly during renaturation, rather than having a direct effect on polymerization capacity. The punctate cytoplasmic signal by IF could be consistent with either effect.

WT CcmA, CcmA^I55A,^and CcmA^L110S^ were not purified under denaturing conditions. While the elution buffer used contains 2 M urea, this is not a sufficient concentration of urea to denature CcmA. In the elution buffer directly off of the column, WT CcmA and CcmA^L110S^ form individual filaments as well as bundles of filaments when visualized by transmission electron microscopy (TEM) and negative staining while CcmA^I55A^ does not form any detectable structures (Author response image 1 and magnified insets). These data suggest that when we purify WT CcmA and CcmA^L100S^ in the presence of 2 M urea the proteins are already folded and spontaneously polymerize. Thus, refolding is not occurring when the proteins are dialyzed against our 25 mM Tris pH 8 buffer used in our experiments in the manuscript because the proteins are already folded.

In another set of experiments not included in this manuscript we found that 8 M urea is sufficient to denature WT CcmA and CcmA^L110S^ and reduce all filament, bundle, and lattice structures observable by TEM and negative staining (Figure 1 D, E). Our lab is continuing to characterize CcmA in ongoing projects and have analyzed the structure of WT CcmA by Circular Dichroism spectroscopy. These data indicate that WT CcmA forms the predicted beta helix structure and can rapidly refold after heat denaturation.

Related to this, the authors say that the mutants "failed to form any higher order structure under any buffer condition tested". What conditions were tested?

We tested polymerization of WT CcmA, CcmA^I55A^, and CcmA^L110S^ in five different buffers chosen based on other published studies of bactofilins. WT CcmA and CcmA^L110S^ show changes in the fraction of filaments observed in larger bundles based on pH but both proteins form filaments in all five buffers. CcmA^I55A^ was unable to form any filaments, bundles, or lattices detectable by TEM and negative staining in any of the buffers. The buffers we have tested are:

· Elution buffer: 25 mM tris pH 8, 2 M urea, 500 mM NaCl, 2% glycerol, 250 mM imidazole

· 25 mM tris pH 8, 20 mM glycine (Zuckerman et al., 2015, PLoS One)

· 25 mM tris pH 8 (Vasa et al., 2014, PNAS)

· 25 mM CAPS pH 11 (similar to a buffer used in Deng et al., 2019, Nature Microbiology)

Did they test whether the mutants that cannot polymerize were soluble when purified?

As explained in the Materials and methods section of the manuscript, to purify CcmA we overexpressed polyhistidine-tagged versions of WT CcmA, CcmA^L110S^, or CcmA^I55A^, in *E. coli*, lysed the cells by sonication, and purified CcmA using nickel affinity resin from the soluble fraction of the sonicated cells. Each time we performed this protocol we recovered very similar yields of WT CcmA, CcmA^L110S^ and CcmA^I55A^ (see SDS-PAGE gel of purified proteins in Figure 8—figure supplement 1D).